# Modeling Human Beliefs about AI Behavior for Scalable Oversight

**Leon Lang**                                                                                   *l.lang@uva.nl*
*University of Amsterdam*

**Patrick Forré**                                                                               *p.d.forre@uva.nl*
*University of Amsterdam*

**Reviewed on OpenReview:** *https://openreview.net/forum?id=gSJfsdQnex*

## Abstract

As AI systems advance beyond human capabilities, scalable oversight becomes critical: how can we supervise AI that exceeds our abilities? A key challenge is that human evaluators may form incorrect *beliefs* about AI behavior in complex tasks, leading to unreliable feedback and poor value inference. To address this, we propose *modeling* evaluators' beliefs to interpret their feedback more reliably. We formalize human belief models, analyze their theoretical role in value learning, and characterize when ambiguity remains. To reduce reliance on precise belief models, we introduce "belief model covering" as a relaxation. This motivates our preliminary proposal to use the internal representations of adapted foundation models to mimic human evaluators' beliefs. These representations could be used to learn correct values from human feedback even when evaluators misunderstand the AI's behavior. Our work suggests that modeling human beliefs can improve value learning and outlines practical research directions for implementing this approach to scalable oversight.

## Contents

# 1 Introduction

In recent years, reinforcement learning from human feedback (RLHF) (Christiano et al., 2017) and variations like direct preference optimization (Rafailov et al., 2023) have become a practical approach for aligning language models (Ziegler et al., 2020; Stiennon et al., 2022; Bai et al., 2022; Ouyang et al., 2022). These techniques have then been used in the alignment of large-scale foundation models like ChatGPT (OpenAI, 2022) and GPT-4 (OpenAI et al., 2024), Gemini (Gemini Team, 2023), Llama (Touvron et al., 2023; Grattafiori et al., 2024), and Claude (Bai et al., 2022; Anthropic, 2023a;b).

RLHF uses feedback of human evaluators on AI behavior to give information about desired behavior. But as AI systems become more capable, they may eventually outstrip our ability to evaluate them. The problem of **scalable oversight** therefore asks how to effectively teach our preferences to AI systems when they become more capable than ourselves (Amodei et al., 2016; Christiano et al., 2018; Irving et al., 2018; Leike et al., 2018; Bowman et al., 2022). Conceptually, if human evaluators lack the capacity to fully understand the AI's behavior, then this may incentivize the AI to perform behavior that the human evaluator only **believes** to be good, possibly at the expense of actual value.

Recent work has in fact revealed problems with RLHF that stem from erroneous human beliefs about AI behavior. In an early example (Amodei et al., 2017), an AI was supposed to grasp a ball with a robot hand in a simulation. The evaluators, who looked at a video of the behavior, sometimes believed that the hand was grasping the ball when it was in fact only *in front of* it, leading to positive feedback for the wrong behavior. In Cloud et al. (2024), a synthetic overseer in a baseline setting is unaware of whether a reached goal is a diamond (positive) or ghost (negative), leading the policy to learn to approach the ghost. Denison et al. (2024) shows a language model modifying a file unbeknownst to a synthetic evaluator, successfully deceiving it into believing a checklist has been accomplished. Similar models have also been shown to mislead humans who are limited in their time or competence to evaluate question answering or programming tasks, leading them to believe that the performance is better than it is (Wen et al., 2024).

Given the crucial role of evaluator beliefs in erroneous feedback, in this theoretical work, we are exploring the idea to *model* human beliefs about AI behavior. Our idea is that if we knew what the human *believes* the AI has done in its environment, then we could properly relate the human's feedback to that believed behavior instead of to the actual behavior. For example, imagine the robot hand from Amodei et al. (2017) is in front of the ball and the human evaluator gives positive feedback. Additionally, assume we know the human believes the ball was *actually grasped*. If so, then we know that the human thinks *grasping the ball* is positive, and can incentivize this behavior from now on.

But what *are* beliefs? In classical work on partial observability, beliefs are formalized as probability distributions over environment states (Kaelbling et al., 1998), possibly including models of other agents (Gmytrasiewicz & Doshi, 2005). In Lang et al. (2024), human evaluator beliefs are given by probability distributions over state sequences in the AI's environment. They show in theoretical toy examples that modeling such beliefs can help to learn effectively from feedback. However, in reality it is unrealistic that humans form probability distributions over entire state sequences, and it would be prohibitive to specify such a belief explicitly. We think it is more realistic that humans think in terms of **features**, which we conceptualize as higher-level and independent properties of trajectories in an environment. We then model a human belief about an AI's behavior as a vector of feature strengths.

This leads to our notion of a **human belief model**, which we introduce in Section 2. This model, together with feedback on observations, then theoretically allows to infer the return function over trajectories up to an inherent **ambiguity**, which we characterize in terms of the human belief model. The ambiguity disappears when the model is **complete**, which holds, in particular, when the human's beliefs of all observations linearly cover the feature combinations that are possible in the environment (Theorem 2.13). We then analyze conceptual toy examples of non-complete and complete models and consequences for the resulting return function inference.

We would like to use human belief modeling in practice to make concrete progress on scalable oversight. However, realistically, we cannot model human beliefs precisely; after all, we do not even have an explicit specification of the human's feature set! In Section 3, we thus relax the requirement of exact modeling

by investigating belief models that can **represent** all return functions and feedback functions that are compatible with the *true* belief model. We then say that these models **cover** the true belief model. When such a model is complete, it can replace the true model for the return function inference (Theorem 3.2). In a conceptual example, we analyze a human with symmetry-invariant features, which we can cover with a complete model that assumes symmetry-invariant reward functions.

We also characterize model coverage by the existence of what we call **model morphisms** and **linear belief-compatible ontology translations** (Theorem 3.5). Intuitively, this means that if one can *linearly translate* from the specified model's features to the human's features in a way that is compatible with their beliefs about observations, then coverage holds. The notion of a linear ontology translation is strikingly similar to work in the field of mechanistic interpretability on sparse autoencoders (Cunningham et al., 2023; Bricken et al., 2023), which linearly map from the internal representation space of language models to a space of human-interpretable features.

Inspired by this connection, in Section 3.4, we make a preliminary proposal to use adapted foundation models to construct a belief model that covers the true human belief model. The resulting training scheme would work like typical reinforcement learning from human feedback, but use an adapted foundation model with internal representations that mimic the human's beliefs of the AI's behavior. After learning the reward model, it can then be applied during policy optimization by using a more capable foundation model for creating the internal representations. Our theory specifies precise conditions under which the procedure leads to a correct return function. This training scheme does not assume the human to fully understand the behavior they evaluate and is not based on amplification of the human's capabilities, which we think provides a new angle to make progress on the problem of scalable oversight.

Finally, in our discussion in Section 4, we summarize our work, survey related work, motivate theoretical and empirical future work, and conclude.

## 2 Human belief models

In this section we define the notion of human belief models that can help to infer return functions from the human's feedback.

In Section 2.1 we start by explaining our conventions and some preliminaries for linear algebra. We recommend also experienced readers to briefly skim this section to understand our notation. In Section 2.2, we briefly define Markov decision processes and slightly adapt them: We do *not* make the assumption that the return function over trajectories comes from a reward function. This allows us to be slightly more general. In Section 2.3, we then introduce the notion of the human's *ontology*, which is a map that assigns a vector of feature strengths to each trajectory. Our crucial assumption is that the return function evaluates a trajectory by summing up the rewards of each feature, weighted by the feature strengths. This allows us to recover classical reward functions as a special case.

In reinforcement learning, the goal is to maximize the return function, but we assume this function to be unknown: It represents the implicit values of a human evaluator and needs to be inferred from the human's *feedback*. In Section 2.4, we explain the human to give feedback based on forming a (possibly incorrect) *belief over features* from *observations*. We call the resulting feedback the *observation return function*. In Section 2.5 we then explain how our problem setup connects to the general problem of scalable oversight and motivate our solution approach: It is based on leveraging *human belief models* to better interpret the human's feedback; we introduce them in Section 2.6. When a model of the human's beliefs is known, one can infer the return function from the human's feedback up to an *ambiguity*, which we characterize in Section 2.7. We also define *complete* human belief models, for which the ambiguity vanishes, and characterize them in Theorem 2.13. In Section 2.8, we introduce the dual notion of faithfulness. In Section 2.9, we study conceptual examples that give an intuition for when the ambiguity does, or does not, vanish.

### 2.1 Conventions and preliminaries for linear algebra

Let $X$ be a set. For $x \in X$, we define the delta function $\delta_x : X \to \mathbb{R}$ as usual by

$$\delta_x(x') = \begin{cases} 1, & x' = x, \\ 0, & \text{else .} \end{cases}$$

Let $\mathbb{R}$ be the real numbers. Then $\mathbb{R}^X$ denotes the vector space of functions from $X$ to $\mathbb{R}$, which one can also view as column-vectors indexed by $x \in X$. For a function $v \in \mathbb{R}^X$, we write $v(x)$ for the entry of $v$ at position $x \in X$. The standard basis functions of $\mathbb{R}^X$ are given by $\{e_x\}_{x \in X}$ with $e_x = \delta_x$. Then every function $v \in \mathbb{R}^X$ can be written as $v = \sum_{x \in X} v(x) \cdot e_x$. We define the standard scalar product $\langle \cdot, \cdot \rangle : \mathbb{R}^X \times \mathbb{R}^X \to \mathbb{R}$ by

$$\langle v, w \rangle := \sum_{x \in X} v(x) w(x).$$

Let $A : \mathbb{R}^X \to \mathbb{R}^Y$ be a linear map. Then we view $A$ also as a matrix with matrix elements $A_{yx} := [A(e_x)](y) \in \mathbb{R}$ for $x \in X$, $y \in Y$. We write $A_y \in \mathbb{R}^X$ for the row of $A$ at index $y \in Y$, which is the function with entries $A_y(x) = A_{yx}$. Consequently, if $v \in \mathbb{R}^X$ and $y \in Y$, then we obtain

$$[A(v)](y) = \sum_{x \in X} v(x) [A(e_x)](y) = \sum_{x \in X} A_{yx} v(x) = \langle A_y, v \rangle.$$

This corresponds to the typical way that linear functions can be represented as matrix-vector products. If $\mathcal{V} \subseteq \mathbb{R}^X$ is a vector subspace, then we write $A|_\mathcal{V} : \mathcal{V} \to \mathbb{R}^Y$ for the restriction of $A$ to $\mathcal{V}$, which is simply given by $(A|_\mathcal{V})(v) = A(v)$ for all $v \in \mathcal{V}$. We denote the kernel and image of $A : \mathbb{R}^X \to \mathbb{R}^Y$ by

$$\ker(A) := \left\{ v \in \mathbb{R}^X \mid A(v) = 0 \right\} \subseteq \mathbb{R}^X,$$

$$\operatorname{im}(A) := \left\{ w \in \mathbb{R}^Y \mid \exists v \in \mathbb{R}^X : A(v) = w \right\} \subseteq \mathbb{R}^Y.$$

I.e., they are the set of functions in $\mathbb{R}^X$ that are sent to zero by $A$, and the set of functions in $\mathbb{R}^Y$ that are hit by $A$, respectively. Both are vector subspaces of their respective surrounding vector spaces. We explain basic properties of kernels and images in Appendix A.2 and will refer to those results when needed.

Sometimes, our linear functions "come from" a deterministic function in the other direction. I.e., if $h : Y \to X$ is a function, then we define the linear dual function $h^* : \mathbb{R}^X \to \mathbb{R}^Y$ for $v \in \mathbb{R}^X$ and $y \in Y$ by

$$[h^*(v)](y) := v(h(y)). \tag{1}$$

The matrix elements are then given by

$$h^*_{yx} = [h^*(e_x)](y) = e_x(h(y)) = \begin{cases} 1, & h(y) = x, \\ 0, & \text{else.} \end{cases} \tag{2}$$

For two linear maps $A : \mathbb{R}^X \to \mathbb{R}^Y$ and $B : \mathbb{R}^Y \to \mathbb{R}^Z$, we write their composition as $B \circ A : \mathbb{R}^X \to \mathbb{R}^Z$, which has matrix elements $(B \circ A)_{zx} = \sum_{y \in Y} B_{zy} A_{yx}$ for $x \in X$, $z \in Z$. This also implies

$$(B \circ A)_z = \sum_{y \in Y} B_{zy} A_y. \tag{3}$$

Finally, whenever we draw a diagram of (usually linear) functions in this paper, the diagram *commutes*, meaning that all directed pathways from one node to another node are the same function. For example, in a diagram of the form

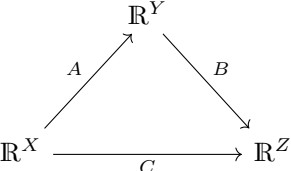

we have $C = B \circ A$. The only exceptions to this convention are Equations (22) and (28) in the appendix.

## 2.2 Preliminaries on Markov Decision Processes

Much prior work involving human feedback, especially in RLHF, considers the contextual bandit framework (Ziegler et al., 2020). However, we are motivated to contribute to a theory that will apply to advanced AI systems interacting with more complex environments, as this is where we expect most of the risks of advanced AI to be. We thus work in the setting of an **MDP** $(\mathcal{S}, \mathcal{A}, \mathcal{T}, P_0, T, G)$, with a finite set of **states** $\mathcal{S}$ and **actions** $\mathcal{A}$, a **transition kernel** $\mathcal{T} : \mathcal{S} \times \mathcal{A} \to \Delta(\mathcal{S})$, an **initial state distribution** $P_0 \in \Delta(\mathcal{S})$, a **finite time horizon** $T \in \{0, 1, 2, 3, \dots\}$, and the human's implicit **return function** $G$, which we clarify after defining trajectories below. We fix this generic MDP for the rest of the paper.

We now define trajectories. In the rest of the paper, whenever we have a state-action sequence $\xi \in (\mathcal{S} \times \mathcal{A})^T \times \mathcal{S}$ present in some context and then write about states and actions $s_0, a_0, \dots, s_{T-1}, a_{T-1}, s_T$, then they implicitly refer to the states and actions in $\xi$. Then the set of **trajectories** be given by

$$\Xi = \left\{ \xi \in (\mathcal{S} \times \mathcal{A})^T \times \mathcal{S} \mid \xi \text{ is } possible \right\} \subseteq (\mathcal{S} \times \mathcal{A})^T \times \mathcal{S},$$

where we call a state-action sequence $\xi$ *possible* if $P_0(s_0) > 0$ and $T(s_t \mid s_{t-1}, a_{t-1}) > 0$ for all $t \geq 1$. Then the return function is a function $G \in \mathbb{R}^\Xi$.

Note that this formalism does *not* assume that $G$ decomposes into a reward function $R : \mathcal{S} \times \mathcal{A} \times \mathcal{S} \to \mathbb{R}$. In other words, the formalism allows for human preferences that do not satisfy the reward hypothesis (Bowling et al., 2022) and is thus slightly more general than typical reinforcement learning settings.

**Policies** are functions $\pi : \mathcal{S} \to \Delta(\mathcal{A})$. A policy $\pi$ together with the MDP induces a distribution $P^\pi \in \Delta(\Xi)$ over trajectories by sampling initial states from $P_0$, actions from $\pi$, and transitions from $\mathcal{T}$. **The policy evaluation function** is then given by

$$J(\pi) := \mathop{\mathbf{E}}_{\xi \sim P^\pi(\cdot)} \left[ G(\xi) \right]. \tag{4}$$

The goal in reinforcement learning is to find a policy $\pi$ that maximizes this evaluation function.

## 2.3 The human's ontology and reward object

We assume that the return function $G : \Xi \to \mathbb{R}$ encodes what a given human cares about. We imagine that $G$ measures the quality of each trajectory linearly in certain feature strengths associated to each trajectory; here, features are high-level return-relevant concepts.

More precisely, we assume the human comes equipped with a finite **feature set** $\Omega$. The human's **ontology** is a function $\lambda : \Xi \to \mathbb{R}^\Omega$ that encodes for each trajectory $\xi \in \Xi$ and feature $\omega \in \Omega$ the extent $\left[\lambda(\xi)\right](\omega)$ to which the feature $\omega$ is present in the trajectory $\xi$. As we discuss in greater detail in Section 2.4, these feature strengths $\lambda(\xi)$ are *idealized*, i.e., the human may not be able to compute them. In the general theory, we allow them to be negative, though it may help to imagine them to be non-negative. The human's **reward object** is a fixed function $R_\Omega \in \mathbb{R}^\Omega$ that assigns to each feature $\omega \in \Omega$ the degree $R_\Omega(\omega)$ to which the human likes this feature. The return function $G : \Xi \to \mathbb{R}$ evaluates a trajectory by summing up the quality of all features, weighted by the extent to which they appear in the trajectory:

$$G(\xi) = \sum_{\omega \in \Omega} \left[\lambda(\xi)\right](\omega) \cdot R_\Omega(\omega) = \langle \lambda(\xi), R_\Omega \rangle. \tag{5}$$

Let $\mathbf{\Lambda} : \mathbb{R}^\Omega \to \mathbb{R}^\Xi$ be given by $\big[\mathbf{\Lambda}(\tilde{R}_\Omega)\big](\xi) := \big\langle \lambda(\xi), \tilde{R}_\Omega \big\rangle$. Then $\mathbf{\Lambda}$ is a linear function from which we can recover $\lambda$ using the matrix elements of $\mathbf{\Lambda}$: $\big[\lambda(\xi)\big](\omega) = \mathbf{\Lambda}_{\xi\omega}$ (Proposition A.1). We slightly abuse the terminology by referring to *both* $\lambda : \Xi \to \mathbb{R}^\Omega$ and $\mathbf{\Lambda} : \mathbb{R}^\Omega \to \mathbb{R}^\Xi$ as the human's ontology. This representation of the ontology satisfies $\mathbf{\Lambda}(R_\Omega) = G$.

**Example 2.1** (Classical reward functions). *Assume that the human's feature set is $\Omega = \mathcal{S} \times \mathcal{A} \times \mathcal{S}$, i.e., the set of all state-action-state transitions. Then the reward object $R_\Omega = R \in \mathbb{R}^{\mathcal{S} \times \mathcal{A} \times \mathcal{S}}$ is a reward function in the classical sense. Let $\gamma \in [0,1]$ be a discount factor and define the ontology $\mathbf{\Lambda} = \mathbf{\Gamma} : \mathbb{R}^{\mathcal{S} \times \mathcal{A} \times \mathcal{S}} \to \mathbb{R}^\Xi$ by*

$$\big[\mathbf{\Gamma}(\tilde{R})\big](\xi) = \sum_{t=0}^{T-1} \gamma^t \tilde{R}(s_t, a_t, s_{t+1}).$$

*$G = \mathbf{\Gamma}(R)$ is then given by the discounted sum of rewards of individual transitions in a trajectory, as is typical in reinforcement learning. The matrix elements of the ontology $\mathbf{\Gamma}$ are given by*

$$\mathbf{\Gamma}_{\xi,(s,a,s')} = \big[\mathbf{\Gamma}(e_{(s,a,s')})\big](\xi) = \sum_{t=0}^{T-1} \gamma^t e_{(s,a,s')}(s_t, a_t, s_{t+1}) = \sum_{t=0}^{T-1} \gamma^t \delta_{(s,a,s')}(s_t, a_t, s_{t+1}),$$

*In other words, the extent to which the "feature" $(s, a, s')$ is present in the trajectory $\xi$ is simply the discounted number of times that it appears.*

**Remark 2.2** (Linearity assumption). *Equation* (5) *encodes that the true return function is* linear *in the features of the human's ontology. This is an important assumption that simplifies our theory substantially, but it also means that our results might hinge on the assumption being true. Is this justified?*

*First of all, linearity assumptions in reward learning are common, for example in Ng et al. (2000); Abbeel & Ng (2004); Ziebart et al. (2008) for inverse reinforcement learning or in parts of Song et al. (2024) for the case of preference finetuning. Additionally, a priori, our linearity assumption is quite flexible. For example, if the feature set is large enough, then any function can be considered linear in such features, as the case of classical reward functions in Example 2.1 demonstrates: In that case, the return functions are linear in features given by state-action-state transitions.*

*Yet later in this work, we don't* only *want to theoretically assume the human's return function to obey a linearity assumption — we also want to be able to concretely* model *the underlying features. If the features that allow for a linear return function get increasingly complex, we expect it to become more difficult to model those correctly in practice. We thus consider it an open question to what extent assuming linearity is appropriate, and we hope for future theoretical work that relaxes this assumption (cf. Section 4.3).*

## 2.4 The human's feature belief and observation return function

In standard reinforcement learning, it is typically assumed that we know the return function $G$ to be maximized by a policy. However, $G$ is the implicit return function of a given human. We must thus rely on *feedback* by the human to infer information about $G$. A naive idea would be to show the human each trajectory $\xi$, ask them to evaluate it by computing $G(\xi)$, and to use these returns to train a policy. There are three challenges with this plan:

(i) The human might not have access to the full trajectory $\xi$ or all trajectories $\xi \in \Xi$. For example, in complex environments, they would usually only receive partial observations.

(ii) Even if $\xi$ were fully accessible, the human might not have the *capacity* to compute the features $\lambda(\xi)$. For example, if $\xi$ encodes a proof-attempt of a mathematical conjecture and $\omega$ encodes "correctness", then the human may not have the competence to determine the extent $\big[\lambda(\xi)\big](\omega)$ to which $\xi$ is correct.

(iii) Even if the human could fully compute $\lambda(\xi)$, they may not have full access to their reward object $R_\Omega$ in order to then compute $G(\xi)$ as an explicit number.

To address (i), we assume a set of **observations** $\mathcal{O}$ that the human can receive. They may come from an observation function $O : \Xi \to \mathcal{O}$, as in some examples from Amodei et al. (2017); Lang et al. (2024); Cloud

et al. (2024); Denison et al. (2024). For example, $O(\xi)$ could be a sequence of observation segments, one for each time-step. Alternatively, $\mathcal{O}$ could be a set of simulations to probe the human's opinion on specific situations that *may* be present in real trajectories . We can also imagine $\mathcal{O}$ to be a subset of $\Xi$. For example, it can make sense to only show trajectories to the human if they are able to correctly evaluate them, as in easy-to-hard generalization (Sun et al., 2024; Hase et al., 2024; Ding et al., 2024). In any case, we assume $\mathcal{O}$ to be a fixed set of observations where, for the most part, we are agnostic about the process that generates them.

To address (ii), we assume the human then has a **feature belief function**, i.e. a function $\epsilon : \mathcal{O} \to \mathbb{R}^\Omega$ that encodes for each $o \in \mathcal{O}$ the extent $\big[\epsilon(o)\big](\omega)$ to which the human *believes* the feature $\omega$ to be present in the observation $o$ or an associated unknown trajectory $\xi$. Even if $\mathcal{O} = \Xi$, we can have $\epsilon \neq \lambda$, which reflects that the human may believe the features of a trajectory to be different from what they actually are.

The human then judges observations $o \in \mathcal{O}$ according to the **observation return function** $G_\mathcal{O} \in \mathbb{R}^\mathcal{O}$. It evaluates an observation by summing up the quality of all features, weighted by the extent to which the human *believes* them to be present:

$$G_\mathcal{O}(o) := \sum_{\omega \in \Omega} \big[\epsilon(o)\big](\omega) \cdot R_\Omega(\omega) = \big\langle \epsilon(o), R_\Omega \big\rangle. \tag{6}$$

Let $\mathcal{E} : \mathbb{R}^\Omega \to \mathbb{R}^O$ be given by $\big[\mathcal{E}(\tilde{R}_\Omega)\big](o) := \big\langle \epsilon(o), \tilde{R}_\Omega \big\rangle$. Then as for $\mathbf{\Lambda}$, $\mathcal{E}$ is a linear function from which we can recover $\epsilon$ using the matrix elements of $\mathcal{E}$: $\big[\epsilon(o)\big](\omega) = \mathcal{E}_{o\omega}$ (Proposition A.1). We again slightly abuse the terminology by referring to *both* $\epsilon : \mathcal{O} \to \mathbb{R}^\Omega$ and $\mathcal{E} : \mathbb{R}^\Omega \to \mathbb{R}^\mathcal{O}$ as the human's feature belief function. This representation of the featre belief function satisfies $\mathcal{E}(R_\Omega) = G_\mathcal{O}$.

To address (iii), in RLHF it is typically assumed that the human can make *choices* between observations (Christiano et al., 2017) that *implicitly* reflect the underlying return function (or, on our case, observation return function). We discuss this setting in Appendix B for the special case that all relevant linear functions are row-constant. However, in the main paper, we do not want to overcomplicate the theory and assume that the human can *directly compute* $G_\mathcal{O}$. In principle, if we would show each $o \in \mathcal{O}$ to the human, we could then record the entire observation return function $G_\mathcal{O}$. Thus, we assume $G_\mathcal{O}$, which represents the human's feedback, to be fully known.

**Remark 2.3** (Comparison to other work involving partial observations). *Since we work with a set of observations $\mathcal{O}$, we should clarify that we do* not *work in a classical POMDP setting. In POMDPs (Kaelbling et al., 1998), the* agent *only partially observes the environment, and may then keep track of a belief state for making decisions in the environment and maximizing reward. In contrast, we work in an MDP setting in which the agent's policy $\pi : \mathcal{S} \to \Delta(\mathcal{A})$ fully observes the environment state $s \in \mathcal{S}$, see Section 2.2. It is the human evaluator who may observe the environment only partially for evaluating the policy.*

*More closely related is the framework of I-POMDPs (Gmytrasiewicz & Doshi, 2005), which is a multi-agent setting in which each agent maintains a belief over not only the environment, but also over other agents and their beliefs. Recent work on partially observable assistance games (Emmons et al., 2024) provides a special case that is more similar to our setting by considering an agent and a human in a shared environment in which the agent is trying to maximize the human's latent reward function. In that work, the human is assumed to form a* calibrated *belief over* entire environment states, *whereas we consider* potentially faulty *beliefs over trajectory features. We think it might be useful to attempt a synthesis of both settings (cf. Sections 4.2 and 4.3.1).*

## 2.5 Connection to the problem of scalable oversight

In this interlude, we explain how the setup that we developed over the previous subsections relates to the problem of scalable oversight (Amodei et al., 2016; Christiano et al., 2018; Irving et al., 2018; Leike et al., 2018; Bowman et al., 2022). We also motivate our solution approach of modeling human beliefs.

Scalable oversight is the general problem of how to oversee AI systems when they are more capable than the human overseers. While it is difficult to directly model the capabilities of AI systems, we can at least note that highly capable AI systems will plausibly act in *large-scale environments* and perform behaviors whose

precise meaning may not be adequately *understood* by humans. This can be modeled in the setting that we developed earlier:

- **Scale**: If the AI acts in an environment with trajectories $\xi \in \Xi$, human overseers may not be able to observe the entirety of the trajectories. We model this by a set of potentially smaller observations $o \in \mathcal{O}$ on which humans give feedback.

- **Understanding**: We model the human's *understanding* of an observation by its feature belief $\epsilon(o) \in \mathbb{R}^\Omega$. In cases where an observation $o \in \mathcal{O}$ belongs to a trajectory $\xi \in \Xi$ in the environment (e.g., by observing a subset or other transformation of the trajectory), a gap in the human's understanding can be characterized by $\epsilon(o) \neq \lambda(\xi)$: The human's understanding of the observation is different from the true features of the corresponding trajectory.

How do these assumptions impact the return function inference? Naively, if we did not properly account for the human's limitations, we might *assume* that the human directly observes the full trajectory $\xi$ and assesses it correctly. If we then try to learn a return function $\tilde{G} : \Xi \to \mathbb{R}$ and observe the human's feedback $G_\mathcal{O}(o)$, we make the inference $\tilde{G}(\xi) = G_\mathcal{O}(o) = \langle \epsilon(o), R_\Omega \rangle$, which can differ from the true return $G(\xi) = \langle \lambda(\xi), R_\Omega \rangle$ since $\epsilon(o) \neq \lambda(\xi)$. Now, assume there is a feature $\omega \in \Omega$ that is very bad ($R_\Omega(\omega) \ll 0$) and present in the true trajectory ($[\lambda(\xi)](\omega) > 0$), but not in the human's belief of the corresponding observation ($[\epsilon(o)](\omega) = 0$). Then this feature leads to a large negative contribution to $G(\xi)$, but not to $G_\mathcal{O}(o)$, which can lead to an overestimation of the inferred return for $\xi$:

$$\tilde{G}(\xi) = \langle \epsilon(o), R_\Omega \rangle > \langle \lambda(\xi), R_\Omega \rangle = G(\xi).$$

Thus, in the policy optimization phase using the inferred return function $\tilde{G}$, $\xi$ may be positively reinforced, leading to a policy with potentially bad performance as judged by the true return function $G$. This phenomenon is analyzed in a special case under the name *deceptive inflation* in Lang et al. (2024), and might also lie at the heart of the empirical failures observed in Amodei et al. (2017); Denison et al. (2024); Wen et al. (2024); Williams et al. (2024).

In our work, we attempt to solve this problem by viewing $G_\mathcal{O}(o) = \langle \epsilon(o), R_\Omega \rangle$ *not* as an estimate for the true return of a trajectory $\xi$, but instead as *evidence* for the human's reward object $R_\Omega$. The idea is that we assume to *know* the human's faulty belief $\epsilon(o)$, which means that $G_\mathcal{O}(o) = \langle \epsilon(o), R_\Omega \rangle$ is a linear equation that can be used to narrow down the space of possibilities for $R_\Omega$. Once $R_\Omega$ is sufficiently determined, and if $\lambda(\xi)$ is also known, we can then infer the true return function $G$ using the equation $G(\xi) = \langle \lambda(\xi), R_\Omega \rangle$. This approach complements other work on scalable oversight, which usually falls into the regime of *amplified oversight* (Leike et al., 2018; Saunders et al., 2022; Irving et al., 2018). Such work attempts to *improve* the human's ability to give accurate feedback, whereas our work attempts to learn the correct return function from potentially *inaccurate* feedback by leveraging additional information in the form of a known model of the human's beliefs. We then get closer to reality by relaxing the assumption of knowing the human's beliefs in Section Section 3.

## 2.6 The definition of human belief models

In order to infer the return function $G$ from the human's feedback $G_\mathcal{O}$, we assume the features $\Omega$, ontology $\Lambda$, and feature belief function $\mathcal{E}$ are all known; additionally, we assume we may have a priori knowledge of a vector subspace of **valid reward objects** $\mathcal{V} \subseteq \mathbb{R}^\Omega$ in which $R_\Omega$ resides: $R_\Omega \in \mathcal{V}$. This may come from any a priori knowledge, e.g. of certain symmetries in the environment that leave rewards invariant (cf. Example 2.19 and Section 3.3). This leads to the following notion:

**Definition 2.4** (Human belief model, representing). *Let $\Xi$ be the set of trajectories in an MDP and $\mathcal{O}$ the fixed set of observations that a human evaluator receives. Then a **human belief model** (or* belief model, *or* model, *for short) is a tuple $\mathcal{M} = (\Omega, \Lambda, \mathcal{E}, \mathcal{V})$, where:*

- $\Omega$ *is a set called feature set;*
- $\Lambda : \mathbb{R}^\Omega \to \mathbb{R}^\Xi$ *is a linear function, called ontology;*

- $\mathcal{E} : \mathbb{R}^{\Omega} \to \mathbb{R}^{\mathcal{O}}$ *is a linear function, called* feature belief function;

- *and* $\mathcal{V} \subseteq \mathbb{R}^{\Omega}$ *is a vector subspace, called* space of valid reward objects.

*A human belief model $\mathcal{M}$ represents the true return function $G : \Xi \to \mathbb{R}$ and observation return function $G_{\mathcal{O}} : \mathcal{O} \to \mathbb{R}$ if there is a reward object $R_{\Omega} \in \mathcal{V}$ with $G = \Lambda(R_{\Omega})$ and $G_{\mathcal{O}} = \mathcal{E}(R_{\Omega})$.*

*When appropriate, we will also use the representation $\lambda : \Xi \to \mathbb{R}^{\Omega}$ with $\big[\lambda(\xi)\big](\omega) = \Lambda_{\xi\omega}$ and $\epsilon : \mathcal{O} \to \mathbb{R}^{\Omega}$ with $\big[\epsilon(o)\big](\omega) = \mathcal{E}_{o\omega}$ of the ontology and feature belief function, respectively (cf. Proposition A.1). We visualize a model $\mathcal{M} = (\Omega, \Lambda, \mathcal{E}, \mathcal{V})$ as*

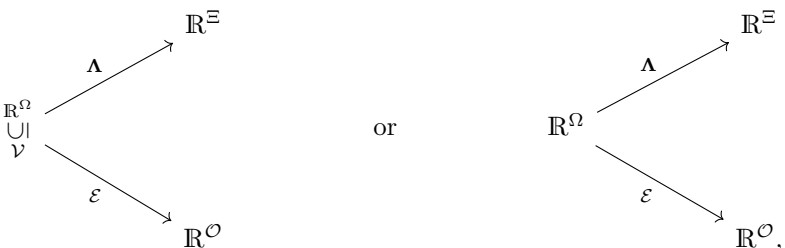

*where we use the latter version in the special case that $\mathcal{V} = \mathbb{R}^{\Omega}$.*

**Remark 2.5** (*Which* human belief model?)*. A priori, there can be* many *human belief models $\widehat{\mathcal{M}} = (\widehat{\Omega}, \widehat{\Lambda}, \widehat{\mathcal{E}}, \widehat{\mathcal{V}})$ that can represent the true return function $G$ and the human's feedback $G_{\mathcal{O}}$. For example, in Section 3.3 we will discuss a situation with three different models, and in Appendix C one with many more. In fact, there is always a trivial belief model that can represent $G$ and $G_{\mathcal{O}}$: View $\mathbb{R}$ as the $\mathbb{R}$-vectorspace $\mathbb{R}^{\{\star\}}$ of functions from a one-element set $\{\star\}$ to $\mathbb{R}$. Associate to $G : \Xi \to \mathbb{R} = \mathbb{R}^{\{\star\}}$ and $G_{\mathcal{O}} : \mathcal{O} \to \mathbb{R} = \mathbb{R}^{\{\star\}}$ via Proposition A.1 the linear functions $\mathrm{lin}(G) : \mathbb{R}^{\{\star\}} \to \mathbb{R}^{\Xi}$ and $\mathrm{lin}(G_{\mathcal{O}}) : \mathbb{R}^{\{\star\}} \to \mathbb{R}^{\mathcal{O}}$ given by*

$$\big[\mathrm{lin}(G)\big](r \cdot e_{\star}) \coloneqq r \cdot G, \quad \big[\mathrm{lin}(G_{\mathcal{O}})\big](r \cdot e_{\star}) \coloneqq r \cdot G_{\mathcal{O}}$$

*for $r \in \mathbb{R}$ and the only basis vector $e_{\star}$. Then, define the human belief model*

$$\mathcal{M} \coloneqq \big(\{\star\}, \ \mathrm{lin}(G), \ \mathrm{lin}(G_{\mathcal{O}}), \ \mathbb{R}^{\{\star\}}\big)$$

*with the feature set $\{\star\}$. This represents $G$ and $G_{\mathcal{O}}$ with the reward object $e_{\star} \in \mathbb{R}^{\{\star\}}$ since $\big[\mathrm{lin}(G)\big](e_{\star}) = G$ and $\big[\mathrm{lin}(G_{\mathcal{O}})\big](e_{\star}) = G_{\mathcal{O}}$. Intuitively, the feature $\star$ then means "goodness", the "ontology" $G : \Xi \to \mathbb{R}^{\{\star\}}$ measures the extent to which "goodness" is present in a trajectory, and the "feature belief function" $G_{\mathcal{O}} : \mathcal{O} \to \mathbb{R}^{\{\star\}}$ measures how much goodness the human believes to be present in an observation. This trivial model is not very interesting for our purposes since assuming that the human belief model is known would then presuppose knowledge of $G$ itself — which is the return function that we want to infer based on feedback $G_{\mathcal{O}}$.*

*A belief model $\mathcal{M} = (\Omega, \Lambda, \mathcal{E}, \mathcal{V})$ is more interesting if $\Omega$ consists of a variety of meaningful concepts such that $G$ and $G_{\mathcal{O}}$ decompose linearly over these features, in such a way that:*

*(i) It is easier to know the model $\mathcal{M}$ than to know the return function $G$;*

*(ii) Given $\mathcal{M}$ and $G_{\mathcal{O}}$, it is easy to determine $G$.*

*For the rest of this section, we imagine to know the "true" belief model $\mathcal{M}$, which we assume to have these properties. Note that these are philosophical assumptions that guide how we talk about $\mathcal{M}$ and how we imagine its application. But mathematically, all of our claims here and in the upcoming sections are correct whenever $\mathcal{M}$ represents $G$ and $G_{\mathcal{O}}$.*

*In Section 3, we then relax the requirement (i) of "knowledge" of $\mathcal{M}$, and specifically in Section 3.4 we make a proposal for model specification using foundation models. We then also show how $G$ could in principle be learned by learning $G_{\mathcal{O}}$ via supervised learning.*

**Example 2.6.** *Assume the setting from Example 2.1, in which $\Omega = \mathcal{S} \times \mathcal{A} \times \mathcal{S}$ and $\mathbf{\Lambda} = \mathbf{\Gamma} : \mathbb{R}^{\mathcal{S} \times \mathcal{A} \times \mathcal{S}} \to \mathbb{R}^{\Xi}$. In this example we add a feature belief function to specify a full human belief model.*

*Assume the human comes equipped with a belief function $b : \mathcal{O} \to \mathbb{R}^{\Xi}$ that assigns a probability distribution over trajectories $b(o) \in \Delta(\Xi) \subseteq \mathbb{R}^{\Xi}$ to each observation $o \in \mathcal{O}$. For example, this could be a posterior belief if $o$ is sampled by first sampling a trajectory $\xi$ according to some policy and then computing the observation as $o = O(\xi)$ for an obervation function $O : \Xi \to \mathcal{O}$. Then we assume that the feature belief function $\epsilon : \mathcal{O} \to \mathbb{R}^{\mathcal{S} \times \mathcal{A} \times \mathcal{S}}$ computes, for each $o \in \mathcal{O}$ and $(s, a, s') \in \mathcal{S} \times \mathcal{A} \times \mathcal{S}$, the expected discounted number of times that the transition $(s, a, s')$ appears in the trajectory $\xi$ that gave rise to observation $o$:*

$$\left[\epsilon(o)\right](s, a, s') := \sum_{\xi \in \Xi} \left[b(o)\right](\xi) \cdot \mathbf{\Gamma}_{\xi, (s, a, s')}.$$

*We now determine the matrix form of $\epsilon$. Let $\mathbf{B} : \mathbb{R}^{\Xi} \to \mathbb{R}^{\mathcal{O}}$ be the linear function given by $\left[\mathbf{B}(G)\right](o) := \langle b(o), G \rangle$, which has matrix elements $\mathbf{B}_{o\xi} = \left[b(o)\right](\xi)$ by Proposition A.1. We obtain*

$$\mathcal{E}_{o, (s, a, s')} = \left[\epsilon(o)\right](s, a, s') = \sum_{\xi \in \Xi} \mathbf{B}_{o\xi} \, \mathbf{\Gamma}_{\xi, (s, a, s')} = (\mathbf{B} \circ \mathbf{\Gamma})_{o, (s, a, s')}.$$

*Thus, $\mathcal{E} = \mathbf{B} \circ \mathbf{\Gamma}$.*

*Finally, assume that the set of valid reward objects is simply given by $\mathcal{V} = \mathbb{R}^{\mathcal{S} \times \mathcal{A} \times \mathcal{S}}$. Then, in total, our model is given by $(\Omega, \mathbf{\Lambda}, \mathcal{E}, \mathcal{V}) = (\mathcal{S} \times \mathcal{A} \times \mathcal{S}, \ \mathbf{\Gamma}, \ \mathbf{B} \circ \mathbf{\Gamma}, \ \mathbb{R}^{\mathcal{S} \times \mathcal{A} \times \mathcal{S}})$:*

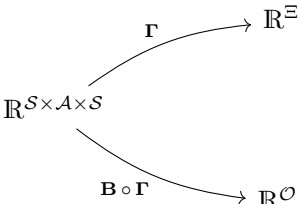

*One can compare this with Lang et al. (2024) and realize that our result coincides with their belief model, thus showing that we generalize their work.*

## 2.7 Complete belief models and the ambiguity

In this whole subsection, we fix an MDP with trajectories $\Xi$, observations $\mathcal{O}$, and corresponding return function $G \in \mathbb{R}^{\Xi}$ and observation return function $G_{\mathcal{O}} \in \mathbb{R}^{\mathcal{O}}$. We also fix a human belief model $\mathcal{M} = (\Omega, \mathbf{\Lambda}, \mathcal{E}, \mathcal{V})$ that represents $G$ and $G_{\mathcal{O}}$ with an implicit reward object $R_{\Omega} \in \mathcal{V}$: $\mathbf{\Lambda}(R_{\Omega}) = G$ and $\mathcal{E}(R_{\Omega}) = G_{\mathcal{O}}$. We can visualize the model together with the reward object and return functions as follows:

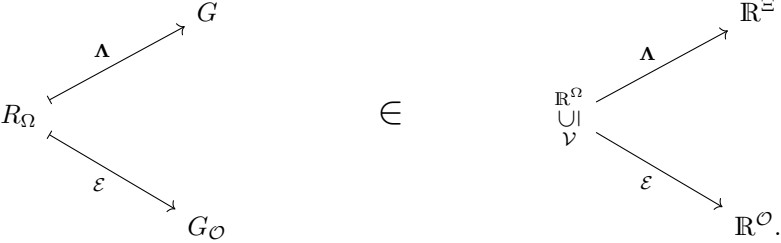

We are concerned with the following question:

**Question 2.7.** *Assume the human belief model $\mathcal{M}$ and the human feedback, in the form of the observation return function $G_{\mathcal{O}}$, are known. Under what conditions, or to what extent, can we infer the return function $G$?*

In Section 3, we will relax the assumption that the belief model is known precisely.

The idea for the answer is as follows: When trying to infer $G$ from $G_{\mathcal{O}}$, we make use of the knowledge that they are connected by the unknown reward object $R_\Omega \in \mathcal{V}$. Thus, one can first try to determine a reward object $\tilde{R}_\Omega \in \mathcal{V}$ with the correct observation return function $\mathcal{E}(\tilde{R}_\Omega) = G_{\mathcal{O}}$. Then the question is whether the corresponding return function $\mathbf{\Lambda}(\tilde{R}_\Omega)$ equals the true return function $G$. They differ by the return function $G' = \mathbf{\Lambda}(\tilde{R}_\Omega) - G$. The *ambiguity* will then be defined as the set of all these differences, and the question will be how to characterize it and when it vanishes, leading to the notion of a *complete* human belief model.

**Definition 2.8** (Feedback-compatible, ambiguity). *We define the set of return functions that are **feedback-compatible** with $G_{\mathcal{O}}$ as*

$$\mathrm{FC}^{\mathcal{M}}(G_{\mathcal{O}}) \coloneqq \left\{ \tilde{G} \in \mathbb{R}^\Xi \mid \exists \tilde{R}_\Omega \in \mathcal{V} \colon \mathcal{E}(\tilde{R}_\Omega) = G_{\mathcal{O}} \text{ and } \mathbf{\Lambda}(\tilde{R}_\Omega) = \tilde{G} \right\}.$$

*We define the **ambiguity** left in the return function $G$ after the observation return function $G_{\mathcal{O}}$ is known by*

$$\mathrm{Amb}^{\mathcal{M}}(G, G_{\mathcal{O}}) \coloneqq \left\{ G' \in \mathbb{R}^\Xi \mid G' = \tilde{G} - G \text{ for } \tilde{G} \in \mathrm{FC}^{\mathcal{M}}(G_{\mathcal{O}}) \right\}.$$

Then clearly, we have

$$\mathrm{FC}^{\mathcal{M}}(G_{\mathcal{O}}) = G + \mathrm{Amb}^{\mathcal{M}}(G, G_{\mathcal{O}}).$$

This leaves open the question of how to characterize the ambiguity and when it vanishes.

**Proposition 2.9** (Ambiguity characterization). *We have*

$$\mathrm{Amb}^{\mathcal{M}}(G, G_{\mathcal{O}}) = \mathbf{\Lambda}(\ker(\mathcal{E}) \cap \mathcal{V}).$$

*Proof.* For one direction, let $G' \in \mathrm{Amb}^{\mathcal{M}}(G, G_{\mathcal{O}})$. Then $G' = \mathbf{\Lambda}(\tilde{R}_\Omega) - G$ for $\tilde{R}_\Omega \in \mathcal{V}$ with $\mathcal{E}(\tilde{R}_\Omega) = G_{\mathcal{O}}$. By the linearity of $\mathcal{E}$ we obtain

$$\mathcal{E}(\tilde{R}_\Omega - R_\Omega) = \mathcal{E}(\tilde{R}_\Omega) - G_{\mathcal{O}} = 0$$

and thus $\tilde{R}_\Omega - R_\Omega \in \ker(\mathcal{E}) \cap \mathcal{V}$. Since $\mathbf{\Lambda}$ is linear, we obtain

$$G' = \mathbf{\Lambda}(\tilde{R}_\Omega) - G = \mathbf{\Lambda}(\tilde{R}_\Omega) - \mathbf{\Lambda}(R_\Omega) = \mathbf{\Lambda}(\tilde{R}_\Omega - R_\Omega) \in \mathbf{\Lambda}(\ker(\mathcal{E}) \cap \mathcal{V}).$$

For the other direction, let $G' \in \mathbf{\Lambda}(\ker(\mathcal{E}) \cap \mathcal{V})$. Then $G' = \mathbf{\Lambda}(R'_\Omega)$ for $R'_\Omega \in \ker(\mathcal{E}) \cap \mathcal{V}$. Define $\tilde{R}_\Omega \coloneqq R_\Omega + R'_\Omega \in \mathcal{V}$. By the linearity of $\mathcal{E}$ and by $R'_\Omega \in \ker(\mathcal{E})$ we obtain

$$\mathcal{E}(\tilde{R}_\Omega) = \mathcal{E}(R_\Omega) + \mathcal{E}(R'_\Omega) = G_{\mathcal{O}} + \mathcal{E}(R'_\Omega) = G_{\mathcal{O}}.$$

Finally, the linearity of $\mathbf{\Lambda}$ shows

$$G' = \mathbf{\Lambda}(R'_\Omega) = \mathbf{\Lambda}(\tilde{R}_\Omega - R_\Omega) = \mathbf{\Lambda}(\tilde{R}_\Omega) - \mathbf{\Lambda}(R_\Omega) = \mathbf{\Lambda}(\tilde{R}_\Omega) - G.$$

This shows $G' \in \mathrm{Amb}^{\mathcal{M}}(G, G_{\mathcal{O}})$. □

See Proposition B.6 for a version of the preceding proposition where the feedback is given by a choice probability function instead of $G_{\mathcal{O}}$. See Appendix C.4 for the ambiguities of many human belief models.

**Remark 2.10.** *In light of the previous proposition, it turns out that the ambiguity does not depend on the true return function and observation return function, and we can thus simply write it as $\mathrm{Amb}^{\mathcal{M}}$.*

**Example 2.11.** *We continue Example 2.6, with the model given by $\mathcal{M} = (\mathcal{S} \times \mathcal{A} \times \mathcal{S}, \mathbf{\Gamma}, \mathbf{B} \circ \mathbf{\Gamma}, \mathbb{R}^{\mathcal{S} \times \mathcal{A} \times \mathcal{S}})$. Proposition 2.9 implies that the ambiguity is given by $\mathrm{Amb}^{\mathcal{M}} = \mathbf{\Gamma}(\ker(\mathbf{B} \circ \mathbf{\Gamma})) = \ker(\mathbf{B}) \cap \mathrm{im}(\mathbf{\Gamma})$. Interested readers can find characterizations of this ambiguity in examples and special cases in Lang et al. (2024). See Sections 2.9 and 3.3 for self-contained examples of ambiguity characterizations in our work.*

It is important to know when the ambiguity disappears, which is captured as *completeness* in the following definition.

**Definition 2.12** (Completeness)**.** *We call the human belief model $\mathcal{M} = (\Omega, \boldsymbol{\Lambda}, \mathcal{E}, \mathcal{V})$ **complete** if $\ker(\mathcal{E}) \cap \mathcal{V} \subseteq \ker(\boldsymbol{\Lambda}) \cap \mathcal{V}$, i.e. if $\mathrm{Amb}^{\mathcal{M}} = \boldsymbol{\Lambda}(\ker(\mathcal{E}) \cap \mathcal{V}) = 0$ (cf. Proposition 2.9).*

We will explain in Interpretation 2.14 why we call this property completeness. We find equivalent and sufficient conditions of completeness in the following theorem:

**Theorem 2.13** (Completeness characterization)**.** *Remember the representations $\lambda : \Xi \to \mathbb{R}^{\Omega}$ and $\epsilon : \mathcal{O} \to \mathbb{R}^{\Omega}$ of the ontology $\boldsymbol{\Lambda} : \mathbb{R}^{\Omega} \to \mathbb{R}^{\Xi}$ and feature belief function $\mathcal{E} : \mathbb{R}^{\Omega} \to \mathbb{R}^{\mathcal{O}}$, respectively. Consider the following four statements:*

*1. $\mathcal{M}$ is complete.*

*2. There exists a linear function $Z : \mathbb{R}^{\mathcal{O}} \to \mathbb{R}^{\Xi}$ with $\boldsymbol{\Lambda}|_{\mathcal{V}} = Z \circ \mathcal{E}|_{\mathcal{V}}$.*

*3. There exists a linear function $Z : \mathbb{R}^{\mathcal{O}} \to \mathbb{R}^{\Xi}$ with $\boldsymbol{\Lambda} = Z \circ \mathcal{E}$.*

*4. For all $\xi \in \Xi$, $\lambda(\xi) \in \mathbb{R}^{\Omega}$ is contained in the span of the image of $\epsilon$: $\lambda(\xi) \in \left\{ \sum_{o \in \mathcal{O}} Z_o \epsilon(o) \mid Z_o \in \mathbb{R} \right\}$.*

*Then 1 and 2 are equivalent, and 3 and 4 are equivalent and imply 1 and 2:*

$$1 \rightleftarrows 2 \Longleftarrow 3 \rightleftarrows 4$$

*Furthermore, the linear function $Z$ from 2 and 3 satisfies $G = Z(G_{\mathcal{O}})$. Finally, if $\mathcal{V} = \mathbb{R}^{\Omega}$, then all four statements are equivalent.*

*Proof.* Note that $\ker(\boldsymbol{\Lambda}) \cap \mathcal{V} = \ker(\boldsymbol{\Lambda}|_{\mathcal{V}})$ and $\ker(\mathcal{E}) \cap \mathcal{V} = \ker(\mathcal{E}|_{\mathcal{V}})$. Thus, the equivalence of statements 1 and 2 immediately follow from Proposition A.3. That 3 implies 2 is obvious.

Now assume statement 4. For all $\xi \in \Xi$ and $o \in \mathcal{O}$, let $Z_{\xi o} \in \mathbb{R}$ be coefficients with

$$\lambda(\xi) = \sum_{o \in \mathcal{O}} Z_{\xi o} \epsilon(o).$$

This implies

$$\boldsymbol{\Lambda}_{\xi \omega} = \sum_{o \in \mathcal{O}} Z_{\xi o} \mathcal{E}_{o \omega}.$$

Define the linear function $Z : \mathbb{R}^{\mathcal{O}} \to \mathbb{R}^{\Xi}$ to have the matrix elements $Z_{\xi o}$. Then the previous equation implies $\boldsymbol{\Lambda} = Z \circ \mathcal{E}$, proving statement 3. That 3 implies 4 follows by reversing these arguments.

Assume statement 2. Then

$$G = \boldsymbol{\Lambda}|_{\mathcal{V}}(R_{\Omega}) = \left( Z \circ \mathcal{E}|_{\mathcal{V}} \right)(R_{\Omega}) = Z\left( \mathcal{E}(R_{\Omega}) \right) = Z(G_{\mathcal{O}}).$$

That all statements are equivalent if $\mathcal{V} = \mathbb{R}^{\Omega}$ is clear. $\square$

Interestingly, for complete models, the preceding proposition shows that $G = Z(G_{\mathcal{O}})$ for a linear function $Z$ that only depends on the human belief model $\mathcal{M}$, thus showing once more that we can infer $G$ from $G_{\mathcal{O}}$ if the model is complete. Since $Z$ may be impractical to find, it may however be more useful to determine $G$ as the unique feedback-compatible return function by first finding $\tilde{R}_{\Omega}$ with $\mathcal{E}(\tilde{R}_{\Omega}) = G_{\mathcal{O}}$.

**Interpretation 2.14.** *Overall, we have thus answered Question 2.7: When the human belief model is known, then $G$ can be inferred from $G_{\mathcal{O}}$ up to the ambiguity $\mathrm{Amb}^{\mathcal{M}} = \boldsymbol{\Lambda}(\ker(\mathcal{E}) \cap \mathcal{V})$, which vanishes if the model $\mathcal{M}$ is complete. Looking back at the definition of feedback-compatible return functions, $G$ is then determined as $G = \boldsymbol{\Lambda}(\tilde{R}_{\Omega})$ for any $\tilde{R}_{\Omega} \in \mathcal{V}$ that gives rise to the human's feedback: $\mathcal{E}(\tilde{R}_{\Omega}) = G_{\mathcal{O}}$. For completeness, we then found further equivalent and sufficient conditions in Theorem 2.13.*

*We now explain why we chose the name* completeness. *By Theorem 2.13, statement 4, a sufficient condition for completeness is the existence of coefficients $Z_{\xi o} \in \mathbb{R}$ such that for all $\xi \in \Xi$, we can write $\lambda(\xi)$ as a linear combination of the $\epsilon(o)$:*

$$\lambda(\xi) = \sum_{o \in \mathcal{O}} Z_{\xi o} \epsilon(o).$$

*This means that the observation-dependent feature beliefs $\epsilon(o)$ span the true feature strengths of trajectories. In this sense, the belief model is "complete": The human's interpretations of observations sufficiently entail true features of trajectories in the MDP. It is thus not a surprise that completeness implies that the ambiguity disappears, and thus that we can infer the return function from the observation return function $G_{\mathcal{O}}$.*

### 2.8 Faithful belief models

Let again $\mathcal{M} = (\Omega, \mathbf{\Lambda}, \mathcal{E}, \mathcal{V})$ be a human belief model for an MDP together with trajectories $\Xi$ and observations $\mathcal{O}$, and let $R_\Omega \in \mathcal{V}$ with $G = \mathbf{\Lambda}(R_\Omega)$, $G_{\mathcal{O}} = \mathcal{E}(R_\Omega)$. Here, we briefly study the notion of faithfulness of human belief models, which is dual to completeness. It captures the idea that any two reward objects that cannot be distinguished by their return functions should also not differ in their observation return functions if the human's feedback reflects a belief over the actual trajectories in the MDP. We now define the notion and then come back to this interpretation in Interpretation 2.17.

**Definition 2.15** (Faithfulness). *We call the human belief model $\mathcal{M}$ **faithful** if $\ker(\mathbf{\Lambda}) \cap \mathcal{V} \subseteq \ker(\mathcal{E}) \cap \mathcal{V}$.*

Many human belief models we study in this paper are faithful. The model from Example 2.11 is faithful since $\ker(\mathbf{\Gamma}) \subseteq \ker(\mathbf{B} \circ \mathbf{\Gamma})$. All models from Section 3.3 are faithful. In Appendix C.2 we study several more faithful human belief models. We can find a characterization of faithfulness that is completely dual to Theorem 2.13:

**Proposition 2.16** (Faithfulness characterization). *Remember the representations $\lambda : \Xi \to \mathbb{R}^\Omega$ and $\epsilon : \mathcal{O} \to \mathbb{R}^\Omega$ of the ontology $\mathbf{\Lambda} : \mathbb{R}^\Omega \to \mathbb{R}^\Xi$ and feature belief function $\mathcal{E} : \mathbb{R}^\Omega \to \mathbb{R}^\mathcal{O}$, respectively. Consider the following four statements:*

1. *$\mathcal{M}$ is faithful.*

2. *There exists a linear function $Y : \mathbb{R}^\Xi \to \mathbb{R}^\mathcal{O}$ with $\mathcal{E}|_\mathcal{V} = Y \circ \mathbf{\Lambda}|_\mathcal{V}$.*

3. *There exists a linear function $Y : \mathbb{R}^\Xi \to \mathbb{R}^\mathcal{O}$ with $\mathcal{E} = Y \circ \mathbf{\Lambda}$.*

4. *For all $o \in \mathcal{O}$, $\epsilon(o) \in \mathbb{R}^\Omega$ is contained in the span of the image of $\lambda$: $\epsilon(o) \in \left\{ \sum_{\xi \in \Xi} Y_\xi \lambda(\xi) \mid Y_\xi \in \mathbb{R} \right\}$.*

*Then 1 and 2 are equivalent, and 3 and 4 are equivalent and imply 1 and 2:*

$$1 \, \underset{\longleftarrow}{\overset{\longrightarrow}{\phantom{xx}}} \, 2 \Longleftarrow 3 \, \underset{\longleftarrow}{\overset{\longrightarrow}{\phantom{xx}}} \, 4$$

*Furthermore, the linear function $Y$ from 2 and 3 satisfies $G_{\mathcal{O}} = Y(G)$. Finally, if $\mathcal{V} = \mathbb{R}^\Omega$, then all four statements are equivalent.*

*Proof.* The proof is analogous to the one for Theorem 2.13. $\qquad\square$

**Interpretation 2.17.** *By Proposition 2.16, a sufficient condition for faithfulness is the existence of coefficient $Y_{o\xi} \in \mathbb{R}$ such that for all $o \in \mathcal{O}$, we can write $\epsilon(o)$ as a linear combination of the $\lambda(\xi)$:*

$$\epsilon(o) = \sum_{\xi \in \Xi} Y_{o\xi} \lambda(\xi).$$

*We can suggestively interpret $Y_{o\xi}$ as the human's "belief" that the true trajectory $\xi$ was responsible for the observation $o$.[1] Under this interpretation, the feature strengths $\epsilon(o)$ are like an "expected value" of true*

---

[1]Though note that $Y$ is usually not unique and $Y_o \in \mathbb{R}^\Xi$ is usually not an actual probability distribution over $\xi$.

*feature strengths, given the human's beliefs. Thus, the feature beliefs encoded by $\epsilon$ are "faithful" to a belief over the actual MDP. This interpretation is especially valid for Example 2.6, where $Y$ is given by $\mathbf{B}$, and thus $Y_{o\xi} = \mathbf{B}_{o\xi} = \big[b(o)\big](\xi)$ is the probability of the trajectory $\xi$, given $o$.*

*Faithfulness is also philosophically reasonable for a related reason. Assume $R_\Omega, \tilde{R}_\Omega \in \mathcal{V}$ are two reward objects with $\mathbf{\Lambda}(R_\Omega) = \mathbf{\Lambda}(\tilde{R}_\Omega)$. Then the resulting return functions coincide in their evaluation of all trajectories. It would then be reasonable to assume that they also coincide in their evaluation of observations, insofar as observations give information about a state of affairs in the actual MDP: $\mathcal{E}(R_\Omega) = \mathcal{E}(\tilde{R}_\Omega)$. This requires that $\mathbf{\Lambda}(R_\Omega - \tilde{R}_\Omega) = 0$ implies $\mathcal{E}(R_\Omega - \tilde{R}_\Omega) = 0$, which exactly means that $\ker(\mathbf{\Lambda}) \cap \mathcal{V} \subseteq \ker(\mathcal{E}) \cap \mathcal{V}$, i.e. the faithfulness of the model. However, since human evaluators do not necessarily have feature beliefs that are rational to this extent, and since this property is not needed in the rest of our theory, we do not assume faithfulness of our human belief models in the general theory.*

## 2.9 Conceptual examples

We now present some simple conceptual examples to illustrate the theory, and especially the ambiguity. We will not define entire MDPs but only those parts of the formalism that are necessary to reason about the ambiguity. In all our examples, we assume $\mathcal{O} \subseteq \Xi$, i.e., observations are entire trajectories: They are fully observed. This brings other factors than observability into the focus, like the human's *understanding* of the trajectories, and whether there is enough data. We refer to Section 3.3 for a concrete examples where the MDP is fully defined and a possibly remaining ambiguity stems from partial observability. Interested readers can also find more such examples in Lang et al. (2024).

In all examples, we have a feature set $\Omega = \{\omega_1, \omega_2, \omega_3, \omega_4\}$ whose meaning will vary in each scenario. We also have observations $o_1, o_2, o_3, o_4, o_5$, though the observation set $\mathcal{O} \subseteq \Xi$ is sometimes only a subset of $\{o_1, o_2, o_3, o_4, o_5\}$. We will make use of the feature belief function $\epsilon : \mathcal{O} \to \mathbb{R}^\Omega \cong \mathbb{R}^4$. We represent each $\epsilon(o)$ as a row-vector with entries $\epsilon(o)_i = \big[\epsilon(o)\big](\omega_i)$ as follows:

$$\begin{aligned}
\epsilon(o_1) &= (1, 0, 0, 1) \\
\epsilon(o_2) &= (0, 1, 0, 1) \\
\epsilon(o_3) &= (0, 0, 1, 1) \\
\epsilon(o_4) &= (0, 3, 0, 2) \\
\epsilon(o_5) &= (0, 0, 0, 1).
\end{aligned} \tag{7}$$

In all examples, we assume $\Xi$, $\mathcal{V}$, and $\mathbf{\Lambda}$ (and thus overall $\mathcal{M} = (\Omega, \mathbf{\Lambda}, \mathcal{E}, \mathcal{V})$) to also be known, though we do not make them explicit.

**Example 2.18** (Simple Completeness)**.** *The goal is to create children stories $\xi \in \Xi$ that appeal to a specific child Alice. We assume that it is known that children care about the presence or absence of exactly the following four features $\Omega = \{\omega_1, \omega_2, \omega_3, \omega_4\}$:*

- *$\omega_1$: Rule-breaking.*

- *$\omega_2$: Non-linear story-telling.*

- *$\omega_3$: Scariness.*

- *$\omega_4$: Moral lessons.*

*We show Alice four stories $\mathcal{O} = \{o_1, o_2, o_3, o_4\} \subseteq \Xi$ that give rise to this feature belief function (cf. Equation (7)):*

$$\mathcal{E} : \mathbb{R}^\Omega \to \mathbb{R}^\mathcal{O}, \quad \mathcal{E} = \begin{pmatrix} 1 & 0 & 0 & 1 \\ 0 & 1 & 0 & 1 \\ 0 & 0 & 1 & 1 \\ 0 & 3 & 0 & 2 \end{pmatrix}. \tag{8}$$

*Conceptually, we assume these are the true feature strengths, meaning that $\lambda(o) = \epsilon(o)$ for all $o \in \mathcal{O}$, i.e., Alice perceives the "true" amount of the four features for each story. The matrix elements mean that $o_1$ contains rule-breaking and moral lessons, but the story-telling is linear and the story is not scary. $o_4$ is very non-linear and contains quite some moral lessons, but shows no rule-breaking or scariness. Since the rows of Equation (8) form a basis of $\mathbb{R}^\Omega$, we obtain*

$$\left\{ \sum_{o \in \mathcal{O}} Z_o \epsilon(o) \mid Z_o \in \mathbb{R} \right\} = \mathbb{R}^\Omega,$$

*and so by Theorem 2.13 it follows that the model is complete. Thus, the ambiguity vanishes and we can infer the return function from Alice's feedback $G_\mathcal{O}$ on the stories in $\mathcal{O}$ (cf. Interpretation 2.14).*

*We demonstrate this now explicitly. Assume $R_\Omega$ is Alice's (a priori unknown) reward object, and that the resulting observation return function $G_\mathcal{O} = \mathcal{E}(R_\Omega) \in \mathbb{R}^\mathcal{O}$ is given by Alice's explicit feedback on the four stories $o_1, \ldots, o_4$ as follows:*

$$G_\mathcal{O}(o_1) = -3, \quad G_\mathcal{O}(o_2) = 1, \quad G_\mathcal{O}(o_3) = 0, \quad G_\mathcal{O}(o_4) = 4. \tag{9}$$

*Representing $R_\Omega$ and $G_\mathcal{O}$ as column-vectors, this means:*

$$\begin{pmatrix} 1 & 0 & 0 & 1 \\ 0 & 1 & 0 & 1 \\ 0 & 0 & 1 & 1 \\ 0 & 3 & 0 & 2 \end{pmatrix} \cdot R_\Omega = \mathcal{E}(R_\Omega) = G_\mathcal{O} = \begin{pmatrix} -3 \\ 1 \\ 0 \\ 4 \end{pmatrix}.$$

*This results in*

$$R_\Omega = \begin{pmatrix} 1 & 0 & 0 & 1 \\ 0 & 1 & 0 & 1 \\ 0 & 0 & 1 & 1 \\ 0 & 3 & 0 & 2 \end{pmatrix}^{-1} \cdot \begin{pmatrix} -3 \\ 1 \\ 0 \\ 4 \end{pmatrix} = \begin{pmatrix} 1 & -3 & 0 & 1 \\ 0 & -2 & 0 & 1 \\ 0 & -3 & 1 & 1 \\ 0 & 3 & 0 & -1 \end{pmatrix} \cdot \begin{pmatrix} -3 \\ 1 \\ 0 \\ 4 \end{pmatrix} = \begin{pmatrix} -2 \\ 2 \\ 1 \\ -1 \end{pmatrix}.$$

*Thus, we recovered Alice's reward object: She positively values non-linear story-telling (2) and scariness (1) but does not like moral lessons $(-1)$ or rule-breaking $(-2)$. Together with the knowledge of the ontology $\mathbf{\Lambda}$ to associate features to stories, we can create more enjoyable stories for Alice.*

*This example shows that knowledge of return-relevant features and a successful modeling of the feature belief function can go a long way to determine the correct return function: We essentially need only four "data points" (in the form of observation returns $G_\mathcal{O}(o)$) to determine $G$.*

**Example 2.19** (Completeness from symmetries). *We assume the same situation as in Example 2.18 except this time, we assume that we only have three observations $\mathcal{O} = \{o_1, o_2, o_4\}$:*

$$\mathcal{E} : \mathbb{R}^\Omega \to \mathbb{R}^\mathcal{O}, \quad \mathcal{E} = \begin{pmatrix} 1 & 0 & 0 & 1 \\ 0 & 1 & 0 & 1 \\ 0 & 3 & 0 & 2 \end{pmatrix}. \tag{10}$$

*Note that the third row now represents $\epsilon(o_4)$ since $o_3$ is not in $\mathcal{O}$. Assume we have* a priori *information that children who like scariness do not like moral lessons and vice versa; i.e., it is known that Alice's reward object satisfies the following equation:*

$$R_\Omega(\omega_3) = -R_\Omega(\omega_4).$$

*The set of all reward functions with this property forms a vector subspace $\mathcal{V} \subseteq \mathbb{R}^\Omega$. It turns out that this implies $\ker(\mathcal{E}) \cap \mathcal{V} = \{0\}$. Indeed, let $R'_\Omega \in \ker(\mathcal{E}) \cap \mathcal{V}$. Then $\mathcal{E}(R'_\Omega) = 0$ implies $R'_\Omega(\omega_1) = R'_\Omega(\omega_2) = R'_\Omega(\omega_4) = 0$, and $R'_\Omega \in \mathcal{V}$ implies $R'_\Omega(\omega_3) = -R'_\Omega(\omega_4) = 0$. With $\ker(\mathcal{E}) \cap \mathcal{V} = 0$, the ambiguity vanishes: $\mathrm{Amb}^\mathcal{M} = \mathbf{\Lambda}(\ker(\mathcal{E}) \cap \mathcal{V}) = 0$. Thus, the model is complete and the return function can be inferred from Alice's feedback $G_\mathcal{O}$ on $\mathcal{O} = \{o_1, o_2, o_4\}$ (cf. Interpretation 2.14).*

**Example 2.20** (Ambiguity from undetected vulnerabilities)**.** *The goal is to correctly evaluate coding-agents to produce valid and safe code blocks $\xi \in \Xi$. We assume that exactly the following four features are relevant for evaluating code:*

- *$\omega_1$: Code-vulnerability.*

- *$\omega_2$: Syntax error.*

- *$\omega_3$: Simplicity.*

- *$\omega_4$: Validity.*

*We show the human evaluator four code-blocks $\mathcal{O} = \{o_2, o_3, o_4, o_5\} \subseteq \Xi$ that give rise to this feature belief function (cf. Equation (7)):*

$$\mathcal{E} \colon \mathbb{R}^\Omega \to \mathbb{R}^{\mathcal{O}}, \quad \mathcal{E} = \begin{pmatrix} 0 & 1 & 0 & 1 \\ 0 & 0 & 1 & 1 \\ 0 & 3 & 0 & 2 \\ 0 & 0 & 0 & 1 \end{pmatrix}.$$

*Crucially, we can assume that $o_5$ does contain a code-vulnerability, but the human does not detect it. In other words, the ontology $\lambda \colon \Xi \to \mathbb{R}^\Omega$ assigns the feature strengths $\lambda(o_5) = (1, 0, 0, 1) \neq (0, 0, 0, 1) = \epsilon(o_5)$. If the human had detected the code-vulnerability, then $o_5$ would be mathematically equivalent to $o_1$ in Example 2.18, the model would be complete, and the ambiguity would disappear.*

*However, this is not the case. Assuming $\mathcal{V} = \mathbb{R}^\Omega$, we obtain that $\ker(\mathcal{E}) \cap \mathcal{V} \neq \{0\}$ contains the reward object $R'_\Omega \in \mathbb{R}^\Omega$ with $R'_\Omega(\omega_1) = 1$ and $R'_\Omega(\omega) = 0$ for all $\omega \neq \omega_1$. With $R_\Omega$ being the unknown true reward object and $\tilde{R}_\Omega := R_\Omega + R'_\Omega$ we then have $\mathcal{E}(\tilde{R}_\Omega) = \mathcal{E}(R_\Omega) = G_{\mathcal{O}}$, and so we cannot distinguish between the reward object $\tilde{R}_\Omega$ and the true reward object $R_\Omega$ from the human's feedback. Let $G = \boldsymbol{\Lambda}(R_\Omega)$ be the true return function. Then the return function $\tilde{G} = \boldsymbol{\Lambda}(\tilde{R}_\Omega) = G + \boldsymbol{\Lambda}(R'_\Omega) \in \mathrm{FC}^{\mathcal{M}}(G)$ is feetback-compatible and satisfies:*

$$\begin{aligned} \tilde{G}(\xi) &= G(\xi) + \big[ \boldsymbol{\Lambda}(R'_\Omega) \big](\xi) \\ &= G(\xi) + \big\langle \lambda(\xi), R'_\Omega \big\rangle \\ &= G(\xi) + \big[ \lambda(\xi) \big](\omega_1). \end{aligned}$$

*This return function positively rewards code-vulnerabilities and can result from an attempt to infer $G$ from $G_{\mathcal{O}}$.*

**Example 2.21** (Rescuing the return inference)**.** *Consider Example 2.20, but with the code block $o_1$ added to the observations, leading to all five observations $\mathcal{O} = \{o_1, \ldots, o_5\}$. This results in the following feature belief function (cf. Equation (7)):*

$$\mathcal{E} \colon \mathbb{R}^\Omega \to \mathbb{R}^{\mathcal{O}}, \quad \mathcal{E} = \begin{pmatrix} 1 & 0 & 0 & 1 \\ 0 & 1 & 0 & 1 \\ 0 & 0 & 1 & 1 \\ 0 & 3 & 0 & 2 \\ 0 & 0 & 0 & 1 \end{pmatrix}.$$

*Then $\ker(\mathcal{E}) = 0$ and thus the ambiguity disappears: $\boldsymbol{\Lambda}(\ker(\mathcal{E}) \cap \mathcal{V}) = 0$, leading to a correct return function inference (cf. Interpretation 2.14). This example highlights that even when the human misinterprets some observations (in our example: $o_5$), the correct return function can sometimes be inferred as long as the human's feature beliefs over all observations have enough coverage.*

## 3  Human belief model covering

So far, we assumed that the true belief model is known precisely, and studied when this allows to recover the return function on trajectories from the human's feedback on observations. Knowing the true belief model

might not be realistic, and so we need to relax this condition. One possibility is to only require that we specify *a* belief model that can *cover* (i.e., represent) all return functions and observation return functions that are represented by the *true* model. If it does, and if it is complete (meaning the ambiguity disappears), then we can intuitively use such a model for a correct return function inference. In Section 3.1, we define this notion of belief model covering. We show that the ambiguity of a covering model is at least as large as the ambiguity of the true model. If the ambiguity disappears, the covering model can be used to infer the correct return function from the human's feedback (Theorem 3.2).

In Section 3.2, we then find an equivalent condition of belief model covering, based on our notion of a *morphism* between belief models. In many examples in this paper we have a natural morphism that accounts for a model covering. In the same theorem, we also find a sufficient condition for the existence of a morphism in terms of a *linear ontology translation* from the covering model to the covered model that also respects feature beliefs (Theorem 3.5). If such an ontology translation exists, then the covering model has the capacity to simultaneously linearly represent the covered model's concepts and beliefs.

In Section 3.3, we study a detailed example of belief model covering: We consider a human with an ontology that is invariant under symmetry transformations in the environment, which implies that we can cover the human's model with a model that assumes symmetry-invariant reward functions. We also demonstrate that this covering model has a vanishing ambiguity, improving upon the ambiguity of a model that considers general reward functions. In Section 3.4, we conclude with a practical proposal for how to find a belief model that covers the true human model, based on using foundation models for the ontology and feature belief function. That the resulting belief model might cover the true human model is motivated by research on sparse autoencoders (Cunningham et al., 2023; Bricken et al., 2023), which we will interpret as linear ontology translations.

In this whole section, we assume an MDP together with trajectories $\Xi$ and observations $\mathcal{O}$ as given.

## 3.1 Human belief model covering and its implications

**Definition 3.1** (Belief model covering)**.** *Let* $\mathcal{M} = (\Omega, \mathbf{\Lambda}, \mathcal{E}, \mathcal{V})$ *and* $\widehat{\mathcal{M}} = (\widehat{\Omega}, \widehat{\mathbf{\Lambda}}, \widehat{\mathcal{E}}, \widehat{\mathcal{V}})$ *be two human belief models. Then we say that* $\widehat{\mathcal{M}}$ ***covers*** $\mathcal{M}$ *if for all* $v \in \mathcal{V}$ *there exists* $\widehat{v} \in \widehat{\mathcal{V}}$ *with* $\widehat{\mathbf{\Lambda}}(\widehat{v}) = \mathbf{\Lambda}(v)$ *and* $\widehat{\mathcal{E}}(\widehat{v}) = \mathcal{E}(v)$.

This means that $\widehat{\mathcal{M}}$ can represent all return functions $\mathbf{\Lambda}(v)$ and observation return functions $\mathcal{E}(v)$ that are represented by $\mathcal{M}$. We can visualize this as follows:

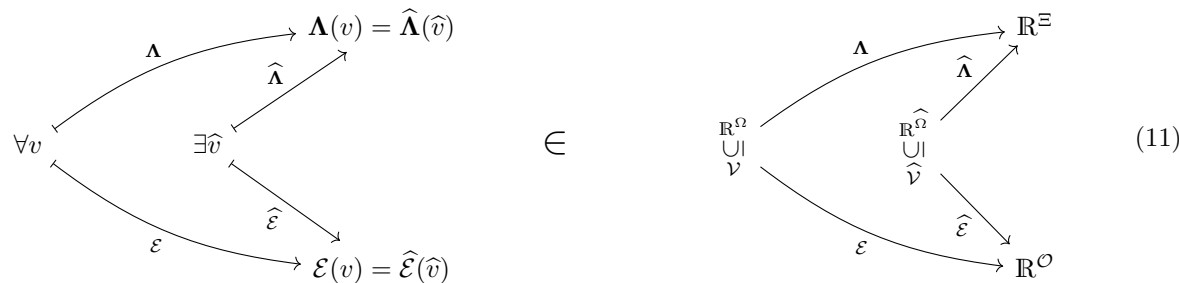

$$(11)$$

**Theorem 3.2.** *Let* $\mathcal{M} = (\Omega, \mathbf{\Lambda}, \mathcal{E}, \mathcal{V})$ *and* $\widehat{\mathcal{M}} = (\widehat{\Omega}, \widehat{\mathbf{\Lambda}}, \widehat{\mathcal{E}}, \widehat{\mathcal{V}})$ *be two human belief models, and assume that* $\widehat{\mathcal{M}}$ *covers* $\mathcal{M}$. *We think of* $\mathcal{M}$ *as the "true" human belief model representing* $G = \mathbf{\Lambda}(R_\Omega)$ *and* $G_\mathcal{O} = \mathcal{E}(R_\Omega)$ *with a reward object* $R_\Omega \in \mathcal{V}$. *Then we have:*

1. *$\mathrm{Amb}^{\mathcal{M}} \subseteq \mathrm{Amb}^{\widehat{\mathcal{M}}}$.*

2. *If $\mathcal{M}$ also covers $\widehat{\mathcal{M}}$, then $\mathrm{Amb}^{\mathcal{M}} = \mathrm{Amb}^{\widehat{\mathcal{M}}}$.*

3. *There is an $R_{\widehat{\Omega}} \in \widehat{\mathcal{V}}$ with $\widehat{\mathcal{E}}(R_{\widehat{\Omega}}) = G_\mathcal{O}$ and $\widehat{\mathbf{\Lambda}}(R_{\widehat{\Omega}}) = G$, and so $\widehat{\mathcal{M}}$ also represents $G$ and $G_\mathcal{O}$.*

4. *Assume $\widehat{\mathcal{M}}$ is complete. Then* every *reward object $\tilde{R}_{\widehat{\Omega}} \in \widehat{\mathcal{V}}$ with $\widehat{\mathcal{E}}(\tilde{R}_{\widehat{\Omega}}) = G_{\mathcal{O}}$ also satisfies $\widehat{\mathbf{\Lambda}}(\tilde{R}_{\widehat{\Omega}}) = G$. In other words, the set of feedback compatible return functions is given by $\mathrm{FC}^{\widehat{\mathcal{M}}}(G_{\mathcal{O}}) = \{G\}$.*

*Proof.* By Proposition 2.9, we have $\mathrm{Amb}^{\mathcal{M}} = \mathbf{\Lambda}(\ker(\mathcal{E}) \cap \mathcal{V})$ and $\mathrm{Amb}^{\widehat{\mathcal{M}}} = \widehat{\mathbf{\Lambda}}\big(\ker(\widehat{\mathcal{E}}) \cap \widehat{\mathcal{V}}\big)$. To prove claim 1, let $G' = \mathbf{\Lambda}(R'_{\Omega}) \in \mathrm{Amb}^{\mathcal{M}}$, where $R'_{\Omega} \in \ker(\mathcal{E}) \cap \mathcal{V}$. Then by the definition of $\widehat{M}$ covering $\mathcal{M}$, there exists an $R'_{\widehat{\Omega}} \in \widehat{\mathcal{V}}$ with $\widehat{\mathbf{\Lambda}}(R'_{\widehat{\Omega}}) = \mathbf{\Lambda}(R'_{\Omega}) = G'$ and $\widehat{\mathcal{E}}(R'_{\widehat{\Omega}}) = \mathcal{E}(R'_{\Omega}) = 0$. The latter implies $R'_{\widehat{\Omega}} \in \ker(\widehat{\mathcal{E}}) \cap \widehat{\mathcal{V}}$, and so $G' = \widehat{\mathbf{\Lambda}}(R'_{\widehat{\Omega}}) \in \widehat{\mathbf{\Lambda}}(\ker(\widehat{\mathcal{E}}) \cap \widehat{\mathcal{V}}) = \mathrm{Amb}^{\widehat{\mathcal{M}}}$. This proves claim 1.

Claim 2 then immediately follows from claim 1. Claim 3 is also immediate by the definition of $\widehat{\mathcal{M}}$ covering $\mathcal{M}$ and using that $G_{\mathcal{O}} = \mathcal{E}(R_{\Omega})$ and $G = \mathbf{\Lambda}(R_{\Omega})$.

Now we prove claim 4. Thus, assume $\widehat{\mathcal{M}}$ is complete and let $\tilde{R}_{\widehat{\Omega}} \in \widehat{\mathcal{V}}$ be a reward object with $\widehat{\mathcal{E}}(\tilde{R}_{\widehat{\Omega}}) = G_{\mathcal{O}}$. By claim 3, there exists an $R_{\widehat{\Omega}} \in \widehat{\mathcal{V}}$ with $\widehat{\mathcal{E}}(R_{\widehat{\Omega}}) = G_{\mathcal{O}}$ and $\widehat{\mathbf{\Lambda}}(R_{\widehat{\Omega}}) = G$. It follows $\tilde{R}_{\widehat{\Omega}} - R_{\widehat{\Omega}} \in \ker(\widehat{\mathcal{E}}) \cap \widehat{\mathcal{V}} \subseteq \ker(\widehat{\mathbf{\Lambda}}) \cap \widehat{\mathcal{V}}$, where we use that $\widehat{\mathcal{M}}$ is complete in the last step. Consequently, we have

$$\widehat{\mathbf{\Lambda}}(\tilde{R}_{\widehat{\Omega}}) = \widehat{\mathbf{\Lambda}}(R_{\widehat{\Omega}}) = G,$$

thus proving the claim. $\qquad\square$

In Theorem B.8 we present a version of the preceding theorem for the case that feedback is given by a choice probability function instead of $G_{\mathcal{O}}$. In Appendix C.4, we see several applications of the first two statements of the theorem to determine inclusions and equalities of ambiguities.

We can interpret Theorem 3.2 as follows: Given a "true" model $\mathcal{M}$, but using a covering model $\widehat{\mathcal{M}}$, we lose something since the ambiguity of $\widehat{\mathcal{M}}$ is possibly larger. However, $\widehat{\mathcal{M}}$ is able to represent the true return function and observation return function, and *if* the ambiguity of $\widehat{\mathcal{M}}$ disappears, then the set of return functions compatible with the human's feedback that can be inferred using $\widehat{\mathcal{M}}$ is exactly $\{G\}$. In other words, we can then infer $G$ from the human's feedback and $\widehat{\mathcal{M}}$ *without knowing the true human belief model* $\mathcal{M}$, thus relaxing the assumptions baked into Question 2.7. In Section 3.4 we discuss hypotheses for finding a covering belief model $\widehat{\mathcal{M}}$ in practice.

## 3.2 Morphisms of human belief models and ontology translations

One drawback of the definition of a belief model covering is that it is hard to directly test: One would need to iterate over all $v \in \mathcal{V}$ to check whether there exists a $\widehat{v} \in \widehat{\mathcal{V}}$ that gives rise to the same return function and observation return function as $v$. This analysis could be simplified if we could find one function $\Phi : \mathbb{R}^{\Omega} \to \mathbb{R}^{\widehat{\Omega}}$ that maps each $v \in \mathcal{V}$ directly to $\widehat{v} \in \widehat{\mathcal{V}}$. The covering property would then translates to properties of such functions. We state these properties in the following definition:

**Definition 3.3** (Morphism of human belief models)**.** *Let $\mathcal{M} = (\Omega, \mathbf{\Lambda}, \mathcal{E}, \mathcal{V})$ and $\widehat{\mathcal{M}} = (\widehat{\Omega}, \widehat{\mathbf{\Lambda}}, \widehat{\mathcal{E}}, \widehat{\mathcal{V}})$ be human belief models. Then a linear function $\Phi : \mathbb{R}^{\Omega} \to \mathbb{R}^{\widehat{\Omega}}$ is called a **morphism of human belief models** if the following holds:*

1. $\Phi(\mathcal{V}) \subseteq \widehat{\mathcal{V}}$.

2. $\mathbf{\Lambda}|_{\mathcal{V}} = \widehat{\mathbf{\Lambda}} \circ \Phi|_{\mathcal{V}}$.

3. $\mathcal{E}|_{\mathcal{V}} = \widehat{\mathcal{E}} \circ \Phi|_{\mathcal{V}}$.

*We write the morphism also as $\Phi : \mathcal{M} \to \widehat{\mathcal{M}}$.*

The following visualization, an adaptation of Equation (11), makes intuitive that the existence of a morphism is equivalent to belief model covering; we will prove this in Theorem 3.5:

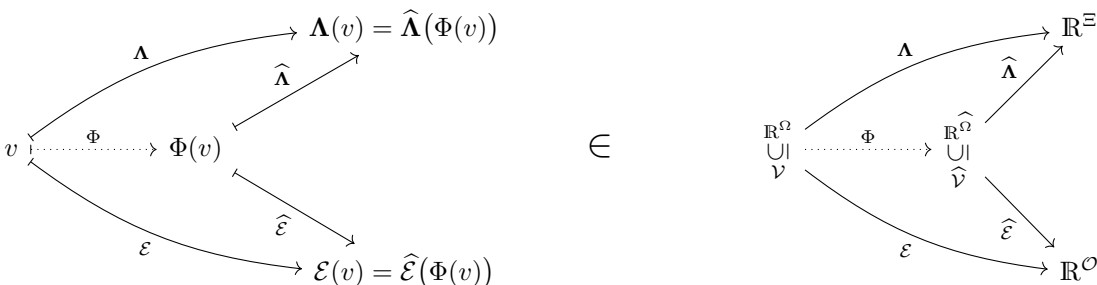

We now work toward a sufficient condition for belief model morphisms that we call *linear ontology translations*. For this relationship, recall the form $\lambda : \Xi \to \mathbb{R}^\Omega$, $\epsilon : \mathcal{O} \to \mathbb{R}^\Omega$ of the ontology and feature belief function of a belief model $\mathcal{M} = (\Omega, \mathbf{\Lambda}, \mathcal{E}, \mathcal{V})$, respectively. Recall from Equation (5) that for a reward object $v \in \mathcal{V}$ and trajectory $\xi \in \Xi$, the corresponding return is given by $\big[\mathbf{\Lambda}(v)\big](\xi) = \big\langle \lambda(\xi), v \big\rangle$. Under this viewpoint, property 2 of belief model morphisms $\Phi$ implies that for all $v \in \mathcal{V}$ and $\xi \in \Xi$, we have

$$\big\langle \widehat{\lambda}(\xi), \Phi(v) \big\rangle = \big\langle \lambda(\xi), v \big\rangle,$$

and similar for $\epsilon$ and $\widehat{\epsilon}$. By a defining property of transpose matrices (also called adjoints), we obtain for all $v \in \mathcal{V}$ and $\xi \in \Xi$:

$$\big\langle \Phi^T \widehat{\lambda}(\xi), v \big\rangle = \big\langle \lambda(\xi), v \big\rangle,$$

and thus $\Phi^T \circ \widehat{\lambda} = \lambda$. This equation means that we can interpret $\Phi^T$ as a *translation* from the ontology $\widehat{\lambda}$ to the ontology $\lambda$. We define:

**Definition 3.4** (Linear ontology translation). *Let $\lambda : \Xi \to \mathbb{R}^\Omega$, $\epsilon : \mathcal{O} \to \mathbb{R}^\Omega$ and $\widehat{\lambda} : \Xi \to \mathbb{R}^{\widehat{\Omega}}$, $\widehat{\epsilon} : \mathcal{O} \to \mathbb{R}^{\widehat{\Omega}}$ be two pairs of an ontology and a feature belief function. A linear function $\Psi : \mathbb{R}^{\widehat{\Omega}} \to \mathbb{R}^\Omega$ with $\Psi \circ \widehat{\lambda} = \lambda$ is called a **linear ontology translation** from $\widehat{\lambda}$ to $\lambda$. Furthermore, we call it **belief-compatible** with $\widehat{\epsilon}$ and $\epsilon$ if we also have $\Psi \circ \widehat{\epsilon} = \epsilon$.*

These notions are connected in the following Theorem:

**Theorem 3.5.** *Let $\mathcal{M} = (\Omega, \mathbf{\Lambda}, \mathcal{E}, \mathcal{V})$ and $\widehat{\mathcal{M}} = (\widehat{\Omega}, \widehat{\mathbf{\Lambda}}, \widehat{\mathcal{E}}, \widehat{\mathcal{V}})$ be two human belief models. Let $\lambda, \epsilon, \widehat{\lambda}, \widehat{\epsilon}$ be the alternative representations of the ontologies and feature belief functions. Consider the following statements:*

1. *$\widehat{\mathcal{M}}$ covers $\mathcal{M}$.*

2. *There exists a morphism of belief models $\Phi : \mathcal{M} \to \widehat{\mathcal{M}}$.*

3. *There exists a function $\Phi : \mathbb{R}^\Omega \to \mathbb{R}^{\widehat{\Omega}}$ with $\widehat{\mathbf{\Lambda}} \circ \Phi = \mathbf{\Lambda}$ and $\widehat{\mathcal{E}} \circ \Phi = \mathcal{E}$.*

4. *There is a function $\Psi : \mathbb{R}^{\widehat{\Omega}} \to \mathbb{R}^\Omega$ that is a linear ontology translation from $\widehat{\lambda}$ to $\lambda$ that is also belief-compatible with $\widehat{\epsilon}$ and $\epsilon$.*

*Then 1 and 2 are equivalent, and 3 and 4 are equivalent and both imply 1 and 2:*

$$1 \rightleftarrows 2 \Longleftarrow 3 \rightleftarrows 4$$

*$\Psi$ from 4 can be defined as $\Psi = \Phi^T$ for $\Phi$ from 3. If $\mathcal{V} = \mathbb{R}^\Omega$, then all four statements are equivalent. Finally, if 2 holds and $\Phi(\mathcal{V}) = \widehat{\mathcal{V}}$, then $\mathcal{M}$ also covers $\widehat{\mathcal{M}}$ and $\mathrm{Amb}^{\mathcal{M}} = \mathrm{Amb}^{\widehat{\mathcal{M}}}$.*

*Proof.* The implication from the second to the first statement follows immediately from setting $\widehat{v} := \Phi(v)$ in the definition of belief model covering. For the other direction, consider the following diagram:

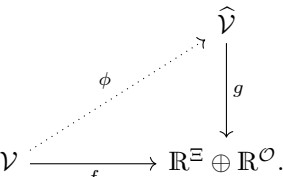

In this diagram, we define $f := (\boldsymbol{\Lambda}|_{\mathcal{V}}, \mathcal{E}|_{\mathcal{V}})$ and $g := (\widehat{\boldsymbol{\Lambda}}|_{\widehat{\mathcal{V}}}, \widehat{\mathcal{E}}|_{\widehat{\mathcal{V}}})$. Then statement 1 is equivalent to $\mathrm{im}(f) \subseteq \mathrm{im}(g)$. By Proposition A.4, there is thus a linear function $\phi : \mathcal{V} \to \widehat{\mathcal{V}}$ with $g \circ \phi = f$:

$$
\begin{array}{ccc}
 & & \widehat{\mathcal{V}} \\
 & \nearrow^{\phi} & \downarrow g \\
\mathcal{V} & \xrightarrow{f} & \mathbb{R}^{\Xi} \oplus \mathbb{R}^{\mathcal{O}}.
\end{array}
$$

Extend $\phi$ arbitrarily to a linear function $\Phi : \mathbb{R}^{\Omega} \to \mathbb{R}^{\widehat{\Omega}}$ with $\Phi|_{\mathcal{V}} = \phi$, e.g., by extending a basis on $\mathcal{V}$ to a basis on all of $\mathbb{R}^{\Omega}$. Then for all $v \in \mathcal{V}$, the diagram shows that we have

$$
\big(\boldsymbol{\Lambda}(v), \mathcal{E}(v)\big) = \big(\widehat{\boldsymbol{\Lambda}}(\Phi(v)), \widehat{\mathcal{E}}(\Phi(v))\big).
$$

Consequently, $\boldsymbol{\Lambda}|_{\mathcal{V}} = \widehat{\boldsymbol{\Lambda}} \circ \Phi|_{\mathcal{V}}$ and $\mathcal{E}|_{\mathcal{V}} = \widehat{\mathcal{E}} \circ \Phi|_{\mathcal{V}}$. Thus, $\Phi$ is a morphism of belief models.

That statements 3 and 4 are equialent and $\Psi$ can be chosen as $\Phi^T$ follows from Proposition A.2. That 3 implies 2 is clear. That all statements are equivalent if $\mathcal{V} = \mathbb{R}^{\Omega}$ is also clear.

For the final statement, assume that 2 holds and that $\Phi(\mathcal{V}) = \widehat{\mathcal{V}}$. Then for all $\widehat{v} \in \widehat{\mathcal{V}}$ there is $v \in \mathcal{V}$ with $\Phi(v) = \widehat{v}$. Since $\Phi$ is a morphism, this results in

$$
\boldsymbol{\Lambda}(v) = \widehat{\boldsymbol{\Lambda}}\big(\Phi(v)\big) = \widehat{\boldsymbol{\Lambda}}(\widehat{v}),
$$
$$
\mathcal{E}(v) = \widehat{\mathcal{E}}\big(\Phi(v)\big) = \widehat{\mathcal{E}}(\widehat{v}),
$$

showing that $\mathcal{M}$ covers $\widehat{\mathcal{M}}$. $\mathrm{Amb}^{\mathcal{M}} = \mathrm{Amb}^{\widehat{\mathcal{M}}}$ then follows from statement 2 in Theorem 3.2. $\qquad\square$

In Appendix C, we show that human belief models, together with their morphisms, form a *category* (Lane, 1998). In Appendix C.4 we then construct a large diagram of human belief models together with their natural morphisms, which then also allows for an analysis of their ambiguities.

The preceding theorem shows that a sufficient condition for $\Phi : \mathcal{M} \to \widehat{\mathcal{M}}$ to be a human belief model morphism is for $\Phi^T : \mathbb{R}^{\widehat{\Omega}} \to \mathbb{R}^{\Omega}$ to be a belief-compatible ontology translation: $\Phi^T \circ \widehat{\lambda} = \lambda$ and $\Phi^T \circ \widehat{\epsilon} = \epsilon$. Intuitively, this means that our model $\widehat{\mathcal{M}}$ is "expressive" enough to allow for the true model's concepts and beliefs to be linearly represented. We will draw more connections to linearly represented beliefs and concepts in Section 3.4.

## 3.3 An example of symmetry-invariant features and reward functions

We now study an MDP with natural symmetries in the environment. We can then reasonably assume the human's ontology to be *invariant* under these symmetries. We explain how one can cover the resulting human belief model with a model that distinguishes between symmetry-related states, but compensates for it by assuming that the *valid reward functions* are symmetry-invariant (van der Pol et al., 2021). The ambiguity in this covering model will disappear, while the ambiguity of a third model that allows for generic reward

functions does not. This demonstrates the usefulness of both covering belief models and a careful choice of the vector space of valid reward objects. In particular, the analysis shows that encoding a priori knowledge of symmetries into the human belief model can be fruitful for inferring the correct return function from the human's feedback.

We proceed by first defining the MDP, set of trajectories, and observations. Afterward, we define all three belief models together with their morphisms, demonstrating model coverage. Finally, we conclude with the ambiguity analysis.

### 3.3.1 Specification of the MDP, $\Xi$, and $\mathcal{O}$

Our MDP is a 2x2 gridworld with a movable hand $H$ and a fixed button $B$, inspired by the robot-hand example from Amodei et al. (2017). States look like this:

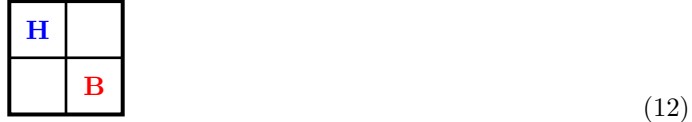

$$(12)$$

In total, the set of states $\mathcal{S}$ has sixteen elements, one for each combination of the position of $H$ and $B$. The set of action is given by $\mathcal{A} = \{L, R, U, D, P\}$, where the first four actions move the hand: $L$ to the left, $R$ to the right, $U$ upward, and $D$ downward. $P$ does not change the state, and is conceptually meant to "press" the button if the hand and button are in the same position. If a movement goes toward an adjacent wall in the gridworld, then the state also does not change. This specifies the transition function $\mathcal{T} : \mathcal{S} \times \mathcal{A} \to \mathcal{S}$. $P_0$, the initial state distribution, is a uniform distribution over the following four states:

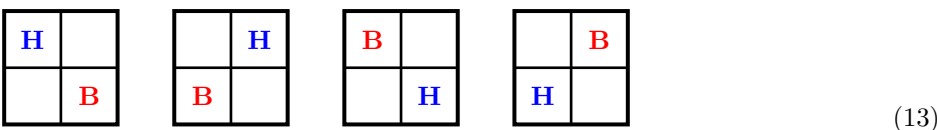

$$(13)$$

The time horizon is $T = 3$. The *unknown* true return function is given by

$$G(\xi) = \sum_{t=0}^{2} R(s_t, a_t),$$

where $R(s_t, a_t) = 1$ if the hand $H$ and button $B$ are in the same position in $s_t$ and if $a_t = P$, and $R(s_t, a_t) = 0$ otherwise. In other words, the return function rewards pressing the button. This completely specifies the MDP $(\mathcal{S}, \mathcal{A}, \mathcal{T}, P_0, T, G)$.

The Trajectories $\Xi$ are given by all sequences of four states and three actions that start with a state sampled from $P_0$, and where each transition is compatible with the description above. The observations $\mathcal{O}$ of the human evaluator are given by views "from below". We assume the human does not observe movement actions (but may be able to infer them if the hand visibly changes position), but *does* observe whether the button was pressed. Formally, $\mathcal{O} = O(\Xi)$ for a surjective function $O : \Xi \to \mathcal{O}$ that projects the view and removes movement information, as we suggestively depict for an example trajectory $\xi$ and its observation $O(\xi)$ in this figure:

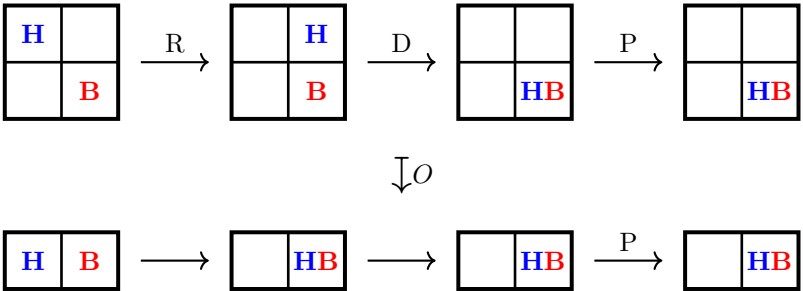

See Appendix D.1 for some mathematical details.

### 3.3.2 Three human belief models and their morphisms

Crucially, we assume that the human evaluator does not use state-action pairs as features in the ontology, but instead *representatives of symmetry-equivalence classes* of state-action pairs. This is reasonable since we can a priori assume that the human evaluator does not care about the orientation of the scene. In other words, we consider the symmetry group $G = D_4$ of the square, which identifies two state-action pairs if they are related by a rotation of $0°, 90°, 180°$, or $270°$, or a reflection along the vertical, horizontal, or one of the diagonal axes. This leads to just three representative states

$$s_0 = \boxed{\phantom{x}} , \qquad s_1 = \boxed{\phantom{x}} , \qquad s_2 = \boxed{\phantom{x}} . \tag{14}$$

Overall, the set of representative state-action pairs is given by $\overline{\mathcal{S} \times \mathcal{A}} = \bigcup_{i=0}^{2}\{s_i\} \times \mathcal{A}^i$ with $\mathcal{A}^0 = \mathcal{A}^2 = \{L, D, P\}$ and $\mathcal{A}^1 = \{L, R, U, D, P\}$.[2] Let $h : \mathcal{S} \times \mathcal{A} \to \overline{\mathcal{S} \times \mathcal{A}}$ map each state-action pair to the unique representative. Details on everything discussed so far can be found in Appendix D.2.

We define the human's ontology $\lambda : \Xi \to \mathbb{R}^{\overline{\mathcal{S} \times \mathcal{A}}}$ via

$$\big[\lambda(\xi)\big](s, a) := \sum_{t=0}^{2} \delta_{(s,a)}(h(s_t, a_t)),$$

which is the number of times that, up to symmetry, the state-action pair $(s, a)$ appears in the trajectory $\xi$. Interestingly, this ontology is *invariant* under transforming trajectories $\xi$ via symmetries since $h$ is invariant under transforming state-action pairs. Set $\mathbf{\Lambda} : \mathbb{R}^{\overline{\mathcal{S} \times \mathcal{A}}} \to \mathbb{R}^{\Xi}$ as the linear function correponding to $\lambda$ via Proposition A.1, and let $\mathbf{\Gamma} : \mathbb{R}^{\mathcal{S} \times \mathcal{A}} \to \mathbb{R}^{\Xi}$ be the function from Example 2.1 without discounting ($\gamma = 1$).[3] Finally, let $h^* : \mathbb{R}^{\overline{\mathcal{S} \times \mathcal{A}}} \to \mathbb{R}^{\mathcal{S} \times \mathcal{A}}$ be the function $h^*(\overline{R}) := \overline{R} \circ h$ (see also the discussion surrounding Equation (1)). Then in Equation (24) we show

$$\mathbf{\Lambda} = \mathbf{\Gamma} \circ h^*. \tag{15}$$

Let $b : \mathcal{O} \to \Delta(\Xi) \subseteq \mathbb{R}^{\Xi}$ be the human's trajectory belief function, where we define $b(o)$ as the uniform distribution over all $\xi \in \Xi$ that get observed as $o$: $O(\xi) = o$ (cf. Equation (25)). We then define the human's feature belief function $\epsilon : \mathcal{O} \to \mathbb{R}^{\overline{\mathcal{S} \times \mathcal{A}}}$ by

$$\big[\epsilon(o)\big](s, a) := \sum_{\xi \in \Xi} \big[b(o)\big](\xi) \cdot \big[\lambda(\xi)\big](s, a).$$

---

[2]$s_0$ and $s_2$ have fewer representative actions since $L$ and $U$, and also $R$ and $D$, are related by the reflection along the diagonal axis from top left to bottom right. This transformation leaves the state invariant and maps between the actions.

[3]Note the small difference that now we consider reward functions that only depend on state-action pairs instead of state-action-state transitions.

This is the *expected* number of times that, up to symmetry, the state-action pair $(s, a)$ occurs in a trajectory that led to observation $o$. Set $\mathcal{E}: \mathbb{R}^{\overline{\mathcal{S} \times \mathcal{A}}} \to \mathbb{R}^{\mathcal{O}}$ and $\mathbf{B}: \mathbb{R}^{\Xi} \to \mathbb{R}^{\mathcal{O}}$ as the linear functions correponding to $\epsilon$ and $b$ via Proposition A.1. Then as a consequence of Equation (15), we obtain

$$\mathcal{E} = \mathbf{B} \circ \mathbf{\Gamma} \circ h^*, \tag{16}$$

as we show in Equation (26).

Finally, we assume that we have *a priori* knowledge that the human's reward object lies in the subvectorspace $\mathcal{V} \subseteq \mathbb{R}^{\overline{\mathcal{S} \times \mathcal{A}}}$ given by reward objects $\overline{R}$ with $\overline{R}(s, a) = 0$ whenever $a \neq P$. In other words, we know that only the pressing-action can be rewarded, but we do *not* know a priori that it is only rewarded when the hand is over the button. Furthermore, define $\mathcal{V}' \subseteq \mathbb{R}^{\mathcal{S} \times \mathcal{A}}$ likewise as reward functions with $R(s, a) = 0$ whenever $a \neq P$. Consider the commutative diagram

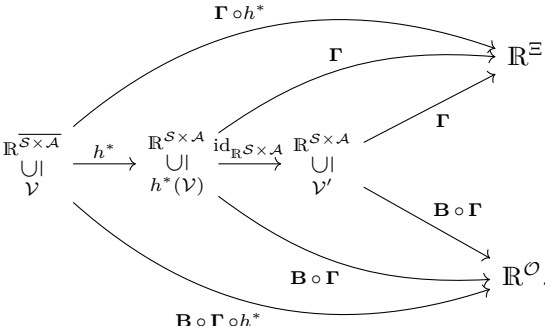

This establishes three human belief models

$$\begin{aligned} \mathcal{M}_1 &= (\overline{\mathcal{S} \times \mathcal{A}}, \ \mathbf{\Gamma} \circ h^*, \ \mathbf{B} \circ \mathbf{\Gamma} \circ h^*, \ \mathcal{V}), \\ \mathcal{M}_2 &= (\mathcal{S} \times \mathcal{A}, \ \mathbf{\Gamma}, \ \mathbf{B} \circ \mathbf{\Gamma}, \ h^*(\mathcal{V})), \\ \mathcal{M}_3 &= (\mathcal{S} \times \mathcal{A}, \ \mathbf{\Gamma}, \ \mathbf{B} \circ \mathbf{\Gamma}, \ \mathcal{V}'), \end{aligned}$$

together with the morphisms

$$\mathcal{M}_1 \xrightarrow{h^*} \mathcal{M}_2 \xrightarrow{\mathrm{id}_{\mathbb{R}^{\mathcal{S} \times \mathcal{A}}}} \mathcal{M}_3.$$

Our aim will be to show that the ambiguities of $\mathcal{M}_1$ and $\mathcal{M}_2$ will vanish since these models leverage a priori knowledge of symmetries, while the ambiguity of $\mathcal{M}_3$ will not vanish. Intuitively, $\mathcal{M}_1$ is the true belief model of a human evaluator with symmetry-invariant features. $\mathcal{M}_2$ is the model that we "specify" and with which we "cover" the true belief model. That it indeed covers $\mathcal{M}_1$ follows from Theorem 3.5 and the existence of the morphism $h^*: \mathcal{M}_1 \to \mathcal{M}_2$.

Write $g.(s, a)$ for the action of a symmetry-transformation $g \in G = D_4$ of the square on a state-action pair $(s, a)$ (cf Appendix D.2). Then the set of valid reward objects of $\mathcal{M}_2$ is given by

$$h^*(\mathcal{V}) = \left\{ R \in \mathbb{R}^{\mathcal{S} \times \mathcal{A}} \ \middle| \ \forall g \in G \colon R(g.(s, a)) = R(s, a) \text{ and } \forall a \neq P \colon R(s, a) = 0 \right\}. \tag{17}$$

In other words, it is the set of *symmetry-invariant* reward functions that don't reward actions unequal to $P$. Such reward functions play a role in symmetry-invariant reinforcement learning (van der Pol et al., 2021). Finally, $\mathcal{M}_3$ is the same model, but with a larger set of valid reward functions that are not necessarily symmetry-invariant.

Mathematically, all three models are faithful by Proposition 2.16. As an aside, they are also balanced (Definition B.2) by Lemma B.3, which essentially means that the ontology and feature belief functions are row-constant. The three models and their morphisms are also closely related to the three models $\mathcal{M}_{\mathcal{F}}^{\mathcal{F}}$, $\mathcal{M}_{\mathcal{F}}^{\mathcal{S} \times \mathcal{A} \times \mathcal{S}}$, and $\mathcal{M}_{\mathcal{S} \times \mathcal{A} \times \mathcal{S}}^{\mathcal{S} \times \mathcal{A} \times \mathcal{S}}$ that we study in Appendix C.4.

### 3.3.3 The ambiguity analysis

We now analyze the ambiguities of the three models $\mathcal{M}_1, \mathcal{M}_2$, and $\mathcal{M}_3$. Since the morphism $h^*$ maps the set of valid reward objects $\mathcal{V}$ of $\mathcal{M}_1$ precisely to the valid reward objects $h^*(\mathcal{V})$ of $\mathcal{M}_2$, by Theorem 3.5, $\mathcal{M}_1$ and $\mathcal{M}_2$ have the same ambiguity. Thus, let us analyze the ambiguities of $\mathcal{M}_1$ and $\mathcal{M}_3$.

For $\mathcal{M}_1$, Proposition 2.9 shows that the ambiguity is given by $(\mathbf{\Gamma} \circ h^*)\big[\ker(\mathbf{B} \circ \mathbf{\Gamma} \circ h^*) \cap \mathcal{V}\big]$. For showing that it vanishes, we simply show $\ker(\mathbf{B} \circ \mathbf{\Gamma} \circ h^*) \cap \mathcal{V} = \ker(\mathcal{E}) \cap \mathcal{V} = 0$. Thus, let $\overline{R} \in \mathbb{R}^{\mathcal{S} \times \mathcal{A}}$ be a reward object in $\mathcal{V}$ with $\mathcal{E}(\overline{R}) = 0$. We need to show $\overline{R} = 0$. Recall the representative states $s_0, s_1, s_2$ from Equation (14). Since $\overline{R} \in \mathcal{V}$, we have $\overline{R}(s, a) = 0$ for all $s \in \{s_0, s_1, s_2\}$ and all $a \neq P$, and so we simply need to show $\overline{R}(s_i, P) = 0$ for all $i = 0, 1, 2$. Consider the following three observations:

Then it is easy to show that $\big[\mathcal{E}(\overline{R})\big](o_2) = 0$ implies $\overline{R}(s_2, P) = 0$ since $(s_2, P)$ is, up to symmetry, the only state-action pair that is compatible with the observation $o_2$. Then, $\big[\mathcal{E}(\overline{R})\big](o_1) = 0$ implies $\overline{R}(s_1, P) = 0$ since $(s_1, P)$ is the only state-action pair *other than* $(s_2, P)$ that is compatible with the observation $o_1$ and could a priori have a non-zero contribution to the reward. Finally, $\big[\mathcal{E}(\overline{R})\big](o_0) = 0$ implies $\overline{R}(s_0, P) = 0$ for similar reasons. We present details of these arguments in Appendix D.3. Overall, we have thus showed that $\overline{R} = 0$, and thus $\ker(\mathcal{E}) \cap \mathcal{V} = 0$, which implies $\mathrm{Amb}^{\mathcal{M}_1} = \mathbf{\Lambda}(\ker(\mathcal{E}) \cap \mathcal{V}) = 0$ (cf. Proposition 2.9). Since $\mathcal{M}_1$ and $\mathcal{M}_2$ have the same ambiguities, also our covering model $\mathcal{M}_2$ has vanishing ambiguity: $\mathrm{Amb}^{\mathcal{M}_2} = 0$.

Now we show that the ambiguity of $\mathcal{M}_3$ does *not* vanish. Consider the following two states, which are symmetry-transformed versions of state $s_1$ from Equation (14):

Then, let $R' : \mathcal{S} \times \mathcal{A} \to \mathbb{R}$ be the reward function with

$$R'(s, a) = \begin{cases} 1, & s = s_1' \text{ and } a = P \\ -1, & s = s_1'' \text{ and } a = P \\ 0, \text{ else.} \end{cases}$$

Clearly, we have $R' \in \mathcal{V}'$. Then note that for all observations $o$, we have $(\mathbf{B} \circ \mathbf{\Gamma})_{o,(s_1', P)} = (\mathbf{B} \circ \mathbf{\Gamma})_{o,(s_1'', P)}$, for symmetry reasons.[4] Thus, we have $\big[(\mathbf{B} \circ \mathbf{\Gamma})(R')\big](o) = 0$ for all $o \in \mathcal{O}$, as we detail in Equation (27). This shows $0 \neq R' \in \ker(\mathbf{B} \circ \mathbf{\Gamma}) \cap \mathcal{V}'$, and consequently $0 \neq \mathbf{\Gamma}(R') \in \mathbf{\Gamma}(\ker(\mathbf{B} \circ \mathbf{\Gamma}) \cap \mathcal{V}') = \mathrm{Amb}^{\mathcal{M}_3}$ (Proposition 2.9), proving the claim that the ambiguity is nontrivial. Crucially, in order to construct $R'$, we needed to allow that the symmetry-related state-action pairs $(s_1', P)$ and $(s_1'', P)$ have different rewards.

---

[4]For showing this, recall that $(\mathbf{B} \circ \mathbf{\Gamma})_{o,(s,a)}$ is simply the expected number of times that state-action pair $(s, a)$ appears in a trajectory that gives rise to observation $o$. For each trajectory, the up-down mirrored trajectory creates the same observation, leading to the aforementioned symmetry.

### 3.3.4 Conclusion of the example

This example highlights that a priori knowledge of symmetry-invariant reward functions (via model $\mathcal{M}_2$ with its ambiguity $h^*(\mathcal{V})$, see Equation (17)) or symmetry-invariant features (via model $\mathcal{M}_1$) can help to infer the correct return function from the human's feedback. We highlight again that one could in practice work with model $\mathcal{M}_2$ even if $\mathcal{M}_1$ were the "true" model, the reason being that $\mathcal{M}_2$ covers $\mathcal{M}_1$ and has a vanishing ambiguity. Future work could analyze this case in more detail, by developing a more general theory of symmetry-invariant human belief models.

### 3.4 A proposal for belief model covering in practice

So far, our discussion has been purely theoretical: We showed that if a model $\widehat{\mathcal{M}} = (\widehat{\Omega}, \widehat{\Lambda}, \widehat{\mathcal{E}}, \widehat{\mathcal{V}})$ is complete and covers the true belief model $\mathcal{M} = (\Omega, \Lambda, \mathcal{E}, \mathcal{V})$, then the true return function $G$ can can be inferred from the human's feedback $G_{\mathcal{O}}$ (Theorem 3.2, statement 4). This raises the following two questions:

1. How can $\widehat{\mathcal{M}}$ be specified?

2. How can $G$ be determined in practice, using $\widehat{\mathcal{M}}$ and $G_{\mathcal{O}}$?

We now give preliminary answers to these questions in Sections 3.4.1 and 3.4.2, based on using foundation models for both the ontology $\widehat{\Lambda}$ and the feature belief function $\widehat{\mathcal{E}}$. We hope our ideas can inspire future empirical work.

### 3.4.1 Defining $\widehat{\mathcal{M}}$ for answering question 1

To answer question 1, first, one needs to choose an MDP together with trajectories $\Xi$, and observations $\mathcal{O}$. Ideally, whole trajectories $\xi \in \Xi$ or parts of them, and all observations, can be "tokenized" so that one can feed them into foundation models. Let $\widehat{\lambda} : \Xi \to \mathbb{R}^{\widehat{\Omega}}$ be a foundation model, which we interpret as a function from trajectories to an internal representation space with $|\widehat{\Omega}|$-many neurons.[5] Then, define $\widehat{\Lambda} : \mathbb{R}^{\widehat{\Omega}} \to \mathbb{R}^{\Xi}$ as the linear function corresponding to $\widehat{\lambda}$ according to Proposition A.1.

For $\widehat{\lambda}$ to be a valid ontology that is part of a belief model that covers the true model, by Theorem 3.5, we need there to exist an (implicit) linear ontology translation $\Psi : \mathbb{R}^{\widehat{\Omega}} \to \mathbb{R}^{\Omega}$. There is substantial prior work showing that human concepts are represented linearly in foundation model's representation spaces (Mikolov et al., 2013; Park et al., 2024b; Turner et al., 2024; Nanda et al., 2023; Wang et al., 2023; Gurnee & Tegmark, 2024). Most relevant to our claims, sparse autoencoders (Cunningham et al., 2023; Bricken et al., 2023) directly construct a linear transformation that maps from a foundation model's representation space to a space of human-interpretable features, thus constructing a function akin to our (only implicitly needed) linear ontology translation $\Psi$.

Notably, $\widehat{\lambda}$ needs to be a *capable* foundation model for such a linear function to have any hope to be an exact ontology translation. After all, imagine we show $\widehat{\lambda}$ the Riemann hypothesis: If it is not vastly superhuman, then it cannot determine whether this hypothesis is *true*, which we consider an important feature that likely appears in the human's ontology $\lambda$. Since we are concerned with scalable oversight, which is about the problem of ensuring alignment of future, powerful AI systems, we assume $\widehat{\lambda}$ to be very capable, and thus that $\Psi$ is an exact ontology translation. Overall, this gives some preliminary confidence that for a capable foundation model $\widehat{\lambda} : \Xi \to \mathbb{R}^{\widehat{\Omega}}$, there will exist a linear ontology translation $\Psi : \mathbb{R}^{\widehat{\Omega}} \to \mathbb{R}^{\Omega}$, such that $\Psi \circ \widehat{\lambda} = \lambda$.

Now, let $\widehat{\epsilon} : \mathcal{O} \to \mathbb{R}^{\widehat{\Omega}}$ be another foundation model (in the proposals below it will be an adaptation of $\widehat{\lambda}$). Then, define $\widehat{\mathcal{E}} : \mathbb{R}^{\widehat{\Omega}} \to \mathbb{R}^{\mathcal{O}}$ as the linear functions corresponding to $\widehat{\epsilon}$ according to Proposition A.1. Set $\widehat{\mathcal{V}} = \mathbb{R}^{\widehat{\Omega}}$. Set $\widehat{\mathcal{M}} = (\widehat{\Omega}, \widehat{\Lambda}, \widehat{\mathcal{E}}, \widehat{\mathcal{V}})$.

---

[5]This means that we remove the output functionality from this model.

Why or when would this be a useful specification? As discussed before, we need $\widehat{\mathcal{M}}$ to be complete and to cover the (implicit) "true" belief model $\mathcal{M}$. Based on sufficient conditions we found in Theorem 3.5 and Theorem 2.13, we thus need to ensure the following two properties:

(i) The ontology translation $\Psi : \mathbb{R}^{\widehat{\Omega}} \to \mathbb{R}^{\Omega}$ is belief-respecting: $\Psi \circ \widehat{\epsilon} = \epsilon$.

(ii) For all $\xi \in \Xi$, $\widehat{\lambda}(\xi)$ is contained in the span of the image of $\widehat{\epsilon}$: $\widehat{\lambda}(\xi) \in \left\{ \sum_{o \in \mathcal{O}} Z_o \widehat{\epsilon}(o) \mid Z_o \in \mathbb{R} \right\}$.

**Ensuring (i).** This is the most speculative part of our proposal. Crucially, if the human does not recognize the presence of a feature in $o \in \mathcal{O}$ due to limited capabilities, then $\widehat{\epsilon}$ should ideally *also* not recognize this feature so that translating from one ontology into the other respects beliefs. Conceptually, this means that $\widehat{\epsilon}$ should *simulate*, in the feature space $\widehat{\Omega}$, the human's beliefs and understanding. We make three proposals:

- For research prototyping, one could restrict to a problem with "pure partial observability". In other words, choose a setting where observations are given as $o = O(\xi)$ for trajectories $\xi \in \Xi$, and where potentially superhuman capabilities do not make it easier to infer further return-relevant aspects of $\xi$ from $O(\xi)$. This is intuitively the case if information is simply entirely "missing" from observations. Then, simply choose $\widehat{\epsilon} := \widehat{\lambda}$, applied to observations instead of trajectories. In this case, the fact that crucial information is equally obstructed to the human with feature belief function $\epsilon$ and to the AI with $\widehat{\epsilon}$ should ensure that the ontology translation also correctly translates feature beliefs: $\Psi \circ \widehat{\epsilon} = \epsilon$.

- Now consider a setting that may go beyond "pure partial observability". One speculative idea for how to construct $\widehat{\epsilon}$ is to prepend a "belief prompt" bp to inputs of $\widehat{\lambda}$ that nudges the model to think more in the way a human evaluator would think (Park et al., 2024a):

$$\widehat{\epsilon}(o) := \widehat{\lambda}_{\mathrm{bp}}(o) := \widehat{\lambda}(\mathrm{bp}, o).$$

  An example of such a prompt would be

    bp = "Think about the following input like a typical human evaluator:"

  Alternatively, one could potentially achieve this by finetuning $\widehat{\lambda}$ to obtain $\widehat{\epsilon}$. Unfortunately, while foundation models can simulate the behavior of specific people in their *outputs*, it is unclear whether this also reflects in their *internal representations*. For example, prior work shows that the truth-value of statements can sometimes be linearly predicted from internal representations even when the model lies, leading to the potential for AI lie detectors (Azaria & Mitchell, 2023; Burns et al., 2023b). However, other work trains a foundation model to predict human behavior in experiments and finds it to have internal representations that can predict human neural activity when engaging in the same task (Binz et al., 2024).

- Alternatively, one could also consider defining $\widehat{\epsilon} := \widehat{\lambda}_{\mathrm{early}}$ as an earlier training checkpoint of $\widehat{\lambda}$. Being an earlier training checkpoint, $\widehat{\epsilon}$ would then be less capable than $\widehat{\lambda}$, leading to the potential that it has the same blindspots in understanding observations $o \in \mathcal{O}$ as the human evaluator.

**Ensuring (ii).** To ensure that for all $\xi \in \Xi$ we have $\widehat{\lambda}(\xi) \in \left\{ \sum_{o \in \mathcal{O}} Z_o \widehat{\epsilon}(o) \mid Z_o \in \mathbb{R} \right\}$, it is important to ensure that the image of $\widehat{\epsilon} : \mathcal{O} \to \mathbb{R}^{\widehat{\Omega}}$ is "large", i.e., spans as much as possible of the representation space. To form more intuitions on this, note that by Theorem 3.2, $\widehat{\mathcal{M}}$ being complete (which would be implied by property (ii)) requires also the true belief model $\mathcal{M}$ to be complete. Again, by Theorem 2.13, a sufficient condition for this is that for all $\xi \in \Xi$, $\lambda(\xi) \in \left\{ \sum_{o \in \mathcal{O}} Z_o \epsilon(o) \mid Z_o \in \mathbb{R} \right\}$. In particular, any "bad" feature $\omega \in \Omega$ that can ever be present in a trajectory (meaning $[\lambda(\xi)](\omega) \neq 0$) needs to also sometimes be *believed* to be present by the human (meaning there exists an $o \in \mathcal{O}$ with $[\epsilon(o)](\omega) \neq 0$). This is a relaxation from the requirement that the human understands *all* observations perfectly, but it does mean that there needs to be a variety of observations $o \in \mathcal{O}$ that the human understands well enough to detect their diverse underlying problems. It will depend on the specific MDP and setup of an experiment to reason about how to ensure or purposefully violate this property.

### 3.4.2   Learning $G$ using $\widehat{\mathcal{M}}$ and $G_{\mathcal{O}}$ for answering question 2

Now that we have discussed preliminary approaches for how to specify a complete model $\widehat{\mathcal{M}} = \big(\widehat{\Omega}, \widehat{\mathbf{\Lambda}}, \widehat{\mathcal{E}}, \widehat{\mathcal{V}}\big)$ that covers the true belief model $\mathcal{M}$, we can turn to the question of how to use $\mathcal{M}$ and the human's feedback $G_{\mathcal{O}} : \mathcal{O} \to \mathbb{R}$ to determine the true return function $G$. By statement 4 in Theorem 3.2, we can determine $G$ as $G = \widehat{\mathbf{\Lambda}}(R_{\widehat{\Omega}})$ for $R_{\widehat{\Omega}} \in \mathbb{R}^{\widehat{\Omega}}$ with $\widehat{\mathcal{E}}(R_{\widehat{\Omega}}) = G_{\mathcal{O}}$.

We now unpack that. Remembering the relation between $\widehat{\mathbf{\Lambda}}$ and the foundation model $\widehat{\lambda}$, and $\widehat{\mathcal{E}}$ and the foundation model $\widehat{\epsilon}$ via Proposition A.1, we want to determine $R_{\widehat{\Omega}}$ such that for all $o \in \mathcal{O}$, we have

$$G_{\mathcal{O}}(o) = \big[\widehat{\mathcal{E}}(R_{\widehat{\Omega}})\big](o) = \langle\widehat{\epsilon}(o), R_{\widehat{\Omega}}\rangle. \tag{18}$$

This can be achieved by attaching $R_{\widehat{\Omega}}$ as a linear reward probe to the representation space of $\widehat{\epsilon}$ and learning the function $G_{\mathcal{O}}$ by supervised learning, with all parameters except the reward head being frozen.

In the case that $G_{\mathcal{O}}$ cannot be directly evaluated but that it is indirectly accessible in the form of *choice probabilities* between observations, one can learn $R_{\widehat{\Omega}}$ by logistic regression as in Christiano et al. (2017). See Appendix B.2 for a possible correspondence between $\widehat{\mathcal{E}}(R_{\widehat{\Omega}})$ and the resulting choice probabilities. Notably, the approach via logistic regression, however, loses a theoretical guarantee: Namely, $G_{\mathcal{O}}$ can at best be learned up to an additive constant. If $\widehat{\epsilon}$ and $\widehat{\lambda}$ were *balanced* (meaning that the total weight of feature strengths is constant over all observations and trajectories, respectively), this would lead to the inferred $G$ also being correct up to an additive constant by Proposition B.6. Since additive constants in return functions are inconsequential for policy optimization, this would be fine. However, typically the representation spaces of foundation models are not normalized, the balancedness property does not hold, and this guarantee breaks. It is then an empirical question to what extent this breakage is an issue or how to resolve it.

After successful learning, we can then compute the true return function $G$ for $\xi \in \Xi$ as:

$$G(\xi) = \big[\widehat{\mathbf{\Lambda}}(R_{\widehat{\Omega}})\big](\xi) = \langle\widehat{\lambda}(\xi), R_{\widehat{\Omega}}\rangle. \tag{19}$$

In other words, we attach the learned linear reward probe $R_{\widehat{\Omega}}$ to the representation space of $\widehat{\lambda}$ and use it to compute returns. These can then be used to train a policy to maximize the policy evaluation function Equation (4) via standard reinforcement learning techniques.

### 3.4.3   Further remarks

Looking at the definition of the human belief model $\widehat{\mathcal{M}}$, we see that it includes the ontology $\widehat{\mathbf{\Lambda}} : \mathbb{R}^{\widehat{\Omega}} \to \mathbb{R}^{\Xi}$ and feature belief function $\widehat{\mathcal{E}} : \mathbb{R}^{\widehat{\Omega}} \to \mathbb{R}^{\mathcal{O}}$. These can be extremely large matrices. However, since in the process of training $R_{\widehat{\Omega}}$ and computing $G$, we only need to be able to *query* the resulting observation return function and return function on specific observations and trajectories as in Equations (18) and (19), the matrices never need to be stored or used in their entirety. Thus, the size of the matrices is not a concern.

We also remark on a special case we mentioned before: If we are in a case of "pure" partial observability where observations $o \in \mathcal{O}$ contain no information that $\widehat{\lambda}$ understands better than the human, then we proposed to simply set $\widehat{\epsilon} := \widehat{\lambda}$, applied to observations $o \in \mathcal{O}$. In that case, the procedure we describe is essentially classical RLHF, with the only — crucial — difference that during training of $R_{\widehat{\Omega}}$, we only show observations, instead of full trajectories, to the reward model. This prevents a model misspecification where the data the model reads differs from what the human sees, and could theoretically allow generalization to data $\xi \in \Xi$. Lang et al. (2024) consider naive RLHF, where the initialized return function reads entire trajectories during training time while the human's view is obstructed, leading to issues of deceptive inflation and overjustification after policy optimization.

### 3.4.4   Limitations for the proposal

We conclude by summarizing the limitations of the proposal in its current form. At the very foundations, it assumes that real humans are compatible with a suitable belief model as defined in this work, that human

values are captured by return functions over trajectories, and that the model allows for a *linear* decomposition of the return function over features. All of these assumptions are speculative and may need refinement.

The proposal then assumes that $\widehat{\mathcal{M}}$ is an *exact* covering belief model for the latent human belief model. In particular, it is assumed that $\widehat{\lambda}$ reads entire trajectories and allows for the existence of a linear ontology translation to the true human ontology. Finally, the feature belief function $\widehat{\epsilon}$ needs to "translate" to the true human belief, and it is unclear whether our proposals can achieve this sufficiently. To achieve completeness, the proposal also requires careful environment design to ensure that the beliefs $\widehat{\epsilon}(o)$ "cover" the feature combinations $\widehat{\lambda}(\xi)$ sufficiently.

In conclusion, substantial future work is needed to validate the utility of the proposal. We propose such work in Section 4.3 for many of the limitations above.

## 4 Discussion

In this discussion, we summarize our work, survey related work, propose ideas for future work, and conclude.

### 4.1 Summary

In this work, we have introduced the notion of a human belief model, based on modeling a human's ontology and feature belief function. The goal of such a model is to aid the inference of the human's implicit return function from feedback. In our framework, the feedback, in the form of an observation return function, is viewed as carrying information about the reward of features that the human *believes* to be associated with an observation. Once the feature rewards, in the form of a reward object within a valid set, are inferred, they can be used together with the human's ontology to infer a return function. We defined and characterized the resulting *ambiguity* in the return function in terms of the belief model and showed that for *complete* models, the ambiguity disappears. Complete models have an important sufficient condition in terms of a linear coverage of the features of any trajectory by feature beliefs of observations (Theorem 2.13). This shows that for observations that are varied enough and provide ample information in total, a correct return function inference is possible, which we demonstrated in simple conceptual examples in Section 2.9.

Since the human belief model is not known in practice, we then introduced the notion of belief model *coverage*. Here, a model $\widehat{\mathcal{M}}$ covers another model $\mathcal{M}$ if $\widehat{\mathcal{M}}$ can represent all return functions and observation return functions that can be expressed in $\mathcal{M}$. If a complete model covers the true belief model, then it can be used for the return function inference just as well (Theorem 3.2). We then characterized belief model coverage in terms of belief model morphisms, and found an important sufficient condition given by the existence of a linear translation from the covering model's ontology to the true model's ontology that is also compatible with the feature belief functions (Theorem 3.5). We then studied a conceptual example of a human with symmetry-invariant feature beliefs, which we could cover with a model whose completeness stems from the valid reward functions being symmetry-invariant.

Finally, in Section 3.4 we sketched a proposal for how to find covering human belief models in practice, by using foundation models for both the human's ontology and feature belief function. For the latter, it is important to ensure that the foundation model has a similar understanding of the observtions as the human evaluator, for which we sketched out three proposals. That the resulting belief model might cover the true belief model relies on prior research on the linear representation hypothesis, which provides evidence that a belief-respecting linear ontology translation could indeed exist.[6] Our hope is that our proposal can help to find covering models that are easier to determine than the return function itself, which might subsequently be learned with modest effort as an approach to scalable oversight (cf. Section 2.5).

### 4.2 Related work

**Other human modeling approaches.** The human belief models we introduce in this work are largely about modeling a human's ontology and feature belief function for learning from the human's feedback.

---

[6]We emphasize again that our theory only requires its existence and no explicit specification of this ontology translation.

Similar work (Marklund & Roy, 2024) is about the special case of humans observing *partial trajectories* and forming beliefs over the rest of the trajectory.

There is also work that models aspects about humans different from their beliefs. For example, reward-rational choice (Jeon et al., 2020) requires modeling human choice probabilities for choices over various options like trajectory-pairs, language-utterances, or initial environment states. An example of this framework is inverse reinforcement learning (Ng et al., 2000), which has also been considered in partially observable environments (Choi & Kim, 2011). Reward-rational choice can be regarded a special case of assistance problems (Fern et al., 2014; Hadfield-Menell et al., 2016; Shah et al., 2021), which requires a model of the human's action selection in a cooperative two-player game. This framework has recently been generalized to a partially observable setting (Emmons et al., 2024). Much of this work makes specific assumptions about the human's model, e.g. by assuming a Boltzmann-rational or optimal selection of choices. Zhi-Xuan et al. (2024) instantiates a version of assistance games in which a human's utterance is modeled using a language model. Finally, Hatgis-Kessell et al. (2025) researches how to influence human evaluators to conform with a theoretical model of human choices. Compared to all this work, we model human *beliefs* about AI behavior instead of human actions or choices.[7] In Section 4.3.1 we propose research directions to combine these types of work.

**Deception in AI.** Our work generalizes the human belief modeling from Lang et al. (2024), which is meant to address deceptive AI behavior that also surfaces in recent empirical work for cases where humans lack evaluative capacity (Cloud et al., 2024; Denison et al., 2024; Wen et al., 2024; Williams et al., 2024). Deception in AI systems can also occur for various other reasons (Park et al., 2024c), e.g., when language agents are put under pressure (Scheurer et al., 2023). An important theoretical concern is deceptive alignment, in which an AI system follows the given goals while actually planning a later takeover (Hubinger et al., 2019). Recent work (Greenblatt et al., 2024) substantiates this concern by showing that the language model Claude plays along with a new *harmful* goal for the purpose of preventing that the learning process updates its safety behavior in the long-term. Finally, deception has also been formalized for structural causal games (Ward et al., 2023).

**Surfacing latent knowledge.** In Section 3.4, we already mentioned sparse autoencoders (Cunningham et al., 2023; Bricken et al., 2023), which construct an explicit linear transformation from a foundation model's representation space to human-interpretable features, which is in the spirit of a linear ontology translation (Definition 3.4). Instead of decoding the AI's *entire* ontology in a human-interpretable way, other paradigms seek to construct a reporter that can be queried for specific information (Christiano et al., 2021). In this direction, recent work builds toward AI lie detectors by finding internal linear representations of truth (Burns et al., 2023b; Azaria & Mitchell, 2023; Marks & Tegmark, 2024), with alternative interpretations of such findings discussed in Liu et al. (2023). Other work linearly predicts concepts like harmfulness (Zou et al., 2024), theft advice (Roger, 2023), the activation of a harmful backdoor (MacDiarmid et al., 2024), and concepts related to honesty and power, among others (Zou et al., 2023).

**Amplified oversight.** While our work aims to learn from feedback of humans who potentially lack capabilities, work on amplified oversight tries to *increase* the human's evaluation capabilities through AI assistance to achieve scalable oversight. Recursive reward modeling is the general proposal to train AI models by reward modeling and then using their assistance to evaluate the next generation of AIs (Leike et al., 2018). Saunders et al. (2022) shows that model-written critiques of summaries can help humans find flaws that they would have missed on their own. This raises the question why to trust the critiquing AI, which leads to the idea to, in turn, criticize the critic. Recursively, this leads to a debate in which the debaters are trained to produce arguments that are persuasive to a human judge (Irving et al., 2018). This requires for debates to surface true and useful information to the human judges, which has found support for reading comprehension tasks (Michael et al., 2023). Optimizing debaters to be persuasive then increases the human's ability to identify the truth (Khan et al., 2024; Kenton et al., 2024). Finally, some work uses AI to directly give feedback based on a constitution (Bai et al., 2022) or model spec (Guan et al., 2025), thus removing humans from the evaluation process.

---

[7] Only in Appendix B do we consider human choices.

**Easy-to-hard and weak-to-strong generalization.** Instead of amplifying the evaluator capabilities, other work for scalable oversight relies on easy examples that humans can reliably evaluate, which then requires the reward model to generalize to data that is harder to evaluate (Sun et al., 2024). Language models have also shown to generalize tasks like STEM questions from easy to hard data (Hase et al., 2024). Weak-to-strong generalization differs by trying to generalize from weak *evaluation* on potentially *hard* data (Burns et al., 2023a). Our proposal from Section 3.4 can be considered an approach to weak-to-strong generalization since we aim to learn a correct return function $G$ from feedback of an evaluator with potentially faulty beliefs. Since weak supervision is plausibly cheaper to obtain than strong supervision, there is also work that investigates tradeoffs to find the correct allocation of a fixed budget to label data with different data labeling qualities (Mallen & Belrose, 2024).

In Appendix E, we briefly interpret amplified oversight, easy-to-hard generalization, and classical RLHF together with weak-to-strong generalization in terms of our theoretical framework by interpreting their underlying evaluator belief modeling assumptions and reasoning about their ambiguities and learned return functions.

### 4.3 Future work

#### 4.3.1 Theory extensions

Several extensions and generalizations of our theory could be studied in future work. We assumed that we can exactly specify a belief model that covers the true human belief model. Future work could develop an approximate theory, where there is a quantifiable error in the belief model. This could proceed along similar lines as Lang et al. (2024, Theorem 5.4), where the belief matrix is perturbed by a known error, leading to a quantifiable error in the inferred return function. Furthermore, we assumed that the human's return function is linear in the features of trajectories. One could study non-linear models, which would also theoretically ground the use of non-linear reward probes instead of the linear probes we propose in Section 3.4. We also assumed that learned return functions can read entire trajectories, which might be unrealistic for very complex environments. It would thus be interesting to develop a theory based on a second set of observations $\mathcal{O}'$ for the learned return function and the trained policy. In the practical proposal, this would also mean that we cannot assume to have a foundation model $\widehat{\lambda}$ that translates to the human's true ontology — instead, it would represent another feature belief function. Relaxing the capabilities of the foundation model could also help to model a case in which the human evaluator has information that is hidden from the learned return function or resulting policy.

The ambiguity is a measure of the *information* that is available in the feedback, given a belief model, for determining the human's return function. Future work could theoretically study concrete learning procedures, which are about *extracting* said information. One could, for example, study training distributions over the observations $\mathcal{O}$ to determine sample complexity bounds for the error of the resulting return function, or the regret of the resulting policy. This is theoretically interesting since the usage of the learned return function in policy optimization involves two shifts compared to the return function's learning process: First, it needs to evaluate trajectories instead of observations, making use of an ontology instead of a feature belief function; and second, the policy optimization leads to a further distribution shift over trajectories, which can in turn lead to increased regret even if the return function seemed accurate before (Fluri et al., 2024). Finally, if there is remaining ambiguity, it would also be desirabe to study protocols for choosing a return function *within* the ambiguity, possibly using a priori knowledge about "human-like" return functions that is not captured by our notion of valid reward objects.

Going beyond our framework, one could attempt a synthesis with work that models the human's *action selection*, which we discussed at the start of Section 4.2. For example, one could consider reward-rational choice or assistance games under the assumption that humans form beliefs based on observations. Emmons et al. (2024) does a first step in this direction by assuming humans form rational beliefs based on knowledge of a prior of the agent's policy. In that framework, they then study the notion of information interference, which leads to an increase in the human's *uncertainty* about the world state. If future work were to study more general, possibly faulty, belief models for humans in partially observable assistance games, this could make it possible to go beyond information interference to study deception.

### 4.3.2 Empirical work

We would be interested in attempts to instantiate our proposal from Section 3.4. One could test the proposal in settings with synthetic humans with a known ontology and feature belief function, which can for small MDPs allow to compute the ambiguity explicitly. This should make it possible to make concrete predictions about experimental results. It would also be interesting to study settings with partial observability in which capable AI does *not* have an advantage over humans in understanding the meaning of the observations. As we discussed in our proposal, that should allow to use the same foundation model for the ontology and feature belief function, with the latter only applied to observations. This could then be compared with a baseline of "naive" RLHF in which the initialized return function reads entire trajectories during the learning process, similar to the conceptual examples from Lang et al. (2024). It would be interesting to study different levels of observability to dial the ambiguity up or down. We also encourage to break some of our theoretical assumptions, e.g., by using non-linear reward probes. Finally, it would be desirable to learn how our approach can be *combined* with approaches in the direction of amplified oversight, easy-to-hard generalization, or weak-to-strong generalization discussed in Section 4.2. For example, a combination of our work with easy-to-hard generalization could seek to only train the reward probe on data where the human beliefs are *modeled correctly* by $\widehat{\epsilon}$, which should be a superset of easy data.

Instead of studying the whole pipeline, one could also empirically assess the underlying assumptions. The most precarious assumption is that it is possible to construct a feature belief function $\widehat{\epsilon}$, for which we discuss proposals in Section 3.4.1. In situations that are not purely based on partial observability, we proposed to prompt or finetune the model to step "into the shoes" of a human evaluator, or to use earlier training checkpoints of a model to decrease its capabilities to that of a human evaluator (assuming future models that would otherwise be superhuman). For this to work, there needs to be a linear ontology translation that is compatible with $\widehat{\epsilon}$ and the human's true feature belief function $\epsilon$. One could test this empirically by using sparse autoencoders (Cunningham et al., 2023; Bricken et al., 2023) trained on an unobstructed capable foundation model, and evaluating the resulting human-readable features when applied to $\widehat{\epsilon}$.

### 4.4 Conclusion

In this work we theoretically showed that *if* we could model human beliefs about AI behavior, then under certain conditions, this could help for the correct inference of human goals from human feedback. Since the theory applies even in cases where the human's beliefs are erroneous and based on partial observations, it is relevant to a setting of scalable oversight where the AI is more capable than the human overseers. We hope that future work will build on our theory, empirically study our practical proposal, or engage in other research on how to make AI systems safe and aligned with the goal of reducing the risks of advanced AI.

### Author contributions

Leon Lang developed the ideas, derived all theoretical results, and wrote the paper. Patrick Forré gave general guidance and feedback.

### Acknowledgments and disclosure of funding

We thank Scott Emmons, Davis Foote, Erik Jenner, Micah Carroll, and Alex Cloud for discussions that shaped how we think about human evaluators with limited capabilities. We thank Open Philanthropy for financial support.

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

# Appendices

In the appendices, we present mathematical details and content that goes beyond the main text. Appendix A lists preliminary results on linear algebra together with their proofs. In Appendix B, we present a theory of *balanced* belief models and the ambiguity for feedback that is given by binary choices between observations. This complements the core theory from Sections 2 and 3, where the feedback is given by an observation return function. In Appendix C, we complement Section 3 by showing that human belief models and their morphisms form a category, and we construct a variety of natural models composing a diagram. Appendix D contains further mathematical details for the example in Section 3.3 on symmetry-invariant belief models and reward functions. Finally, Appendix E interprets some of the related work from Section 4.2 in our framework.

## A    Preliminary results on linear algebra

For general notation and conventions on linear algebra, see Section 2.1.

### A.1    Different representations of linear functions

Let $X, Y$ be two sets. Then by $\mathrm{Lin}(\mathbb{R}^X, \mathbb{R}^Y)$ we denote the set of all linear maps $F : \mathbb{R}^X \to \mathbb{R}^Y$. By $\mathrm{Maps}(Y, \mathbb{R}^X)$, we denote the set of all (simple) maps, or functions, $f : Y \to \mathbb{R}^X$. Intuitively, these encode the same information: A linear function $F$, when represented as a matrix, is a collection of rows indexed by $y \in Y$, which are "picked out" by a function $f : Y \to \mathbb{R}^X$. This correspondence is made precise in the following proposition, which is a "transposed" version of the classical statement that linear functions on vector spaces correspond to functions on a basis:

**Proposition A.1.** *Define the two functions*

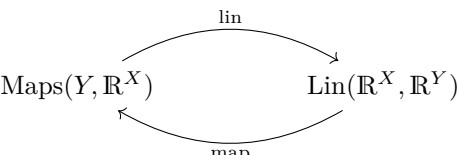

*as follows: For $f \in \mathrm{Maps}(Y, \mathbb{R}^X)$, $v \in \mathbb{R}^X$ and $y \in Y$, we define*

$$\Big[ \big[ \mathrm{lin}(f) \big](v) \Big](y) := \langle f(y), v \rangle .$$

*For $F \in \mathrm{Lin}(\mathbb{R}^X, \mathbb{R}^Y)$, $y \in Y$ and $x \in X$, we define*

$$\Big[ \big[ \mathrm{map}(F) \big](y) \Big](x) := F_{yx}.$$

*Then $\mathrm{lin}(f)$ has matrix elements*

$$\mathrm{lin}(f)_{yx} = \big[ f(y) \big](x)$$

*for $x \in X$ and $y \in Y$. Furthermore,* $\mathrm{lin}$ *and* $\mathrm{map}$ *are mutually inverse bijections.*

*Proof.* First, note that for each $f \in \mathrm{Maps}(Y, \mathbb{R}^X)$, $\mathrm{lin}(f) : \mathbb{R}^X \to \mathbb{R}^Y$ is indeed a linear function since the scalar product is linear in the second component. Its matrix elements are given by

$$\mathrm{lin}(f)_{yx} = \Big[ \big[ \mathrm{lin}(f) \big](e_x) \Big](y) = \langle f(y), e_x \rangle = \big[ f(y) \big](x).$$

Now we show that lin and map are mutually inverse bijections, i.e., $\mathrm{lin} \circ \mathrm{map} = \mathrm{id}_{\mathrm{Lin}(\mathbb{R}^X, \mathbb{R}^Y)}$ and $\mathrm{map} \circ \mathrm{lin} = \mathrm{id}_{\mathrm{Maps}(Y, \mathbb{R}^X)}$. Indeed, for $F \in \mathrm{Lin}(\mathbb{R}^X, \mathbb{R}^Y)$, $x \in X$, and $y \in Y$, we have

$$\big[ (\mathrm{lin} \circ \mathrm{map})(F) \big]_{yx} = \big[ \mathrm{lin}(\mathrm{map}(F)) \big]_{yx}$$

$$\begin{aligned}
&= \Big[\big[\operatorname{lin}(\operatorname{map}(F))\big](e_x)\Big](y) \\
&= \Big\langle \big[\operatorname{map}(F)\big](y), e_x \Big\rangle \\
&= \Big[\big[\operatorname{map}(F)\big](y)\Big](x) \\
&= F_{yx}
\end{aligned}$$

Since linear functions are fully characterized by their matrix elements, this shows $(\operatorname{lin} \circ \operatorname{map})(F) = F$, and thus $\operatorname{lin} \circ \operatorname{map}$ is the identity.

For the other direction, for $f \in \operatorname{Maps}(Y, \mathbb{R}^X)$, $y \in Y$, and $x \in X$, we have

$$\begin{aligned}
\Big[\big[(\operatorname{map} \circ \operatorname{lin})(f)\big](y)\Big](x) &= \Big[\big[\operatorname{map}(\operatorname{lin}(f))\big](y)\Big](x) \\
&= \operatorname{lin}(f)_{yx} \\
&= \big[f(y)\big](x).
\end{aligned}$$

This shows that $(\operatorname{map} \circ \operatorname{lin})(f) = f$, and so $\operatorname{map} \circ \operatorname{lin}$ is also the identity. $\square$

**Proposition A.2.** *Let $X, \widehat{X}, Y$ be sets and $F : \mathbb{R}^X \to \mathbb{R}^Y$, $\widehat{F} : \mathbb{R}^{\widehat{X}} \to \mathbb{R}^Y$, and $\Phi : \mathbb{R}^X \to \mathbb{R}^{\widehat{X}}$ be linear functions. Let $f = \operatorname{map}(F) : Y \to \mathbb{R}^X$ and $\widehat{f} = \operatorname{map}(\widehat{F}) : Y \to \mathbb{R}^{\widehat{X}}$ be the functions corresponding to $F$ and $\widehat{F}$ by Proposition A.1. Let $\Phi^T : \mathbb{R}^{\widehat{X}} \to \mathbb{R}^X$ be the transpose of $\Phi$, with matrix elements $\Phi^T_{x\widehat{x}} = \Phi_{\widehat{x}x}$. Then $F = \widehat{F} \circ \Phi$ if and only if $f = \Phi^T \circ \widehat{f}$:*

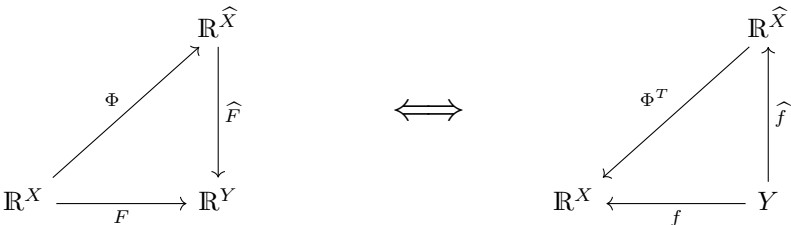

*Proof.* We have

$$\begin{aligned}
F = \widehat{F} \circ \Phi \quad &\Longleftrightarrow \quad \forall y \in Y, x \in X \colon F_{yx} = (\widehat{F} \circ \Phi)_{yx} = \sum_{\widehat{x} \in \widehat{X}} \widehat{F}_{y\widehat{x}} \Phi_{\widehat{x}x} \\
&\Longleftrightarrow \quad \forall y \in Y, x \in X \colon \big[f(y)\big](x) = \sum_{\widehat{x} \in \widehat{X}} \Phi^T_{x\widehat{x}} \big[\widehat{f}(y)\big](\widehat{x}) = \big\langle \Phi^T_x, \widehat{f}(y) \big\rangle \\
&\qquad\qquad\qquad\qquad = \Big[\Phi^T\big(\widehat{f}(y)\big)\Big](x) = \big[(\Phi^T \circ \widehat{f})(y)\big](x) \\
&\Longleftrightarrow \quad f = \Phi^T \circ \widehat{f}.
\end{aligned}$$

That was to show. $\square$

### A.2 Properties of kernels and images of linear functions

The following two propositions list basic and well-known properties of kernels and images that we use in the paper:

**Proposition A.3.** *Let $A : \mathcal{V} \to \mathcal{U}$ and $B : \mathcal{V} \to \mathcal{W}$ be linear functions. Then the following statements are equivalent:*

*1. $\ker(A) \subseteq \ker(B)$;*

2. *There exists a linear function $C : \mathcal{U} \to \mathcal{W}$ with $C \circ A = B$:*

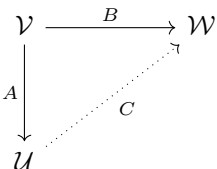

*Proof.* Clearly, the second claim implies the first. So now assume 1. Let $\{u^1, \ldots, u^m\}$ be a basis for $\operatorname{im}(A)$ and complement it to a basis $\{u^1, \ldots, u^n\}$ for all of $\mathcal{U}$, where $n \geq m$. For each $i \in \{1, \ldots, m\}$, let $v^i \in \mathcal{V}$ be any element with $A(v^i) = u^i$. Define $C(u^i) := B(v^i)$ for $i \in \{1, \ldots, m\}$ and $C(u^i) = 0$ if $i > m$. Linearly extend $C$ to a linear function $C : \mathcal{U} \to \mathcal{W}$.

To show that $C \circ A = B$, let $v \in \mathcal{V}$ be arbitrary. Then $A(v) \in \operatorname{im}(A)$, and thus there exist coefficients $\lambda_i \in \mathbb{R}$ for $i \in \{1, \ldots, m\}$ with

$$A(v) = \sum_{i=1}^{m} \lambda_i u^i.$$

Note that

$$A\left(v - \sum_{i=1}^{m} \lambda_i v^i\right) = A(v) - \sum_{i=1}^{m} \lambda_i A(v^i) = A(v) - \sum_{i=1}^{m} \lambda_i u^i = 0$$

Thus, $v - \sum_{i=1}^{m} \lambda_i v^i \in \ker(A) \subseteq \ker(B)$. Consequently, we obtain

$$B(v) = B\left(\sum_{i=1}^{m} \lambda_i v^i\right) = \sum_{i=1}^{m} \lambda_i B(v^i) = \sum_{i=1}^{m} \lambda_i C(u^i)$$
$$= C\left(\sum_{i=1}^{m} \lambda_i u^i\right) = C\big(A(v)\big) = (C \circ A)(v).$$

This shows $C \circ A = B$, and thus the claim. $\qquad\square$

**Proposition A.4.** *Let $A : \mathcal{U} \to \mathcal{W}$ and $B : \mathcal{V} \to \mathcal{W}$ be linear functions. Then the following statements are equivalent:*

1. $\operatorname{im}(A) \subseteq \operatorname{im}(B)$.

2. *There exists a "lift", i.e., a linear map $C : \mathcal{U} \to \mathcal{V}$ with $B \circ C = A$:*

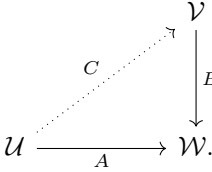

*Proof.* It can easily be checked that the second claim implies the first. For the other direction, let $\{u^1, \ldots, u^n\}$ be a basis for $\mathcal{U}$. For $i \in \{1, \ldots, n\}$, choose $v^i \in \mathcal{V}$ with $B(v^i) = A(u^i)$, which exists since $\operatorname{im}(A) \subseteq \operatorname{im}(B)$. Define $C : \mathcal{U} \to \mathcal{V}$ as the unique linear function with $C(u^i) = v^i$ for $i \in \{1, \ldots, n\}$. We obtain

$$A(u^i) = B(v^i) = B\big(C(u^i)\big) = (B \circ C)(u^i).$$

Since linear functions are determined on a basis, it follows $A = B \circ C$, proving the claim. $\qquad\square$

# B   Balanced human belief models and choices

In this appendix, we present the core theory from Section 2 and Section 3 for the case that the feedback is in the form of *choices* instead of the observation return function $G_{\mathcal{O}}$. To still get a useful theory, we will then need to assume our human belief models to be *balanced* to ensure that constants are "propagated" appropriately through the model. In this whole appendix, we fix an MDP with a set of trajectories $\Xi$ and observations $\mathcal{O}$.

## B.1   Balanced belief models

**Definition B.1** (Row-constant). *Let $X$, and $Y$ be sets. We call a linear function $A\colon \mathbb{R}^X \to \mathbb{R}^Y$ **row-constant** if for all $y, y' \in Y$ we have $0 \neq \sum_{x \in X} \boldsymbol{X}_{yx} = \sum_{x \in X} \boldsymbol{X}_{y'x}$.*

**Definition B.2** (Balanced). *Let $\mathcal{M} = (\Omega, \boldsymbol{\Lambda}, \mathcal{E}, \mathcal{V})$ be a human belief model. We call $\mathcal{M}$ **balanced** if $\boldsymbol{\Lambda}$ and $\mathcal{E}$ are row-constant, and if $\mathcal{V} \subseteq \mathbb{R}^\Omega$ contains all constant functions.*

$\boldsymbol{\Lambda}$ and $\mathcal{E}$ being row-constant means that the corresponding functions $\lambda : \Xi \to \mathbb{R}^\Omega$ and $\epsilon : \mathcal{O} \to \mathbb{R}^\Omega$ (cf. Proposition A.1) map to vectors of feature strengths with a constant *total* weighting, as is for example the case for probability distributions. That $\mathcal{V}$ contains all constant functions is often naturally the case. To demonstrate this in a simple example we first prove the following lemma:

**Lemma B.3.** *Let $X, Y, Z$ be sets and $A : \mathbb{R}^X \to \mathbb{R}^Y$, $B : \mathbb{R}^Y \to \mathbb{R}^Z$ be row-constant linear functions. Then the composition $B \circ A : \mathbb{R}^X \to \mathbb{R}^Z$ is also row-constant.*

*Proof.* Let $a, b$ be the row-sums of $A$ and $B$, respectively. Then for all $z \in Z$, we obtain

$$\sum_{x \in X} (B \circ A)_{zx} = \sum_{x \in X} \sum_{y \in Y} B_{zy} A_{yx} = \sum_{y \in Y} B_{zy} \sum_{x \in X} A_{yx} = b \cdot a \neq 0,$$

which shows the claim. $\qquad\qquad\square$

**Example B.4.** *We continue Example 2.6 and show that the model is balanced. Note that for all $\xi \in \Xi$, we have*

$$\sum_{(s,a,s') \in \mathcal{S} \times \mathcal{A} \times \mathcal{S}} \boldsymbol{\Gamma}_{\xi,(s,a,s')} = \sum_{(s,a,s') \in \mathcal{S} \times \mathcal{A} \times \mathcal{S}} \sum_{t=0}^{T-1} \gamma^t \delta_{(s,a,s')}(s_t, a_t, s_{t+1})$$

$$= \sum_{t=0}^{T-1} \gamma^t \sum_{(s,a,s') \in \mathcal{S} \times \mathcal{A} \times \mathcal{S}} \delta_{(s,a,s')}(s_t, a_t, s_{t+1})$$

$$= \sum_{t=0}^{T-1} \gamma^t$$

$$\neq 0.^8$$

*Thus, $\boldsymbol{\Gamma}$ is row-constant. $\mathbf{B}$ is also row-constant since all rows are probability distributions, and so Lemma B.3 implies that also $\mathcal{E} = \mathbf{B} \circ \boldsymbol{\Gamma}$ is row-constant. $\mathcal{V} = \mathbb{R}^{\mathcal{S} \times \mathcal{A} \times \mathcal{S}}$ also clearly contains all constant functions. Overall, this means the model $(\mathcal{S} \times \mathcal{A} \times \mathcal{S}, \boldsymbol{\Gamma}, \mathbf{B} \circ \boldsymbol{\Gamma}, \mathbb{R}^{\mathcal{S} \times \mathcal{A} \times \mathcal{S}})$ is balanced.*

For more examples of balanced human belief models, see Appendix C.2.

## B.2   The ambiguity for balanced belief models and choices

Let $\sigma : \mathbb{R} \to (0, 1)$ be any *known* bijective function with $\sigma(r) + \sigma(-r) = 1$ for all $r \in \mathbb{R}$ (e.g., the sigmoid function). For $o, o' \in \mathcal{O}$, we define the probability that a human with feature belief function $\mathcal{E} : \mathbb{R}^\Omega \to \mathbb{R}^{\mathcal{O}}$ and reward object $\tilde{R}_\Omega \in \mathbb{R}^\Omega$ prefers $o$ over $o'$ by

$$P_{\mathcal{E}}^{\tilde{R}_\Omega}(o \succ o') := \sigma\Big( \big[\mathcal{E}(\tilde{R}_\Omega)\big](o) - \big[\mathcal{E}(\tilde{R}_\Omega)\big](o') \Big). \qquad (20)$$

For the rest of the section, we fix a human belief model $\mathcal{M} = (\Omega, \mathbf{\Lambda}, \mathcal{E}, \mathcal{V})$. Furthermore, we fix the *implicit true reward object* $R_\Omega \in \mathcal{V}$ together with the return function $G = \mathbf{\Lambda}(R_\Omega)$, the observation return function $G_\mathcal{O} = \mathcal{E}(R_\Omega)$ and the choice probability function $P_\mathcal{O} \coloneqq P_\mathcal{E}^{R_\Omega}$ that serves as our operationalization of "feedback".

We use the following adaptation of Definition 2.8:

**Definition B.5.** *We define the set of return functions that are **feedback-compatible** with $P_\mathcal{O}$ as*

$$\mathrm{FC}^\mathcal{M}(P_\mathcal{O}) \coloneqq \left\{ \tilde{G} \in \mathbb{R}^\Xi \mid \exists \tilde{R}_\Omega \in \mathcal{V} \colon P_\mathcal{E}^{\tilde{R}_\Omega} = P_\mathcal{O} \text{ and } \mathbf{\Lambda}(\tilde{R}_\Omega) = \tilde{G} \right\}.$$

*We define the **ambiguity** left in the return function $G$ after the choice probability function $P_\mathcal{O}$ is known by*

$$\mathrm{Amb}^\mathcal{M}(G, P_\mathcal{O}) \coloneqq \left\{ G' \in \mathbb{R}^\Xi \mid G' = \tilde{G} - G \text{ for } \tilde{G} \in \mathrm{FC}^\mathcal{M}(P_\mathcal{O}) \right\}.$$

Clearly, we have

$$\mathrm{FC}^\mathcal{M}(G_\mathcal{O}) = G + \mathrm{Amb}^\mathcal{M}(G, P_\mathcal{O}).$$

Recall the ambiguity $\mathrm{Amb}^\mathcal{M}(G, G_\mathcal{O})$ defined in Definition 2.8. We obtain:

**Proposition B.6.** *Assume $\mathcal{M} = (\Omega, \mathbf{\Lambda}, \mathcal{E}, \mathcal{V})$ is balanced. Let $\mathbf{1} \in \mathbb{R}^\Xi$ denote the function that is constant 1. Then:*

$$\mathrm{Amb}^\mathcal{M}(G, P_\mathcal{O}) = \mathrm{Amb}^\mathcal{M}(G, G_\mathcal{O}) + \left\{ c \cdot \mathbf{1} \mid c \in \mathbb{R} \right\} = \mathbf{\Lambda}\big( \ker(\mathcal{E}) \cap \mathcal{V} \big) + \left\{ c \cdot \mathbf{1} \mid c \in \mathbb{R} \right\}.$$

*Proof.* The second equality follows from Proposition 2.9, so we are left with proving the first. By abuse of notation, we will write $\mathbf{1}$ for the three functions that are constant 1 on $\Xi$, $\mathcal{O}$, or $\Omega$.

Let $G' \in \mathrm{Amb}^\mathcal{M}(G, P_\mathcal{O})$. Then $G' = \mathbf{\Lambda}(\tilde{R}_\Omega) - G$ for $\tilde{R}_\Omega \in \mathcal{V}$ with $P_\mathcal{E}^{\tilde{R}_\Omega} = P_\mathcal{O}$. The latter means the following for all $o, o' \in \mathcal{O}$:

$$\sigma\Big( \big[\mathcal{E}(\tilde{R}_\Omega)\big](o) - \big[\mathcal{E}(\tilde{R}_\Omega)\big](o') \Big) = \sigma\Big( \big[\mathcal{E}(R_\Omega)\big](o) - \big[\mathcal{E}(R_\Omega)\big](o') \Big).$$

Since $\sigma$ is invertible, we obtain

$$\big[\mathcal{E}(\tilde{R}_\Omega)\big](o) - \big[\mathcal{E}(\tilde{R}_\Omega)\big](o') = \big[\mathcal{E}(R_\Omega)\big](o) - \big[\mathcal{E}(R_\Omega)\big](o')$$

Fix any $o' \in \mathcal{O}$ and set $c_\mathcal{O} \coloneqq \big[\mathcal{E}(\tilde{R}_\Omega)\big](o') - \big[\mathcal{E}(R_\Omega)\big](o')$. Then for all $o \in \mathcal{O}$, we have

$$\big[\mathcal{E}(\tilde{R}_\Omega)\big](o) = \big[\mathcal{E}(R_\Omega)\big](o) + c_\mathcal{O}.$$

Or, equivalently:

$$\mathcal{E}(\tilde{R}_\Omega) = \mathcal{E}(R_\Omega) + c_\mathcal{O} \cdot \mathbf{1} = G_\mathcal{O} + c_\mathcal{O} \cdot \mathbf{1}.$$

Since $\mathcal{E}$ is row-constant, there exists a $c_\Omega \in \mathbb{R}$ with $\mathcal{E}(c_\Omega \cdot \mathbf{1}) = c_\mathcal{O} \cdot \mathbf{1}$. This implies

$$\mathcal{E}(\tilde{R}_\Omega - c_\Omega \cdot \mathbf{1}) = G_\mathcal{O}.$$

We also have $\tilde{R}_\Omega - c_\Omega \cdot \mathbf{1} \in \mathcal{V}$ since $\mathcal{V}$ contains all constant functions. Furthermore, $\mathbf{\Lambda}(c_\Omega \cdot \mathbf{1}) = c_\Xi \cdot \mathbf{1}$ for another constant $c_\Xi \in \mathbb{R}$ since $\mathbf{\Lambda}$ is row-constant. Overall, we thus obtain

$$G' = \mathbf{\Lambda}(\tilde{R}_\Omega) - G = \mathbf{\Lambda}(\tilde{R}_\Omega - c_\Omega \cdot \mathbf{1}) - G + c_\Xi \cdot \mathbf{1} \in \mathrm{Amb}^\mathcal{M}(G, G_\mathcal{O}) + \left\{ c \cdot \mathbf{1} \mid c \in \mathbb{R} \right\}.$$

For the other direction, let $G' \in \mathrm{Amb}^\mathcal{M}(G, G_\mathcal{O}) + \left\{ c \cdot \mathbf{1} \mid c \in \mathbb{R} \right\}$. Then $G' = \tilde{G} - G + c_\Xi \cdot \mathbf{1}$ for $\tilde{G} = \mathbf{\Lambda}(\tilde{R}_\Omega)$ with $\tilde{R}_\Omega \in \mathcal{V}$ with $\mathcal{E}(\tilde{R}_\Omega) = G_\mathcal{O}$. We have

$$\tilde{G} + c_\Xi \cdot \mathbf{1} = \mathbf{\Lambda}(\tilde{R}_\Omega + c_\Omega \cdot \mathbf{1})$$

for a constant $c_\Omega \in \mathbb{R}$ with $\mathbf{\Lambda}(c_\Omega \cdot \mathbf{1}) = c_\Xi \cdot \mathbf{1}$. Since $\mathcal{V}$ contains all constant functions, we have $\tilde{R}_\Omega + c_\Omega \cdot \mathbf{1} \in \mathcal{V}$. We also have

$$P_\mathcal{E}^{\tilde{R}_\Omega + c_\Omega \cdot \mathbf{1}} = P_\mathcal{E}^{\tilde{R}_\Omega} = P_\mathcal{O}$$

since the constant gets cancelled out in the definition of the choice probabilities, and since $\mathcal{E}(\tilde{R}_\Omega) = G_\mathcal{O}$. All of this implies

$$G' = (\tilde{G} + c_\Xi \cdot \mathbf{1}) - G \in \mathrm{Amb}^\mathcal{M}(G, P_\mathcal{O}).$$

That proves the claim. $\qquad\square$

Note that for the purpose of policy optimization it is not an issue that the ambiguity has an "irreducible" constant term since this does not change the ordering of policies under the policy evaluation function $J(\pi) = \mathbf{E}_{\xi \in P^\pi(\cdot)}[G(\xi)]$.

**Remark B.7.** *In light of the previous proposition, it turns out that the ambiguity does not depend on the true return function and choice probabilities, and we can thus write it as $\mathrm{Amb}_P^\mathcal{M} = \mathrm{Amb}^\mathcal{M}(G, P_\mathcal{O})$. The "P" is added to distinguish from the ambiguity $\mathrm{Amb}^\mathcal{M} = \mathrm{Amb}^\mathcal{M}(G, G_\mathcal{O})$ that we study in the main paper.*

Using this result, we also obtain a version of Theorem 3.2, with the ambiguity replaced by the one we use in this appendix:

**Theorem B.8.** *Let $\mathcal{M} = (\Omega, \mathbf{\Lambda}, \mathcal{E}, \mathcal{V})$ and $\widehat{\mathcal{M}} = (\widehat{\Omega}, \widehat{\mathbf{\Lambda}}, \widehat{\mathcal{E}}, \widehat{\mathcal{V}})$ be two balanced human belief models and assume that $\widehat{\mathcal{M}}$ covers $\mathcal{M}$. We think of $\mathcal{M}$ as the "true" human belief model with reward object $R_\Omega \in \mathcal{V}$ and corresponding return function $G = \mathbf{\Lambda}(R_\Omega)$ and choice probability function $P_\mathcal{O} = P_\mathcal{E}^{R_\Omega}$. Then we have:*

1. *$\mathrm{Amb}_P^\mathcal{M} \subseteq \mathrm{Amb}_P^{\widehat{\mathcal{M}}}$.*

2. *If $\mathcal{M}$ also covers $\widehat{\mathcal{M}}$, then $\mathrm{Amb}_P^\mathcal{M} = \mathrm{Amb}_P^{\widehat{\mathcal{M}}}$.*

3. *There is an $R_{\widehat{\Omega}} \in \widehat{\mathcal{V}}$ with $P_{\widehat{\mathcal{E}}}^{R_{\widehat{\Omega}}} = P_\mathcal{O}$ and $\widehat{\mathbf{\Lambda}}(R_{\widehat{\Omega}}) = G$.*

4. *Assume $\widehat{\mathcal{M}}$ is complete. Then every reward object $\tilde{R}_{\widehat{\Omega}} \in \widehat{\mathcal{V}}$ with $P_{\widehat{\mathcal{E}}}^{\tilde{R}_{\widehat{\Omega}}} = P_\mathcal{O}$ also satisfies $\widehat{\mathbf{\Lambda}}(\tilde{R}_{\widehat{\Omega}}) = G + c_\Xi \cdot \mathbf{1}$ for a constant $c_\Xi \in \mathbb{R}$.*

*Proof.* Statements 1 and 2 follow from the ambiguity characterization in Proposition B.6 and the analogous statements in Theorem 3.2. Statements 3 and 4 can be proved with similar arguments as the corresponding statements in Theorem 3.2. $\qquad\square$

## C  A diagram in the category of human belief models

Let us consider an MDP together with a fixed set of trajectories $\Xi$ and observations $\mathcal{O}$. Then in Definition 2.4, we defined the notion of a human belief model $\mathcal{M} = (\Omega, \mathbf{\Lambda}, \mathcal{E}, \mathcal{V})$. In Definition 3.3, we then introduced the notion of a morphism $\Phi : \mathcal{M} \to \widehat{\mathcal{M}}$ between human belief models, which is defined as a linear function $\Phi : \mathbb{R}^\Omega \to \mathbb{R}^{\widehat{\Omega}}$ such that $\Phi(\mathcal{V}) \subseteq \mathcal{V}'$, $\widehat{\mathbf{\Lambda}} \circ \Phi|_\mathcal{V} = \mathbf{\Lambda}$ and $\widehat{\mathcal{E}} \circ \Phi|_\mathcal{V} = \mathcal{E}$. This notion turned out important since it is equivalent to model covering (Definition 3.1), which implies that the covering model can be used for the return function inference from human feedback (Theorem 3.2), especially if its ambiguity disappears.

Belief models for fixed sets of trajectories $\Xi$ and observations $\mathcal{O}$, together with their morphisms, form a category (Lane, 1998), meaning that they satisfy the following simple properties:

- **Composition**: Assume $\mathcal{M}_1, \mathcal{M}_2, \mathcal{M}_3$ are three human belief models and $\Phi : \mathcal{M}_1 \to \mathcal{M}_2$, $\Phi' : \mathcal{M}_2 \to \mathcal{M}_3$ morphisms between them. Then also the composition $\Phi' \circ \Phi : \mathcal{M}_1 \to \mathcal{M}_3$ is a morphism.

- **Identities**: For any human belief model $\mathcal{M} = (\Omega, \mathbf{\Lambda}, \mathcal{E}, \mathcal{V})$, the identity $\mathrm{id}_{\mathbb{R}^\Omega} : \mathcal{M} \to \mathcal{M}$ is a morphism.

- **Associativity**: $(\Phi'' \circ \Phi') \circ \Phi = \Phi'' \circ (\Phi' \circ \Phi)$ for any three morphisms that can be composed in the specified order.

All of these properties can be trivially checked, and so human belief models and their morphisms indeed form a category.

In this appendix, we want to write down a simple commutative diagram of morphisms in this category. Here, a diagram means a graph of human belief models and morphisms between them. For this to be *commutative* means that any pathway from one human belief model to another is the same morphism. We prepare this in Appendix C.1 by writing down all linear functions from which the functions $\mathbf{\Lambda}$, $\mathcal{E}$, and $\Phi$ will be constructed. In Appendix C.2 we then specify the resulting human belief models and briefly consider their properties. In Appendix C.3 we interpret the matrix elements that appear in the feature belief functions of all models. Finally, in Appendix C.4, we write down the resulting commutative diagram and the resulting relations for the ambiguities.

### C.1 Preparing the models

We build on Example 2.6. The idea is that we consider reward objects at four different levels: Return functions, classical reward functions, and return- and reward functions of *abstractions* of trajectories and transitions that the human might care about. By modeling the human as having features at all four of these different levels, we can create a multitude of human belief models.

Let $\mathbf{B} : \mathbb{R}^{\Xi} \to \mathbb{R}^{\mathcal{O}}$ be the matrix corresponding to a trajectory-belief function $b : \mathcal{O} \to \Delta(\Xi) \subseteq \mathbb{R}^{\Xi}$ via Proposition A.1. Let $\mathbf{\Gamma} : \mathbb{R}^{\mathcal{S} \times \mathcal{A} \times \mathcal{S}} \to \mathbb{R}^{\Xi}$ be the linear function mapping reward functions to their corresponding return functions.

Let $\mathcal{F}$ be a set of "abstractions of transitions" and $h : \mathcal{S} \times \mathcal{A} \times \mathcal{S} \to \mathcal{F}$ a function mapping each transition to its abstraction. Write reward objects over abstractions as $R_{\mathcal{F}} \in \mathbb{R}^{\mathcal{F}}$. Then we obtain the induced map

$$h^* : \mathbb{R}^{\mathcal{F}} \to \mathbb{R}^{\mathcal{S} \times \mathcal{A} \times \mathcal{S}}, \quad R_{\mathcal{F}} \mapsto R_{\mathcal{F}} \circ h.$$

$h^*(R_{\mathcal{F}})$ measures a transition $(s, a, s')$ by evaluating $R_{\mathcal{F}}$ at the transition's abstraction: $R_{\mathcal{F}}(h(s, a, s'))$. Thus, $h^*(R_{\mathcal{F}})$ is guaranteed to give the same reward to transitions with the same abstraction.

We can then also consider the space of abstraction sequences $\mathcal{F}^T$ together with the function $h^T : \Xi \to \mathcal{F}^T$ given by

$$h^T(s_0, a_0, \ldots, s_{T-1}, a_{T-1}, s_T) := \big(h(s_0, a_0, s_1), \ldots, h(s_{T-1}, a_{T-1}, s_T)\big).$$

Write return functions over abstraction sequences as $G_{\mathcal{F}^T} \in \mathbb{R}^{\mathcal{F}^T}$. $h^T$ then gives rise to the dual function

$$h^{T^*} : \mathbb{R}^{\mathcal{F}^T} \to \mathbb{R}^{\Xi}, \quad G_{\mathcal{F}^T} \mapsto G_{\mathcal{F}^T} \circ h^T.$$

Thus, $h^{T^*}(G_{\mathcal{F}^T})$ evaluates a trajectory by evaluating the sequence of abstractions using $G_{\mathcal{F}^T}$. As before, two trajectories with the same sequences of abstractions then obtain the same return.

Recall the function $\mathbf{\Gamma} : \mathbb{R}^{\mathcal{S} \times \mathcal{A} \times \mathcal{S}} \to \mathbb{R}^{\Xi}$ mapping a reward function to the corresponding return function. Then we obtain an analogous function for reward objects on abstractions:

$$\mathbf{\Gamma}_{\mathcal{F}} : \mathbb{R}^{\mathcal{F}} \to \mathbb{R}^{\mathcal{F}^T}, \quad \big[\mathbf{\Gamma}_{\mathcal{F}}(R_{\mathcal{F}})\big](f_1, \ldots, f_T) := \sum_{t=0}^{T-1} \gamma^t R_{\mathcal{F}}(f_t).$$

**Proposition C.1.** *The diagram*

$$
\begin{array}{ccc}
\mathbb{R}^{\mathcal{S} \times \mathcal{A} \times \mathcal{S}} & \xrightarrow{\ \mathbf{\Gamma}\ } \mathbb{R}^{\Xi} \xrightarrow{\ \mathbf{B}\ } \mathbb{R}^{\mathcal{O}} \\
{\scriptstyle h^*}\big\uparrow & \qquad {\scriptstyle h^{T^*}}\big\uparrow \\
\mathbb{R}^{\mathcal{F}} & \xrightarrow[\ \mathbf{\Gamma}_{\mathcal{F}}\ ]{} \mathbb{R}^{\mathcal{F}^T}
\end{array}
\tag{21}
$$

*of linear functions commutes, meaning that all pathways with the same start and end are the same function.*

*Proof.* We have

$$
\begin{aligned}
\big[(\boldsymbol{\Gamma}\circ h^*)(R_\mathcal{F})\big](s_0, a_0, \ldots, a_{T-1}, s_T) &= \Big[\boldsymbol{\Gamma}\big(h^*(R_\mathcal{F})\big)\Big](s_0, a_0, \ldots, a_{T-1}, s_T)\\
&= \sum_{t=0}^{T-1} \gamma^t \big[h^*(R_\mathcal{F})\big](s_t, a_t, s_{t+1})\\
&= \sum_{t=0}^{T-1} \gamma^t R_\mathcal{F}\big(h(s_t, a_t, s_{t+1})\big)\\
&= \big[\boldsymbol{\Gamma}_\mathcal{F}(R_\mathcal{F})\big]\big(h(s_0, a_0, s_1), \ldots, h(s_{T-1}, a_{T-1}, s_T)\big)\\
&= \big[\boldsymbol{\Gamma}_\mathcal{F}(R_\mathcal{F})\big]\big(h^T(s_0, a_0, \ldots, s_{T-1}, a_{T-1}, s_T)\big)\\
&= \Big[h^{T^*}\big(\boldsymbol{\Gamma}_\mathcal{F}(R_\mathcal{F})\big)\Big](s_0, a_0, \ldots, a_{T-1}, s_T)\\
&= \big[(h^{T^*}\circ \boldsymbol{\Gamma}_\mathcal{F})(R_\mathcal{F})\big](s_0, a_0, \ldots, a_{T-1}, s_T).
\end{aligned}
$$

This shows $\boldsymbol{\Gamma}\circ h^* = h^{T^*}\circ\boldsymbol{\Gamma}_\mathcal{F}$. Consequently, the diagram commutes. $\qquad\square$

The idea will be that the rows of $\mathbf{B}$, $\mathbf{B}\circ\boldsymbol{\Gamma}$, $\mathbf{B}\circ h^{T^*}$ and $\mathbf{B}\circ\boldsymbol{\Gamma}\circ h^* = \mathbf{B}\circ h^{T^*}\circ\boldsymbol{\Gamma}_\mathcal{F}$ all correspond (via Proposition A.1) to feature beliefs over trajectories, transitions, trajectory abstractions, and transition abstractions, respectively. We explain this interpretation in detail in Appendix C.3. All of these functions "factorize" over trajectories, but of course this need not be the case in reality: A realistic human could have an intrinsic belief over state transitions, sequences of abstractions, or single abstractions, without this belief "factorizing" in a rational way over state sequences.

Thus, let the following be an extended version of the diagram from Proposition C.1, with new linear functions $\mathbf{B}', \mathbf{B}'', \mathbf{B}'''$. This extension is now *not* necessarily commutative anymore:

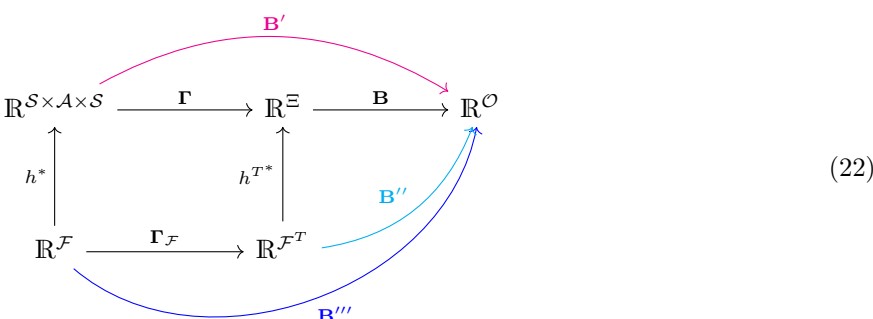

$$(22)$$

To interpret $\mathbf{B}', \mathbf{B}'', \mathbf{B}'''$ on similar grounds as $\mathbf{B}$, it makes sense to assume that they are row-constant (Definition B.1), but otherwise they can be arbitrary.

## C.2 Various human belief models

We take the previous diagrams as the starting point to construct human belief models Remember that a belief model is of the form $\mathcal{M} = (\Omega, \boldsymbol{\Lambda}, \mathcal{E}, \mathcal{V})$. The sets $\Xi, \mathcal{S}\times\mathcal{A}\times\mathcal{S}, \mathcal{F}$, and $\mathcal{F}^T$ are four different possible feature sets $\Omega$. $\boldsymbol{\Lambda}$ is given by a composition of linear functions that maps to $\mathbb{R}^\Xi$. $\mathcal{E}$ is given by a composition mapping to $\mathbb{R}^\mathcal{O}$. The space $\mathcal{V}$ is either given by the full vector space $\mathbb{R}^\Omega$, or by images of functions mapping to $\mathbb{R}^\Omega$. Overall, using the diagram from Proposition C.1, this leads to the following 9 models, with the superscript denoting the feature space, and the subscript indicating where the valid reward objects "originate from":

$$
\begin{aligned}
\mathcal{M}_\mathcal{F}^\mathcal{F} &:= \big(\mathcal{F}, \ \boldsymbol{\Gamma}\circ h^*, \ \mathbf{B}\circ\boldsymbol{\Gamma}\circ h^*, \ \mathbb{R}^\mathcal{F}\big)\\
\mathcal{M}_{\mathcal{S}\times\mathcal{A}\times\mathcal{S}}^{\mathcal{S}\times\mathcal{A}\times\mathcal{S}} &:= \big(\mathcal{S}\times\mathcal{A}\times\mathcal{S}, \ \boldsymbol{\Gamma}, \ \mathbf{B}\circ\boldsymbol{\Gamma}, \ \mathbb{R}^{\mathcal{S}\times\mathcal{A}\times\mathcal{S}}\big)
\end{aligned}
$$

$$\mathcal{M}_{\mathcal{F}}^{\mathcal{S}\times\mathcal{A}\times\mathcal{S}} := \big(\mathcal{S}\times\mathcal{A}\times\mathcal{S},\ \boldsymbol{\Gamma},\ \mathbf{B}\circ\boldsymbol{\Gamma},\ \mathrm{im}(h^*)\big)$$

$$\mathcal{M}_{\mathcal{F}^T}^{\mathcal{F}^T} := \big(\mathcal{F}^T,\ h^{T^*},\ \mathbf{B}\circ h^{T^*},\ \mathbb{R}^{\mathcal{F}^T}\big)$$

$$\mathcal{M}_{\mathcal{F}}^{\mathcal{F}^T} := \big(\mathcal{F}^T,\ h^{T^*},\ \mathbf{B}\circ h^{T^*},\ \mathrm{im}(\boldsymbol{\Gamma}_{\mathcal{F}})\big)$$

$$\mathcal{M}_{\Xi}^{\Xi} := \big(\Xi,\ \mathrm{id}_{\mathbb{R}^{\Xi}},\ \mathbf{B},\ \mathbb{R}^{\Xi}\big)$$

$$\mathcal{M}_{\mathcal{S}\times\mathcal{A}\times\mathcal{S}}^{\Xi} := \big(\Xi,\ \mathrm{id}_{\mathbb{R}^{\Xi}},\ \mathbf{B},\ \mathrm{im}(\boldsymbol{\Gamma})\big)$$

$$\mathcal{M}_{\mathcal{F}^T}^{\Xi} := \big(\Xi,\ \mathrm{id}_{\mathbb{R}^{\Xi}},\ \mathbf{B},\ \mathrm{im}(h^{T^*})\big)$$

$$\mathcal{M}_{\mathcal{F}}^{\Xi} := \big(\Xi,\ \mathrm{id}_{\mathbb{R}^{\Xi}},\ \mathbf{B},\ \mathrm{im}(\boldsymbol{\Gamma}\circ h^*)\big)$$

For example, $\mathcal{M}_{\mathcal{S}\times\mathcal{A}\times\mathcal{S}}^{\mathcal{S}\times\mathcal{A}\times\mathcal{S}}$ is the model from Example 2.6; $\mathcal{M}_{\mathcal{F}}^{\mathcal{S}\times\mathcal{A}\times\mathcal{S}}$ is the same model, but with valid reward functions restricted to those that only "care about" abstractions; $\mathcal{M}_{\Xi}^{\Xi}$ is a model in which the features are given by full trajectories, and there are no restrictions on the valid return functions; etc.

Now, $\mathbf{B}'$ naturally gives rise to the following three models, which we color differently to distinguish them more easily:

$$\mathcal{M}_{\mathcal{F}}'^{\mathcal{F}} = \big(\mathcal{F},\ \boldsymbol{\Gamma}\circ h^*,\ \mathbf{B}'\circ h^*,\ \mathbb{R}^{\mathcal{F}}\big)$$

$$\mathcal{M}_{\mathcal{F}}'^{\mathcal{S}\times\mathcal{A}\times\mathcal{S}} = \big(\mathcal{S}\times\mathcal{A}\times\mathcal{S},\ \boldsymbol{\Gamma},\ \mathbf{B}',\ \mathrm{im}(h^*)\big)$$

$$\mathcal{M}_{\mathcal{S}\times\mathcal{A}\times\mathcal{S}}'^{\mathcal{S}\times\mathcal{A}\times\mathcal{S}} = \big(\mathcal{S}\times\mathcal{A}\times\mathcal{S},\ \boldsymbol{\Gamma},\ \mathbf{B}',\ \mathbb{R}^{\mathcal{S}\times\mathcal{A}\times\mathcal{S}}\big).$$

Similarly, $\mathbf{B}''$ gives rise to the following three models:

$$\mathcal{M}_{\mathcal{F}}''^{\mathcal{F}} = \big(\mathcal{F},\ h^{T^*}\circ\boldsymbol{\Gamma}_{\mathcal{F}},\ \mathbf{B}''\circ\boldsymbol{\Gamma}_{\mathcal{F}},\ \mathbb{R}^{\mathcal{F}}\big)$$

$$\mathcal{M}_{\mathcal{F}}''^{\mathcal{F}^T} = \big(\mathcal{F}^T,\ h^{T^*},\ \mathbf{B}'',\ \mathrm{im}(\boldsymbol{\Gamma}_{\mathcal{F}})\big)$$

$$\mathcal{M}_{\mathcal{F}^T}''^{\mathcal{F}^T} = \big(\mathcal{F}^T,\ h^{T^*},\ \mathbf{B}'',\ \mathbb{R}^{\mathcal{F}^T}\big)$$

Finally, $\mathbf{B}'''$ gives rise to a single model:

$$\mathcal{M}_{\mathcal{F}}'''^{\mathcal{F}} = \big(\mathcal{F},\ \boldsymbol{\Gamma}\circ h^*,\ \mathbf{B}''',\ \mathbb{R}^{\mathcal{F}}\big)$$

Note that all component linear functions appearing in any of these models (identities, $h^*, \boldsymbol{\Gamma}, \boldsymbol{\Gamma}_{\mathcal{F}}, h^{T^*}, \mathbf{B}, \ldots, \mathbf{B}'''$) are row-constant. By Lemma B.3 then, also all compositions are row-constant, which then implies that all 16 models are balanced, as defined in Definition B.2. The first 9 models are also faithful (Definition 2.15) since all feature belief functions factorize as in Proposition 2.16, with $Y$ given by $\mathbf{B}$ in all cases. The other 7 models will typically not be faithful.

### C.3   Interpreting the matrix elements

We now interpret the different feature belief functions that appeared in the nine first models of the previous subsection. Recall that the linear function $\mathbf{B} : \mathbb{R}^{\Xi} \to \mathbb{R}^{\mathcal{O}}$ "comes from" a function $b : \mathcal{O} \to \Delta(\Xi) \subseteq \mathbb{R}^{\Xi}$. Thus, all matrix elements $\mathbf{B}_{o\xi}$ can be interpreted as a probability $\big[b(o)\big](\xi)$ for the trajectory $\xi$ when viewing observation $o$. We now explain similar interpretations for the matrix elements of all the other feature belief functions:

$\mathbf{B}\circ\boldsymbol{\Gamma}$: It contains matrix elements

$$(\mathbf{B}\circ\boldsymbol{\Gamma})_{o,(s,a,s')} = \sum_{\xi} \big[b(o)\big](\xi) \sum_{t=0}^{T-1} \gamma^t \delta_{(s,a,s')}(s_t, a_t, s_{t+1}),$$

the expected discounted number of times the transition $(s, a, s')$ is present in the trajectory.

$\mathbf{B} \circ h^{T^*}$: Write $\mathbf{f}$ for $(f_1, \ldots, f_T)$. Then this contains matrix elements

$$
\begin{aligned}
(\mathbf{B} \circ h^{T^*})_{o\mathbf{f}} &= \sum_{\xi \in \Xi} \big[b(o)\big](\xi) h_{\xi\mathbf{f}}^{T^*} \\
&= \sum_{\xi \in \Xi} \big[b(o)\big](\xi) \delta_{\mathbf{f}}(h^T(\xi)) \\
&= \sum_{\xi \,:\, h^T(\xi)=\mathbf{f}} \big[b(o)\big](\xi) \\
&= \big[b(o)\big]\big((h^T)^{-1}(\mathbf{f})\big) \\
&= \big[b(o)_{h^T}\big](\mathbf{f}).
\end{aligned}
$$

In the second to last step, we view $b(o)$ as a probability distribution that, when evaluated on a set, evaluates to the sum of the probabilities of the set's elements. In the last step, we use the definition of the distributional law of a random variable $X$ with respect to a probability distribution $P$ on the sample space: $P_X(x) = P(X^{-1}(x))$. The result is the believed probability, after observing $o$, of a trajectory with sequence of abstractions $\mathbf{f}$.

Finally, we look at the matrix $\mathbf{B} \circ \mathbf{\Gamma} \circ h^*$ (for which we give two slightly different formulas): The matrix elements are given as

$$
\begin{aligned}
(\mathbf{B} \circ \mathbf{\Gamma} \circ h^*)_{of} &= \sum_{(s,a,s') \in \mathcal{S} \times \mathcal{A} \times \mathcal{S}} (\mathbf{B} \circ \mathbf{\Gamma})_{o,(s,a,s')} \cdot h^*_{(s,a,s'),f} \\
&= \sum_{(s,a,s')\,:\, h(s,a,s')=f} \sum_{\xi \in \Xi} \big[b(o)\big](\xi) \sum_{t=0}^{T-1} \gamma^t \delta_{(s,a,s')}(s_t, a_t, s_{t+1}) \\
&= \sum_{\xi \in \Xi} \big[b(o)\big](\xi) \sum_{t=0}^{T-1} \gamma^t \sum_{(s,a,s')\,:\, h(s,a,s')=f} \delta_{(s,a,s')}(s_t, a_t, s_{t+1}) \\
&= \sum_{\xi \in \Xi} \big[b(o)\big](\xi) \sum_{t=0}^{T-1} \gamma^t \delta_f(h(s_t, a_t, s_{t+1})).
\end{aligned}
$$

This is the expected discounted number of times that one encounters the abstraction $f$. Using that $\mathbf{\Gamma} \circ h^* = h^{T^*} \circ \mathbf{\Gamma}_{\mathcal{F}}$ by Proposition C.1, we can also write this as

$$
\begin{aligned}
\big(\mathbf{B} \circ h^{T^*} \circ \mathbf{\Gamma}_{\mathcal{F}}\big)_{of} &= \sum_{\mathbf{f} \in \mathcal{F}^T} (\mathbf{B} \circ h^{T^*})_{o\mathbf{f}} \cdot (\mathbf{\Gamma}_{\mathcal{F}})_{\mathbf{f}f} \\
&= \sum_{\mathbf{f} \in \mathcal{F}^T} \big[b(o)_{h^T}\big](\mathbf{f}) \sum_{t=0}^{T} \gamma^t \delta_f(f_t).
\end{aligned}
$$

This can also be described as the expected discounted number of times that one encounters the abstraction $f$.

### C.4  The resulting commutative diagram

Building on Appendix C.2, the following is a commutative diagram of belief models and model morphisms, with four differently colored "connected components"; one can sometimes use Proposition C.1 in the process of showing that every linear function in the diagram is a morphism of belief models, and that the final diagram commutes:

$$
\begin{array}{ccccc}
\mathcal{M}'^{\mathcal{S}\times\mathcal{A}\times\mathcal{S}}_{\mathcal{S}\times\mathcal{A}\times\mathcal{S}} & \mathcal{M}^{\mathcal{S}\times\mathcal{A}\times\mathcal{S}}_{\mathcal{S}\times\mathcal{A}\times\mathcal{S}} \xrightarrow{\ \boldsymbol{\Gamma}\ } \mathcal{M}^{\Xi}_{\mathcal{S}\times\mathcal{A}\times\mathcal{S}} \xrightarrow{\ \mathrm{id}_{\mathbb{R}^{\Xi}}\ } \mathcal{M}^{\Xi}_{\Xi} \\
{\scriptstyle\mathrm{id}_{\mathbb{R}^{\mathcal{S}\times\mathcal{A}\times\mathcal{S}}}}\uparrow & {\scriptstyle\mathrm{id}_{\mathbb{R}^{\mathcal{S}\times\mathcal{A}\times\mathcal{S}}}}\uparrow \qquad {\scriptstyle\mathrm{id}_{\mathbb{R}^{\Xi}}}\uparrow \qquad {\scriptstyle\mathrm{id}_{\mathbb{R}^{\Xi}}}\uparrow \\
\mathcal{M}'^{\mathcal{S}\times\mathcal{A}\times\mathcal{S}}_{\mathcal{F}} & \mathcal{M}^{\mathcal{S}\times\mathcal{A}\times\mathcal{S}}_{\mathcal{F}} \xrightarrow{\ \boldsymbol{\Gamma}\ } \mathcal{M}^{\Xi}_{\mathcal{F}} \xrightarrow{\ \mathrm{id}_{\mathbb{R}^{\Xi}}\ } \mathcal{M}^{\Xi}_{\mathcal{F}^{T}} \\
{\scriptstyle h^{*}}\uparrow & {\scriptstyle h^{*}}\uparrow \qquad {\scriptstyle h^{T^{*}}}\uparrow \qquad {\scriptstyle h^{T^{*}}}\uparrow \\
\mathcal{M}'^{\mathcal{F}}_{\mathcal{F}} & \mathcal{M}^{\mathcal{F}}_{\mathcal{F}} \xrightarrow{\ \boldsymbol{\Gamma}_{\mathcal{F}}\ } \mathcal{M}^{\mathcal{F}^{T}}_{\mathcal{F}} \xrightarrow{\ \mathrm{id}_{\mathbb{R}^{\mathcal{F}^{T}}}\ } \mathcal{M}^{\mathcal{F}^{T}}_{\mathcal{F}^{T}}
\end{array}
$$

$$
\mathcal{M}'''^{\mathcal{F}}_{\mathcal{F}} \qquad \mathcal{M}''^{\mathcal{F}}_{\mathcal{F}} \xrightarrow{\ \boldsymbol{\Gamma}_{\mathcal{F}}\ } \mathcal{M}''^{\mathcal{F}^{T}}_{\mathcal{F}} \xrightarrow{\ \mathrm{id}_{\mathbb{R}^{\mathcal{F}^{T}}}\ } \mathcal{M}''^{\mathcal{F}^{T}}_{\mathcal{F}^{T}}
$$

For example, the following diagram visualizes the fact that $h^{*}:\mathcal{M}^{\mathcal{F}}_{\mathcal{F}}\to\mathcal{M}^{\mathcal{S}\times\mathcal{A}\times\mathcal{S}}_{\mathcal{F}}$ is a morphism:

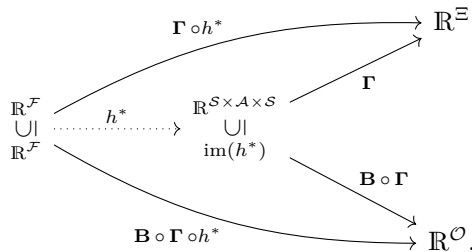

This gives rise to the following diagram of ambiguities:

$$
\begin{array}{ccccc}
\boldsymbol{\Gamma}(\ker(\mathbf{B}')) & \boldsymbol{\Gamma}\big(\ker(\mathbf{B}\circ\boldsymbol{\Gamma})\big) & = & \ker(\mathbf{B})\cap\mathrm{im}(\boldsymbol{\Gamma}) & \subseteq & \ker(\mathbf{B}) \\
\cup| & \cup| & & \cup| & & \cup| \\
\boldsymbol{\Gamma}\big(\ker(\mathbf{B}')\cap\mathrm{im}(h^{*})\big) & \boldsymbol{\Gamma}\big(\ker(\mathbf{B}\circ\boldsymbol{\Gamma})\cap\mathrm{im}(h^{*})\big) & = & \ker(\mathbf{B})\cap\mathrm{im}(\boldsymbol{\Gamma}\circ h^{*}) & \subseteq & \ker(\mathbf{B})\cap\mathrm{im}\big(h^{T^{*}}\big) \\
\| & \| & & \| & & \| \\
(\boldsymbol{\Gamma}\circ h^{*})\big(\ker(\mathbf{B}'\circ h^{*})\big) & (\boldsymbol{\Gamma}\circ h^{*})\big(\ker(\mathbf{B}\circ\boldsymbol{\Gamma}\circ h^{*})\big) & = & h^{T^{*}}\big(\ker(\mathbf{B}\circ h^{T^{*}})\cap\mathrm{im}(\boldsymbol{\Gamma}_{\mathcal{F}})\big) & \subseteq & h^{T^{*}}\big(\ker(\mathbf{B}\circ h^{T^{*}})\big)
\end{array}
$$

$$
(\boldsymbol{\Gamma}\circ h^{*})\big(\ker(\mathbf{B}''')\big) \qquad \big(h^{T^{*}}\circ\boldsymbol{\Gamma}_{\mathcal{F}}\big)\big(\ker(\mathbf{B}''\circ\boldsymbol{\Gamma}_{\mathcal{F}})\big) \ = \ h^{T^{*}}\big(\ker(\mathbf{B}'')\cap\mathrm{im}(\boldsymbol{\Gamma}_{\mathcal{F}})\big) \ \subseteq \ h^{T^{*}}\big(\ker(\mathbf{B}'')\big)
$$

The ambiguities are computed using Proposition 2.9, and the inclusions and equalities of ambiguities follow from Theorem 3.2 and Theorem 3.5. Here, the ambiguity $\boldsymbol{\Gamma}\big(\ker(\mathbf{B}\circ\boldsymbol{\Gamma})\big)=\ker(\mathbf{B})\cap\mathrm{im}(\boldsymbol{\Gamma})$ is the special case discussed in depth in Lang et al. (2024). Note that the models $\mathcal{M}^{\mathcal{F}}_{\mathcal{F}}, \mathcal{M}^{\mathcal{S}\times\mathcal{A}\times\mathcal{S}}_{\mathcal{F}}$ and $\mathcal{M}^{\mathcal{S}\times\mathcal{A}\times\mathcal{S}}_{\mathcal{S}\times\mathcal{A}\times\mathcal{S}}$ are closely related to the models $\mathcal{M}_1,\mathcal{M}_2$ and $\mathcal{M}_3$ from Section 3.3.

Assume we would use one of these models in practice. The further right or up it is in the diagram, the more ambiguity there is, but it is then also more likely that the model covers the true belief model (should it appear in the diagram in the first place). Thus, there is a trade-off between covering the true belief model, and keeping the ambiguity small.

## D    Details on the example with invariant features

Here, we present more mathematical details for Section 3.3. This appendix is not self-contained and we recommend reading it alongside the section in the main paper.

### D.1 Details on the MDP and observations

Formally, the states are given by $\mathcal{S} = (\{L, R\} \times \{U, D\})^2$, with the first component being the hand-position, and the second component being the button position. For example, the state in Equation (12) is given by $((L, U), (R, D))$.

Furthermore, we define functions $\mathrm{Pos}_H : \mathcal{S} \to \{L, R\} \times \{U, D\}$ and $\mathrm{Pos}_B : \mathcal{S} \to \{L, R\} \times \{U, D\}$ as the first and second projection. These are the position of a "hand" $H$ and a "button" $B$, in a 2x2 gridworld. Then the state $s$ from Equation (12) satisfies $\mathrm{Pos}_H(s) = (L, U)$ and $\mathrm{Pos}_B(s) = (R, D)$.

The set of trajectories is formally given by $\Xi = (\mathcal{S} \times \mathcal{A})^3 \times \mathcal{S}$. The set of observations is formally given by

$$\mathcal{O} = \left[ \{L, R\}^2 \times \{P, \overline{P}\} \right]^3 \times \{L, R\}^2.$$

Here, $\overline{P}$ means that it was *not* observed that a button was pressed.

### D.2 Details on the belief models and symmetries

Let $G = D_4$ be the dihedral group of order 8, i.e., the symmetry group of the square. It is given by

$$G = D_4 = \left\{ e, \ r, \ r^2, \ r^3, \ f, \ rf, \ r^2 f, \ r^3 f \right\},$$

where $r$ is a clockwise rotations by $90°$ and $f$ is a flip over the horizontal axis. In compositions, we apply $f$ first. $G$ acts on $\mathcal{S} \times \mathcal{A}$ by individually acting on states and actions:

$$g.(s, a) := (g.s, g.a),$$

where $g.s = g.(\mathrm{Pos}_H(s), \mathrm{Pos}_B(s)) := (g. \mathrm{Pos}_H(s), g. \mathrm{Pos}_B(s))$, where on the generators $g = r$ and $g = f$ we have

$$r.(L, U) = (R, U), \quad r.(R, U) = (R, D), \quad r.(R, D) = (L, D), \quad r.(L, D) = (L, U)$$
$$f.(L, U) = (L, D), \quad f.(R, U) = (R, D), \quad f.(R, D) = (R, U), \quad f.(L, D) = (L, U).$$

This specifies the action on states. On actions, we specify

$$r.L = U, \quad r.U = R, \quad r.R = D, \quad r.D = L, \quad r.P = P.$$
$$f.L = L, \quad f.U = D, \quad f.R = R, \quad f.D = U, \quad f.P = P.$$

Thus, the "pressing" action remains invariant.

With this group action, we obtain a set of equivalence classes of state-action pairs, given by $\overline{\mathcal{S} \times \mathcal{A}}$. A set of *representatives* for the equivalence classes is given by

$$\left( \{s_0\} \times \mathcal{A}^{s_0} \right) \cup \left( \{s_1\} \times \mathcal{A}^{s_1} \right) \cup \left( \{s_2\} \times \mathcal{A}^{s_2} \right), \tag{23}$$

where

$$s_0 = ((R, D), (R, D)), \quad s_1 = ((L, D), (R, D)), \quad s_2 = ((L, U), (R, D)),$$

and where the (state-dependent) set of actions are given by

$$\mathcal{A}^{s_0} = \mathcal{A}^{s_2} = \{L, D, P\}, \quad \mathcal{A}^{s_1} = \{L, R, U, D, P\}.$$

We then have a function

$$h : \mathcal{S} \times \mathcal{A} \to \bigcup_{i \in \{0,1,2\}} \{s_i\} \times \mathcal{A}^{s_i}$$

that maps each state-action pair to a representative, given by $h(s, a) = g.(s, a)$ for the unique $g \in D_4$ for which $g.(s, a)$ is in the set of representatives from Equation (23). Via $h$, we now identify $\overline{\mathcal{S} \times \mathcal{A}}$ with $\bigcup_{i \in \{0,1,2\}} \{s_i\} \times \mathcal{A}^{s_i}$.

Equation (15) can be showed by

$$
\begin{aligned}
\mathbf{\Lambda}_{\xi,(s,a)} &= \big[\lambda(\xi)\big](s,a) \\
&= \sum_{t=0}^{2} \delta_{(s,a)}(h(s_t,a_t)) \\
&= \sum_{(s',a')\in\mathcal{S}\times\mathcal{A}} \delta_{(s,a)}(h(s',a')) \sum_{t=0}^{2} \delta_{(s',a')}(s_t,a_t) \\
&= \sum_{(s',a')\in\mathcal{S}\times\mathcal{A}} h^{*}_{(s',a'),(s,a)} \, \mathbf{\Gamma}_{\xi,(s',a')} \\
&= \sum_{(s',a')\in\mathcal{S}\times\mathcal{A}} \mathbf{\Gamma}_{\xi,(s',a')} \, h^{*}_{(s',a'),(s,a)} \\
&= (\mathbf{\Gamma}\circ h^{*})_{\xi,(s,a)}.
\end{aligned}
\tag{24}
$$

For the matrix elements of $h^{*}$, we used Equation (1).

The human's belief $\mathcal{E}(s,a \mid o)$ for $(s,a) \in \overline{\mathcal{S}\times\mathcal{A}}$ and $o \in \mathcal{O}$ is then given as follows: The human has a uniform prior $B(s) = P_0$ over possible start-states sampled from $P_0$, and a uniform prior over possibly next actions given the current state, leading to a prior distribution over $B(\xi) \in \Delta(\Xi)$. Then, upon seeing $o$, the human implicitly computes a posterior belief over trajectories compatible with the observation, simply given by

$$
B(\xi \mid o) \propto \delta_o(O(\xi)) \cdot B(\xi).
\tag{25}
$$

Equation (16) can be showed by

$$
\begin{aligned}
\mathcal{E}_{o,(s,a)} &= \big[\epsilon(o)\big](s,a) \\
&= \sum_{\xi\in\Xi} \big[b(o)\big](\xi) \cdot \big[\lambda(\xi)\big](s,a) \\
&= \sum_{\xi\in\Xi} \mathbf{B}_{o\xi} \cdot \mathbf{\Lambda}_{\xi,(s,a)} \\
&= (\mathbf{B}\circ\mathbf{\Lambda})_{o,(s,a)} \\
&= (\mathbf{B}\circ\mathbf{\Gamma}\circ h^{*})_{o,(s,a)}
\end{aligned}
\tag{26}
$$

### D.3 Details on the ambiguity analysis for $\mathcal{M}_2$

In all these computations, recall Equation (6) and the sentence following it for computing $\mathcal{E}(\overline{R})(o)$ for an observation $o$.

Recall the observation $o_2$:

$$
o_2 = \boxed{\text{H} \mid \text{B}} \xrightarrow{\text{P}} \boxed{\text{H} \mid \text{B}} \xrightarrow{\text{P}} \boxed{\text{H} \mid \text{B}} \xrightarrow{\text{P}} \boxed{\text{H} \mid \text{B}}
$$

Since we assumed that the starting state is one of the states in Equation (13), the human has the belief $\big[\epsilon(o_2)\big](s_2,P) = 3$, i.e., the human is certain that, up to symmetry, the hand performed a pressing action three times in $s_2$. Thus,

$$
0 = \big[\mathcal{E}(\overline{R})\big](o_2) = \big[\epsilon(o_2)\big](s_2,P) \cdot \overline{R}(s_2,P) = 3 \cdot \overline{R}(s_2,P)
$$

This implies $\overline{R}(s_2,P) = 0$.

Recall observation $o_1$:

$$
o_1 = \boxed{\text{H} \mid \text{B}} \longrightarrow \boxed{\text{H} \mid \text{B}} \xrightarrow{\text{P}} \boxed{\text{H} \mid \text{B}} \xrightarrow{\text{P}} \boxed{\text{H} \mid \text{B}}
$$

Now, the first action could either not change anything, *or* horizontally align $H$ and $B$. An action that does not change anything is more likely (chance $2/3$ since there are two actions, in the direction of two different adjacent walls, that achieve this, which both correspond to action $L$ up to symmetry), and so we obtain

$$[\epsilon(o_1)](s_2, L) = 2/3, \quad [\epsilon(o_1)](s_2, P) = 4/3,$$
$$[\epsilon(o_1)](s_2, D) = 1/3, \quad [\epsilon(o_1)](s_1, P) = 2/3.$$

Compare also with (26). Thus, we obtain

$$
\begin{aligned}
0 &= [\mathcal{E}(\overline{R})](o_1) \\
&= 2/3 \cdot \overline{R}(s_2, L) + 4/3 \cdot \overline{R}(s_2, P) + 1/3 \cdot \overline{R}(s_2, D) + 2/3 \cdot \overline{R}(s_1, P) \\
&= 2/3 \cdot \overline{R}(s_1, P).
\end{aligned}
$$

Here, we used that $\overline{R}(s_2, P) = 0$ by what we showed before, and $\overline{R}(s_2, L) = \overline{R}(s_2, D) = 0$ since $\overline{R}(s, a) = 0$ whenever $a \neq P$ (i.e., since $\overline{R} \in \mathcal{V}$). Thus, we have $\overline{R}(s_1, P) = 0$ as well.

Finally, we look at the observation sequence $o_0$ given as follows:

Again, there is a chance of $2/3$ that the first action does not change anything. Given the first step, everything which follows is deterministic, leading to these feature beliefs:

$$[\epsilon(o_0)](s_2, L) = 2/3, \quad [\epsilon(o_0)](s_2, R) = 2/3, \quad [\epsilon(o_0)](s_1, P) = 2/3$$
$$[\epsilon(o_0)](s_2, D) = 1/3, \quad [\epsilon(o_0)](s_1, R) = 1/3, \quad [\epsilon(o_0)](s_0, P) = 1/3.$$

Compare again with (26). This means that

$$
\begin{aligned}
0 &= [\mathcal{E}(\overline{R})](o_0) \\
&= 2/3 \cdot \overline{R}(s_2, L) + 2/3 \cdot \overline{R}(s_2, R) + 2/3 \cdot \overline{R}(s_1, P) + \\
&\quad + 1/3 \cdot \overline{R}(s_2, D) + 1/3 \cdot \overline{R}(s_1, R) + 1/3 \cdot \overline{R}(s_0, P) \\
&= 1/3 \cdot \overline{R}(s_0, P).
\end{aligned}
$$

Here, we used that $\overline{R}(s_1, P)$ by what we showed before, together, again, with the fact that $\overline{R}(s, a) = 0$ for all $a \neq P$. That shows $\overline{R}(s_0, P) = 0$.

### D.4 Details on the ambiguity analysis for $\mathcal{M}_3$

We have

$$
\begin{aligned}
[(\mathbf{B} \circ \mathbf{\Gamma})(R')](o) &= (\mathbf{B} \circ \mathbf{\Gamma})_{o, (s_1', P)} \cdot R'(s_1', P) + (\mathbf{B} \circ \mathbf{\Gamma})_{o, (s_1'', P)} \cdot R'(s_1'', P) \\
&= (\mathbf{B} \circ \mathbf{\Gamma})_{o, (s_1', P)} - (\mathbf{B} \circ \mathbf{\Gamma})_{o, (s_1'', P)} \\
&= 0
\end{aligned}
\tag{27}
$$

In the computation, the second step follows from the definition of $R'$. The last step follows from the symmetry remarked on before.

## E Mathematical interpretations of related work in our framework

In this appendix, we briefly interpret some of the related work from Section 4.2 in our framework for the special case that they learn linear reward probes. Note that these interpretations are not meant to

capture everything there is to say about that work — the summaries we provide are quite coarse. In all examples below, we assume access to a very capable foundation model $\widehat{\lambda} : \Xi \to \mathbb{R}^{\widehat{\Omega}}$ that allows for a linear ontology translation $\Psi : \mathbb{R}^{\widehat{\Omega}} \to \mathbb{R}^{\Omega}$ to the human's ontology $\lambda$, as in Section 3.4.1: $\Psi \circ \widehat{\lambda} = \lambda$. Define $\Phi := \Psi^T : \mathbb{R}^{\Omega} \to \mathbb{R}^{\widehat{\Omega}}$, which then satisfies $\widehat{\Lambda} \circ \Phi = \Lambda$ by Proposition A.2. In all approaches below, we define $\widehat{\epsilon}$ and assume that the return function is learned with the same method as in Section 3.4.2. Notably, in all of the approaches one essentially just defines $\widehat{\epsilon} := \widehat{\lambda}$, i.e., no explicit modeling of humans is performed.

### E.1 Amplified oversight and eliciting latent knowledge

In amplified oversight, one *amplifies* the human to give accurate feedback, which means we can assume $\mathcal{O} = \Xi$ and $\epsilon = \lambda$. One approach to achieve this would be to essentially *define* $\epsilon := \Psi \circ \widehat{\lambda}$ by giving the human access to the linear ontology translation $\Psi$ for understanding the foundation model's thoughts. This would roughly be in the spirit of eliciting latent knowledge (Christiano et al., 2021), where the human can query a reporter to give information about arbitrary latent knowledge of an AI.

Accordingly, one can also choose $\widehat{\epsilon} = \widehat{\lambda}$, leading to the following coverage diagram:

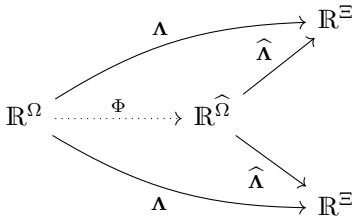

The ontologies and feature belief functions are then the same, which automatically means that the ambiguity disappears: $\Lambda\big(\ker(\Lambda)\big) = 0$.

### E.2 Easy-to-hard generalization

In this setting, $\mathcal{O} \subseteq \Xi$ is a *subset* of trajectories that the human correctly understands. Thus, for $\xi \in \mathcal{O}$, one has $\epsilon(\xi) = \lambda(\xi)$, and so $\epsilon = \lambda|_{\mathcal{O}}$ is simply a restriction. In this setting, one can also set $\widehat{\epsilon} = \widehat{\lambda}|_{\mathcal{O}}$. Now, let $\Lambda|$ and $\widehat{\Lambda}|$ be the linear functions corresponding to $\lambda|_{\mathcal{O}}$ and $\widehat{\lambda}|_{\mathcal{O}}$, respectively, via Proposition A.1. One obtains the following diagram:

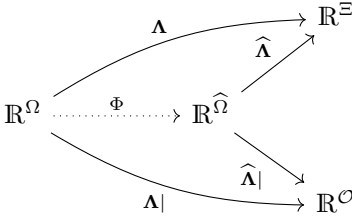

For $\Phi$ in this diagram to be a morphism, we need that the lower diagram commutes. With Proposition A.2, this follows from the assumption that $\Phi^T = \Psi$ is an ontology translation: $\Psi \circ \widehat{\lambda} = \lambda$ implies $\Psi \circ \widehat{\lambda}|_{\mathcal{O}} = \lambda|_{\mathcal{O}}$. The ambiguity is now given by $\widehat{\Lambda}\big(\ker(\widehat{\Lambda}|)\big)$. By using Theorem 2.13, this ambiguity disappears if and only if for all $\xi \in \Xi$, we have $\widehat{\lambda}(\xi) \in \left\{ \sum_{\xi \in \Xi} Z_\xi \widehat{\lambda}(\xi) \mid Z_\xi \in \mathbb{R} \right\}$. Thus, the ambiguity vanishes if the trajectories that the human understands have enough variety in the vector space of feature strengths.

### E.3 Classical RLHF and weak-to-strong generalization

In classical RLHF, without any safeguards, one just uses the model $\widehat{\epsilon} = \widehat{\lambda}$ as the feature belief function even though $\epsilon \neq \lambda$ and hopes for the best:

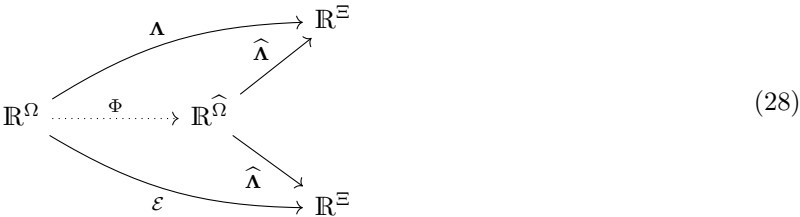

$$(28)$$

In this case, the lower triangle does not commute since $\widehat{\boldsymbol{\Lambda}} \circ \Phi \neq \mathcal{E}$, which, using Proposition A.2, is due to $\Psi \circ \widehat{\lambda} = \lambda \neq \epsilon$. This means that the second model does not cover the true belief model, and so the guarantee from Theorem 3.2 breaks. In fact, the return function that would be inferred using this model is $G_{\mathcal{O}}$, the observation return function itself (cf. the definition of feedback-compatible return functions, Definition 2.8, applied to this faulty model). Lang et al. (2024) extensively discuss failure modes in this case, called deceptive inflation and overjustification. We note that weak-to-strong generalization (Burns et al., 2023a), when used without additional techniques, also considers this setting, but tries to ensure that the learning process or $\widehat{\lambda}$ contains inductive biases that steer the learning process to learn $G$ anyway.

