# OpenReview forum: "Modeling Human Beliefs about AI Behavior for Scalable Oversight"
_TMLR — Accepted by TMLR_

### Review · Reviewer_YrGE · 2025-04-15

**Summary Of Contributions:**

This paper address the problem so-called "scalable oversight" on how to supervise AI systems that is more intelligent than humans. This paper propose to model the human evaluator’s beliefs about how AI system behave, to better interpret the human’s feedback. The authors formalize human belief models and theoretically analyze their role in inferring human values, while discussing the inference ambiguity and its disappearing conditions. Finally, the authors propose using foundation models to construct covering belief models, providing a new potential approach to scalable oversight.

**Audience:**

Yes

**Broader Impact Concerns:**

This paper is purely theoretical, so I do not see any ethical concern in this paper.

**Claims And Evidence:**

No

**Requested Changes:**

- First, I would like to ask the authors to write this paper in a more reader-friendly manner while referring to related studies. For example, I am wondering whether it is possible to frame this paper in the context of the game theory. On the other hand, if this paper is truly novel, it would be impossible to do so. For this point, I would like to refer to the comments by other reviewers or Action Editor. To be honest, I am not fully confident about the novelty of this paper.
- Second, I recommend that the authors include some empirical proof-of-concept experiments to demonstrate the practical usefulness of their approach. Without it, I feel it is difficult for the readers to understand the true value of this paper. Given that this paper is highly theoretical, I don't think the authors need to conduct extensive experiments or benchmarking.

**Strengths And Weaknesses:**

To be honest, I feel it is very difficult to assess this paper. I guess the reason would be either 1) this paper is really innovative or 2) this paper is not well-written and hard to understand. But, let me summarize the Strengths and Weaknesses.

### Strength
- This paper seems to pose some novel ideas for discussing scalable oversights. Since AI safety has recently been a very important topics in our society, this paper can be a good anchor for the future research and development.
- If I understand correctly, the theoretical analyses looks novel. The proofs are well-written (I didn't check all the proofs, though).

### Weakness
- Introduction is well-written and fun to read, but there is a large distance between what is discussed in the introduction and the essential contributions in this paper. I am wondering whether it is really impossible to clarify the contributions of this paper step-by-step based on more closely-related existing literature. After all, after Section 2.5, there is effectively almost no citations. It is fine if the written content is really new and there is literally no related work, but I do not think so and this simply makes this paper hard to understand.  For example, I think this paper would be easier to read if the authors wrote this paper while referring to literature on POMDPs or Markov games, rather than formulating a completely new problem. Note that POMDP or Markov games are just examples. What I want to argue is that this paper will be much easier to understand if well-known notions that are closely related to this paper are properly referred after Section 2.5.
- The authors seem to suppose that every reader fully understand Lang et al. (2024). I read through Lang et al. (2024) for reviewing this submitted paper, but I feel there are still leaps of logic in many parts of this paper. I would like the authors to properly introduce the related work as preliminary and write this paper in a more reader friendly manner.
- My biggest concern regarding this paper is that it is unclear how to use the key idea or knowledge for scalable oversight. This may simply because I cannot fully understand this paper, but at least I think I can understand the main content of this paper. The current paper is too vague as a whole including theoretical analyses and examples, the reader cannot obtain useful knowledge from this paper. After all, while I don't think extensive experiments are necessary, I believe that conducting some experiments in a setting close to real-world applications (using small LLMs or API) would significantly enhance the reader's understanding of the true value of this paper.

---
- Lang, Leon, et al. "When your AIs deceive you: Challenges of partial observability in reinforcement learning from human feedback." Advances in Neural Information Processing Systems 37 (2024): 93240-93299.

---

> ### Author Response · Authors · 2025-06-16
> **Rebuttal 1 of 3**
>
> Thank you for your thoughtful review! We will **highlight in bold changes we plan to make on the paper to address your points.** Please let us know whether these proposed changes would satisfy you.
>
> >  After Section 2.5, there is effectively almost no citations.
>
> We agree that the main content of our paper comes with only a few citations after Section 2.5, which we think stems from our work actually being novel. However, note that we do provide a more extensive related work section, Section 4.2, which places our work into a broader context.
> > For example, I think this paper would be easier to read if the authors wrote this paper while referring to literature on POMDPs or Markov games, rather than formulating a completely new problem.
>
> *On POMDPs*: While there is some similarity between our paper and work on POMDPs, we think the settings are substantially different:
> - In POMDPs, the agent partially observes the environment. We, however, work in an MDP setting in which the agent *fully* observes the environment, see Section 2.2, where we clarify that the policy $\pi: S \to \Delta(A)$ has entire environment states as inputs. This setting applies to the entire paper.
> - In Section 2.4, we state that the human evaluator observes observations $o \in O$ to evaluate the AI’s behavior. While partial observability is an important aspect of many of our examples, note that our setting also applies to the case that the human *fully* observes the environment and simply does *not understand* the agent’s behavior fully. To this end, we write on page 7: “Even if $O = \Xi$, we can have  $\epsilon \neq \lambda$, which reflects that the human may believe the features of a trajectory to be different from what they actually are.”
> - In POMDP solvers [6], the goal is sometimes to find efficient methods for the agent to use its observations to update a belief about the environment state. In our work, we take the human’s belief function as a given and study *how to model* it instead of *improving it*, which is substantially different in emphasis.
>
> **We propose to include some classical citations in Section 2.4 (including [6]) that clearly highlight the distinction between our framework and the POMDP framework. Additionally, we will include citations in the introduction on other work that talks about formalizations of beliefs, including I-POMDPs [11]. This will help the reader to contrast our work from prior work.**
>
> *On Markov Games*: To a first approximation, our work is not in a game setting: We are trying to make progress on a single-agent MDP setting by first learning a return function and then optimizing a policy against that return function. However, there is a connection to Markov games that is implicitly already acknowledged in the paper. Here we spell out that relationship:
> - Our human gives feedback, either in the form of returns of observations (main paper), or in the form of choices between observations (Appendix B)
> - Giving choices between observations is a special case of reward-rational (implicit) choice [1].
> - Reward-rational (implicit) choice was proven to be a special case of assistance games [2], which are a slight reframing of cooperative inverse reinforcement learning, which is a “two-player Markov game with identical payoffs between a human or principal, H, and a robot or agent, R” [3].
> - Assistance games were in a concurrent preprint generalized to a setting with partial observability [4].
>
> Note that we already cite all four of these works in the first paragraph of the related work section (Section 4.2, page 26). We mention [4] a second time in the last paragraph of our future theory work section (Section 4.3.1, page 28).
>
> This leaves us to explain why we didn’t build our work more closely on the foundation of such work, especially on the most closely related concurrent preprint [4], which is a Markov game setting. In [4], an agent and a human interact in a partially observable environment and try to act optimally to achieve the human’s values, which are unobserved to the agent. However, *the human is assumed to have calibrated beliefs*, whereas we assume *arbitrary beliefs that may also be irrational*. Furthermore, their work does not directly study the question of how the agent can infer the human’s values from the human’s actions, and instead analyzes observation interference under the assumption that the agent’s policy is an optimal response policy to the human. We do think it would be interesting to generalize our setting to the assistance game framework in [4], but we cannot directly build on their work as it is currently framed.
>
> **We propose to lengthen the discussion of [4] to more clearly establish how they’re studying a different question from ours. Additionally, I will mention other work like I-POMDPs [11] (which are a generalization of assistance games that also seem to assume rationality in other agent’s beliefs) to provide a more complete picture of related works.**

---

> > ### Author Response · Authors · 2025-06-16
> > **Rebuttal 2 of 3**
> >
> > > The authors seem to suppose that every reader fully understands Lang et al. (2024).
> >
> > This was not our aim, but we appreciate the feedback. We do think that the technical contribution of our work can be understood fully independently of Lang et al. (2024), but we acknowledge that part of the *motivation* for our work may depend on a good understanding of that paper (which actually ties into your question on scalable oversight that we address further below).
> >
> > To make it fully clear that our technical contribution can be understood independently from Lang et al. (2024), **we propose the following changes at all places where we refer to concrete results from that paper:**
> > - Example 2.4: In the current version, we write “We now explain the belief model from Lang et al. (2024) in our framework, thus showing that we generalize their work.” This incorrectly makes the impression that Lang et al. (2024) needs to be understood to understand Ex. 2.4, which is not the case: The entire computation is developed independently. **We will make this clear by writing that the result in the example can be compared with Lang et al. (2024) to check consistency with their framework.**
> > - Example 2.9: Currently, we write “Lang et al. (2024) contain characterizations of this ambiguity in examples and special cases.”: **We will make clear that it is not necessary to understand those examples and characterizations to read the rest of our paper.** They are just a pointer for anyone who is interested. We have many examples (Section 2.8, Section 3.3, Appendix C) that are developed fully independently.
> > - Start of Section 2.8: Currently we write “We refer to Section 3.3 and [5] for concrete examples where the MDP is defined entirely and a possibly remaining ambiguity stems from partial observability.” **We will adapt this to make clear that this just provides further examples for interested readers (whereas our examples are self-contained and can be independently understood).**
> > - In Section 4.3.1, we write “Future work could develop an approximate theory, akin to Lang et al. (2024, Theorem 5.4).” **We will clarify in a self-contained way that this approximation theorem assumes that there is a perturbation in the known belief matrix (which in our work is generalized to the feature belief function) and studies how this affects the reward inference.**
> > - **We will closely read our work again and try to spot any place where we can make our work more self-contained, even if just to give the reader more confidence that our work can be understood independently.**
> >
> > We believe that any other issues that may make our work not self-contained enough are related to us not having been sufficiently clear about the motivation of our work, which we hope will be addressed below by our answers to your concerns on scalable oversight.
> >
> > > My biggest concern regarding this paper is that it is unclear how to use the key idea or knowledge for scalable oversight.
> >
> > Thank you for this feedback, we should address this and make the connection much clearer.
> >
> > Scalable oversight is the problem of how to oversee AI systems when they are more capable than the human overseers. Our problem setting does not *directly* model what it means for an AI system to be more capable than a human evaluator, but our setting *implicitly applies* to that setting. Namely, our human makes observations $o \in O$ and then forms a belief $\epsilon(o) \in \mathbb{R}^{\Omega}$, which is a vector of feature strengths for features $\omega \in \Omega$. This applies to a situation in which an AI system is more capable than the human evaluator for the following reasons:
> >
> > - *Scale of the environment*: We assume the policy $\pi: S \to \Delta(A)$ observes entire environment states $s \in S$, which can be at a larger scale than what the human observes. This is realistic for very capable future AI systems that have a greater capacity to oversee a large environment than humans.
> > - *Understanding of the observations*: In the future, we will train AI systems to perform tasks that are harder to perform than what the human could accomplish on their own. Thus, even if the environment were fully observed to the human (case $O = \Xi$), it can then be hard for the human to understand and to assess the quality of trajectories. This is modeled in our work by a feature belief function $\epsilon: O \to \mathbb{R}^{\Omega}$ that, even in the fully observable case, may be wrong: $\epsilon(\xi) \neq \lambda(\xi)$, where $\lambda: \Xi \to \mathbb{R}^{\Omega}$ is the true ontology that extracts the true feature strengths from the trajectories. As an example, imagine a setting where an AI is tasked with proving the Riemann hypothesis. Let $\xi$ be the AI’s resulting proof. Assume that $\lambda(\xi)$ contains features that measure the validity of steps in the proof. The human may not understand the proof attempt, leading to features $\epsilon(\xi) \neq \lambda(\xi)$.

---

> > > ### Author Response · Authors · 2025-06-16
> > > **Rebuttal 3/3**
> > >
> > > **We propose to adapt the paper by adding a new Section 2.5 (between what is currently Section 2.4 and Section 2.5) that makes this conceptual argument that partial observability and wrong human beliefs are (a) modeled by our setting and (b) a highly plausible setting for AI systems whose capabilities exceed those of the human evaluators – meaning that our work applies to scalable oversight. We will also emphasize that our work complements work on amplified oversight (cf. our Section 4.2; most work on scalable oversight falls under the amplified oversight umbrella) by *not improving* the human’s understanding and feedback, but by using a model of the human belief for learning a correct return function *despite human limitations*.**
> > >
> > > We believe another potential issue is that it may not be sufficiently clear what goes wrong when not making explicit use of belief models. I.e., while our paper provides a self-contained treatment of what belief models are and how they could in principle be used for a return function inference, it doesn’t yet explain well what goes wrong when  using a more naive reward function inference (as in classical RLHF). The reason we did not focus on this is that prior work already discusses this extensively, and we cited several such treatments in the introduction [5,7 - 10]. Yet I want to also acknowledge that not mentioning these issues further in our work may increase the perceived distance between our introduction and the main text that you lamented, and thus **we propose to also augment our new proposed Section 2.5 with a self-contained treatment of the conceptual failure modes to be expected under naive reward inference**, which we hope further motivates our definition of belief models that will follow in Section 2.6.
> > >
> > > **In summary, our upcoming Section 2.5 will close the connection between our introduction and the main text by**
> > > - **Explaining how our work applies to a setting of scalable oversight;**
> > > - **Synthesizing prior work on what goes wrong if human belief modeling is not attempted**;
> > > - **Building up a more precise motivation for the belief modeling that follows in Section 2.6 (which is named Section 2.5 in the current version)**
> > >
> > > > while I don't think extensive experiments are necessary, I believe that conducting some experiments in a setting close to real-world applications (using small LLMs or API) would significantly enhance the reader's understanding of the true value of this paper.
> > >
> > > Thank you for this suggestion. We would prefer to not add experiments as this is an extensive theoretical paper that we consider a complete investigation on its own. Empirical evidence that would go beyond what can already be found in prior work ([5, 7-10]) would be a significant amount of work. Yet we highly welcome any effort to conduct an extensive application of our ideas in practice, as sketched in our Section 3.4 and our future work, Section 4.3.2.
> > >
> > > —
> > >
> > > Thank you again for your thorough review! Please let us know whether our proposed changes would satisfy you and, in particular, whether there is anything else we can do to convince you (or make it so) that our paper provides evidence for all its claims.
> > >
> > > [1] Hong Jun Jeon, Smitha Milli, Anca D. Dragan, Reward-rational (implicit) choice: A unifying formalism for reward learning, NeurIPS 2020
> > >
> > > [2] Rohin Shah et al., Benefits of Assistance over Reward Learning, URL: https://people.eecs.berkeley.edu/~russell/papers/neurips20ws-assistance
> > >
> > > [3] Dylan Hadfield-Menell et al., Cooperative Inverse Reinforcement Learning, NeurIPS 2016
> > >
> > > [4] Scott Emmons et al., Observation Interference in Partially Observable Assistance Games, 2024, URL: https://arxiv.org/abs/2412.17797v1
> > >
> > > [5] Leon Lang et al., When Your AIs Deceive You: Challenges of Partial Observability in Reinforcement Learning from Human Feedback, NeurIPS 2024
> > >
> > > [6] Russell, Stuart and Norvig, Peter. Artificial Intelligence: A Modern Approach. 3 : Prentice Hall, 2010.
> > >
> > > [7] Dario Amodei, Paul Christiano, and Alex Ray. Learning from human preferences. https://openai.com/research/learning-from-human-preferences, 2017. Accessed: 2023-12-13.
> > >
> > > [8] Alex Cloud, Jacob Goldman-Wetzler, Evžen Wybitul, Joseph Miller, and Alexander Matt Turner. Gradient Routing: Masking Gradients to Localize Computation in Neural Networks, 2024. URL https://arxiv.org/abs/2410.04332.
> > >
> > > [9] Carson Denison et al.. Sycophancy to Subterfuge: Investigating Reward-Tampering in Large Language Models, 2024. URL https://arxiv.org/abs/2406.10162.
> > >
> > > [10] Jiaxin Wen, Ruiqi Zhong, Akbir Khan, Ethan Perez, Jacob Steinhardt, Minlie Huang, Samuel R. Bowman, He He, and Shi Feng. Language Models Learn to Mislead Humans via RLHF, 2024. URL https://arxiv.org/abs/2409.12822.
> > >
> > > [11] Piotr J. Gmytrasiewicz et al., A Framework for Sequential Planninging in Multi-Agent Settings, JAIR, 2005

---

> > > > ### Author Response · Authors · 2025-06-22
> > > > **Comment on the revisions**
> > > >
> > > > Dear reviewer,
> > > >
> > > > Thank you again for reviewing our work! We have now posted a revision. You can find notes on the precise updates at the top of this page. We particularly want to point you to Remark 2.3 (a comparison to other work involving partial observations, and also multi-agent settings) and Section 2.5 (an elaboration of how we relate to the problem of scalable oversight that should address several of your concerns).
> > > >
> > > > On the concrete changes to examples etc. to make it clearer that our work can be understood independently from Lang et al. (2024): Note that some numbers changed compared to my rebuttal. Example 2.4 is now 2.6; Example 2.9 is now 2.11; and Section 2.8 is now 2.9.
> > > >
> > > > We believe with these changes we have addressed all your concerns and hope that you are satisfied with the changes!

---

> > > > > ### Comment · Reviewer_YrGE · 2025-07-08
> > > > >
> > > > > Thank you for the responses. I am mostly satisfied with the revision and authors' rebuttal.

---

### Review · Reviewer_25yK · 2025-05-12

**Summary Of Contributions:**

The paper proposes a comprehensive and novel mathematical framework for preference inference/learning through human feedback. It considers an MDP setup for the reward modeling and introduces human belief model to formalize the human's "annotation" of the data. The paper mainly addresses the question of under what conditions, we can infer the reward function via human feedback, and proposes a full rigorous framework to characterize the mathematics behind the problem. The discussion comes with examples to better generate intuitions.

**Audience:**

Yes

**Broader Impact Concerns:**

Nan.

**Claims And Evidence:**

Yes

**Requested Changes:**

See above.

**Strengths And Weaknesses:**

I think the paper is well-written. It might not be easy to follow for people who don't have strong math background or who don't read it carefully, but I think the authors handled all the notations, setups, and theorems elegantly, and I appreciate the authors' efforts in this.

I don't have much question or requested change for the paper. But I'd like to raise a few points that arise when I read the paper. I hope the authors can think about these questions.

The paper is really mathematical. We are not short of empirical papers at all. The LLM/GenAI community does need mathematical papers to clarify the foundation of our current models and methods. But what would be the audience of the paper?

- For ML people, this paper is too technical for them to draw insights from, or I think the ML engineers working on the problem hardly understand the paper. For me, I want to be honest that given I think I am relatively on the theoretical side of the ML community, I tried hard and understood the contents of Section 2 but got lost in Section 3.2; the example of Section 3.3 is good. So if the authors want ML people to read and understand the paper (though you don't have to if you don't want), you need to write it in more plain language. Also, the MDP formulation of human feedback/RLHF is reasonable but over-complicates the understanding for the general ML audience. You don't have to rewrite the paper; just make one extra section in the very beginning, summarizing the key insights and take-away messages in layman's terms. And that would be great.

- I think the paper is a good example to attract more mathematicians or people with math background to work on LLM/GenAI-related problems. Just like this paper, I don't think we have a satisfactory mathematical definition of "hallucination". I'd love to see a paper on that topic, just like this one.

---

> ### Author Response · Authors · 2025-06-16
>
> Thank you for your review! Thank you for acknowledging that our paper is well-written and elegant despite possibly being difficult to read depending on background. **We will mention proposed changes in bold.**
>
> > For me, I want to be honest that given I think I am relatively on the theoretical side of the ML community, I tried hard and understood the contents of Section 2 but got lost in Section 3.2
>
> Thank you for raising this issue in accessibility. The high-level take-away is that “belief-model covering” (the notion that allows to “relax” the exact knowledge of a human belief model) has several sufficient conditions. The most important one we draw on in Section 3.4 is the existence of a “belief-compatible linear ontology translation”. **In an updated version, we will provide more intuitions on both morphisms and ontology translations to complement our currently purely formal description.**
>
> > the MDP formulation of human feedback/RLHF is reasonable but over-complicates the understanding for the general ML audience.
>
> While many contemporary reward learning problems (especially in RLHF) are simplified to a contextual bandit setting, we adopt the more general MDP framework. This is motivated by the increasing prevalence of multi-turn, agentic AI systems (including e.g. software agents), and it allows our approach to naturally speak to the sequential decision-making challenges inherent in scalable oversight for future AI systems.
> **In Section 2.2 where we introduce the setting, we will briefly clarify why we think this generality is necessary for our motivations.**
>
> On general accessibility for ML people:
>
> > You don't have to rewrite the paper; just make one extra section in the very beginning, summarizing the key insights and take-away messages in layman's terms
>
> Thank you, this is a good suggestion. We have to think about how to accomplish this precisely. **When adding our more precise description of how our work relates to our primary motivation — scalable oversight — in response to reviewer YrGE, we will also keep in mind how we can improve the framing of the introduction for better take-away messages.** Please take a look at our answer for YrGE to get a sense of how we want to clarify our contribution.
>
> Thank you again for your review. Please let us know what you think of the proposed changes and make us aware of anything else we should take into account when improving our paper.

---

> > ### Author Response · Authors · 2025-06-22
> > **Comment on the revisions**
> >
> > Dear reviewer,
> >
> > We thank you again for your time to review our work. We have now posted a revision of our work, for which you can read the changes at the top of this page.
> > Note that instead of updating the introduction for better take-away messages as promised, we made changes to the abstract and conclusion. We hope they will satisfy you and increase the clarity for a broad audience. We plan to also improve the clarity of the introduction in the future.
> >
> > We also hope that we addressed all your other points to your satisfaction.

---

### Review · Reviewer_MZRf · 2025-06-08

**Summary Of Contributions:**

This paper proposes a framework that models error in perception or belief in RLHF. The human is assumed to give feedback that is perfectly rational given their flawed understanding. This is known as scalable oversight in the literature.

To achieve this, this framework proposes turning trajectories into features and then using beliefs and rewards over those to capture the preferences of humans. However there are cases that capturing the preference of the human is problematic:

Ambiguity: when the value system (reward function) of the human is recoverable / identifiable from the feedback because many different reward functions could have produced the same feedback. The paper suggests that the solution is to have a set of observations for which the collection of human beliefs is rich enough to cover all the different true feature vectors that can be produced by the ontology.

Faithfulness: if any two value systems that are indistinguishable in reality are also indistinguishable in the human's feedback.

The authors define a "complete" belief model as one for which the ambiguity disappears. They provide a clear sufficient condition for completeness: a model is complete if the feature vector of any possible trajectory can be expressed as a linear combination of the belief vectors from the set of observations. This provides a clear target for designing practical feedback-gathering processes.

Recognizing that the true human belief model is unknowable, the paper introduces the crucial concept of "human belief model covering". The core idea is to specify a known belief model that is expressive enough to represent all possible return functions and feedback that could be generated by the true, unknown model. The authors show that if this covering model is also "complete," it enables the correct inference of the human's values from their feedback, providing a practical path forward for the paper's theoretical framework.

The paper bridges the gap between deep theory and practice by proposing a method to construct a covering model using foundation models. The novel idea is to use a powerful, state-of-the-art model to represent the ontology and a separate, belief-simulating model (e.g., an earlier checkpoint or a prompted version) to represent the feature belief function. This proposal, while speculative, provides a possible direction for future empirical research.

**Audience:**

Yes

**Broader Impact Concerns:**

-

**Claims And Evidence:**

Yes

**Requested Changes:**

**Clarify the Speculative Nature of the Practical Proposal**: The theoretical contributions of this paper are its strongest aspect. The proposal in Section 3.4 for instantiating the framework with foundation models is compelling but highly speculative, relying on the challenging task of creating a "belief-simulating" model that accurately mirrors human cognitive limitations. I would recommend the authors more clearly delineate the established theoretical results from this more forward-looking, hypothetical proposal. Adding a dedicated "Limitations" subsection to discuss the significant empirical hurdles in creating such a belief-simulating model would help manage reader expectations and highlight the most important avenues for future work.

**Expand on the Linearity Assumption in the Main Text**: The authors correctly note in the future work section that their framework assumes the human's return function is linear in the features. Given how central this assumption is, I believe the paper would benefit from a more prominent discussion of its implications in the main text. For example, many crucial human values (e.g., regarding fairness, safety, or avoiding catastrophic outcomes) might not be simple linear combinations of features. Briefly exploring why this assumption might be a significant limitation would provide important context for the scope of the current framework's applicability. Perhaps a suggestion - could a non-linear theory be exposed via the use of information theory (e.g. mutual information between ontology and beliefs)?

Nit: This paper reminds me a lot of POMDP-IRL however I understand that there are some differences (e.g. assumptions about optimality or deceptive nature of AI systems). I’d like to some some thoughts on that.

**Strengths And Weaknesses:**

**Weaknesses**

Writing and formalism: The paper is difficult to read because the mathematical formulas are defined once and not referenced with simple reminders in the text. For example “Even if the human could fully compute λ(ξ)…” would be more reader friendly if it was “Even if the human could fully compute the ontology of the trajectory ξ, λ(ξ)…”. Also, the algebraic and category theory terminology is inaccessible for the vast ML community, but I appreciate the effort the authors put in making these concepts as accessible as possible.

The Speculative Nature of a Belief-Simulating Model: The practical proposal in Section 3.4 hinges on the ability to construct a feature belief function that accurately simulates the human's beliefs and blind spots. The paper suggests methods like "belief prompting" or using earlier model checkpoints, but acknowledges this is highly speculative. As the authors note, it is unclear whether such techniques can reliably shape a model's internal representations to match a human's cognitive limitations, especially when other work shows models can have latent knowledge of the truth even when outputting falsehoods. This is the most significant leap of faith required for the proposal to work.

Lienear assumption: The entire framework, including the decomposition of rewards and the use of linear maps, assumes that human preferences are linear in a given feature space. The authors acknowledge this as a potential area for future work, but it is a significant limitation of the current theory. Many complex human values are likely non-linear, and it would strengthen the paper to discuss the potential implications of this assumption in more detail in the main text.

**Strengths**

The framework is appealing and mathematical rigorous: Besides the mathematical heaviness though, the paper is very well written with a lot of educational content to give context and bring the reader up to speed with the terminology. The introduction of "human belief models", the formal characterization of "ambiguity" in terms of linear algebra, and the precise conditions for its disappearance ("completeness")  are well-defined and well-argued. This formalism allows for a clear and unambiguous analysis of the problem.

The paper frames the scalable oversight problem not just as one of partial observability, but as one of flawed human belief. This is a crucial and insightful distinction. By proposing to model the discrepancy between an AI's actual behavior and what a human believes its behavior to be, the work provides a new and powerful lens through which to view and address potential AI deception and misalignment arising from human limitations.

Pragmatic Relaxation of Assumptions: A key strength is the move from the unrealistic assumption of a "known true model" to the more plausible goal of finding a "complete covering model". The introduction of belief model covering, morphisms, and ontology translations provides a sophisticated theoretical toolkit for reasoning about this more realistic scenario. This demonstrates a thoughtful approach to bridging the gap between pure theory and potential application.

---

> ### Author Response · Authors · 2025-06-16
> **Rebuttal part 1 of 2**
>
> Thank you for this thorough review! We appreciate you explaining the value that you got out of this paper, that you found this work rigorous and educational, and that you think it “provides a new and powerful lens” to analyze misalignment arising from human limitations.
>
> **We provide in bold our proposed changes to address your concerns**. We hope to address all your points below, but first we want to correct some potential misconception:
>
> > The human is assumed to give feedback that is perfectly rational given their flawed understanding. This is known as scalable oversight in the literature.
>
> This is not how we understand the scalable oversight problem. Scalable oversight is the problem of how to oversee AI systems when they are more capable than the human overseers. The human overseers may give rational feedback based on their understanding (and this is the setting we work in), but in general the problem is broader and can also encompass human actions that are not rational, as in [1]. For a general explanation of how we relate to scalable oversight, see our answer to reviewer YrGE. **In our upcoming Section 2.5 that we promised to reviewer YrGE, we will make sure to make clear what we mean with scalable oversight and what aspects of it our work does and does not address.**
>
> > Writing and formalism: The paper is difficult to read because the mathematical formulas are defined once and not referenced with simple reminders in the text.
>
> **When revising our work, we will make an effort to make all sections as self-contained as possible**, so thank you for this feedback. We appreciate that despite these obstacles, you do think that “the paper is very well written”.
>
> > Also, the algebraic and category theory terminology is inaccessible for the vast ML community, but I appreciate the effort the authors put in making these concepts as accessible as possible.
>
> We do think that the main paper should in principle be accessible to the ML community since the content is based on standard linear algebra (albeit formulated slightly more abstractly than what's typical in an ML context). Additionally, all our linear algebra notation and concepts are explained in Section 2.1. **We will slightly simplify the notation by removing the notation for the span and, possibly, for the scalar product** – probably it is easier for the readers to see these expressions spelled out in any context where they appear. **When revising the work, we will look for ways to make it more accessible.**
>
> Regarding category theory, note that the main paper is entirely free from it. All content that makes use of that language is in Appendix C, to be read by readers who appreciate that additional discussion.
>
> > I would recommend the authors more clearly delineate the established theoretical results from this more forward-looking, hypothetical proposal. Adding a dedicated "Limitations" subsection to discuss the significant empirical hurdles in creating such a belief-simulating model would help manage reader expectations and highlight the most important avenues for future work.
>
> Thank you, it is important for us to set the right expectations about our work. We do think that many places of our work are already clear about the state of the proposal, e.g. by calling it a “potential approach” (abstract), a “preliminary proposal” (introduction), and writing at the start of Section 3.4 “We hope our ideas can inspire future empirical work” and that the approach “leaves many open choices to be made” (Conclusion of the proposal in Sec. 3.4.4).
>
> Yet we agree that we could be clearer about the limitations. **We will replace Section 3.4.4 (Conclusion of the proposal) with a section titled “Limitations of the proposal” in which we explicitly list the ways in which its applicability is unclear, with the ability to model the human’s beliefs being the biggest current impediment. The section will still link to the future work, Section 4.3.2, where we will still recommend the same attempts to instantiate the proposal. We will also carefully review our language about the proposal in other parts of the paper to ensure consistency of the messaging.**

---

> > ### Author Response · Authors · 2025-06-16
> > **Rebuttal part 2 of 2**
> >
> > > Nit: This paper reminds me a lot of POMDP-IRL however I understand that there are some differences (e.g. assumptions about optimality or deceptive nature of AI systems). I’d like some thoughts on that.
> >
> > Yes, thanks for making us aware of inverse reinforcement learning in partially observable environments [2]. In that setting, an “expert” (in our case the human) also only partially observes the environment, and the goal is also to infer the human’s values from the human’s actions (in our case, from the human’s feedback). The most important difference:
> > In our setting, the human’s actions (feedback) are a response to the agent’s behavior in the environment; the human does not themselves act in the environment. In contrast, in this paper, and in IRL in general, the human’s actions take place in the environment over which the values are supposed to be learned (see e.g. [1] again for the fully observed case for a discussion on (sub)optimality of human actions in IRL).
> > While the settings of IRL and feedback are quite different, **we still acknowledge the similarities in key aspects of partial observability and value inference and will cite this work in an updated version.**
> >
> > > The authors correctly note in the future work section that their framework assumes the human's return function is linear in the features. Given how central this assumption is, I believe the paper would benefit from a more prominent discussion of its implications in the main text. For example, many crucial human values (e.g., regarding fairness, safety, or avoiding catastrophic outcomes) might not be simple linear combinations of features.
> >
> > We fully agree that we should expand on this! There is a subtle philosophical point, which is that there can be many different human models, and our theory is fully applicable if there is *some* human model that manages to linearly represent the human’s return function and observation return function, and such that we find a covering model of that human model. (Some details on this point are already in Remark 2.3).
> >
> > Thus, when you say “many crucial human values [...] might not be simple linear combinations of features”, we counter that it might in fact be possible to find some way to decompose the world into human-interpretable features such that values are linear. For example, consider the case of car safety. Assume we have two features, which encode the car's position and speed. With this information, we can likely not determine whether the car’s movement is safe in a linear way. However, if we additionally have a feature that tells us the speed limit at the car’s current position, then the speed limit minus the speed might be a reasonable proxy of some safety property! One does lose something, however: The more complex the features are that we add, the less likely we expect it to be that a foundation model could help to find a “covering model” as in Section 3.4, as it becomes more and more difficult to represent (or linearly translate to) the human’s complex features.
> >
> > **We will add a remark to Section 2.3 to clarify this point, acknowledging both the limitation while also making the point that we only need *some* human model in which the return function is linear. We will also cite work on e.g. IRL [3], where linear reward functions are often a common assumption, to contextualize our assumption in prior work.**
> >
> > Thank you again for your detailed review! Please let us know what you think of the proposed changes in bold, and anything else we should take into account to improve our paper.
> >
> >
> > [1] Joar Skalse et al., Partial Identifiability and Misspecification in Inverse Reinforcement Learning, arxiv, 2024. URL: https://arxiv.org/abs/2411.15951
> >
> > [2] Jaedeug Choi, Kee-Eung Kim, Inverse Reinforcement Learning in Partially Observable Environments, JMLR, 2011
> >
> > [3] Andrew Ng et al., Algorithms for Inverse Reinforcement Learning, ICML, 2000

---

> > > ### Author Response · Authors · 2025-06-22
> > > **Comment on the revisions**
> > >
> > > Dear reviewer,
> > >
> > > We thank you again for your time to review our work. We have now posted a revision of our work, for which you can read the changes at the top of this page. Some minor changes that we promised are still open, or we decided against doing them. These are:
> > > - We thought about removing the notation for the scalar product, but decided against doing so since we find it quite useful.
> > > - We still plan to make a full revision in which we will try to spot all places where we can explain notation in words, and other ways to make our work more accessible and locally self-contained.
> > >
> > > We hope that we addressed all your other points to your satisfaction.

---

### Decision · Action_Editor_fSrp · 2025-07-15

**Recommendation:** Accept as is

**Additional Comments:**

This is a theory paper that models that different humans perceive interaction trajectories with generative AI models differently. This difference is modeled by introducing observations.

This is a technically-sound paper on a timely topic. The authors exposed the high-level idea in a way that can be appreciated by a wider audience. Finally, although this is a theory paper, the authors make a practical proposal for implementing their approach with generative AI models. My main concern is that this paper does not contain any experiments. Therefore, its impact in today's world, which is largely experiment driven, will be limited. That being said, this is not an acceptance criterion at TMLR. The paper delivers on its claims and definitely will have audience. All reviewers lean towards accepting it and so do I.

**Audience:**

Yes

**Audience Explanation:**

This paper bridges two communities: it is a theory paper for learning human preferences, which has been central to reinforcement learning from human feedback. It does so only partially because there are no experiments. Therefore, while this paper will have audience, the impact will be limited because it is unclear if the theory is practical.

**Claims And Evidence:**

Yes

**Claims Explanation:**

This paper proposes a mathematic model for inferring the reward of trajectories with generative AI models in the setting where different humans may perceive them differently. This difference is modeled through observations. This is a theory paper without experiments and the authors are clear about it throughout the paper.